# OMNITEXT: A TRAINING-FREE GENERALIST FOR CONTROLLABLE TEXT-IMAGE MANIPULATION

**Agus Gunawan**[1][*]**, Samuel Teodoro**[1][*]**, Yun Chen**[1]**, Soo Ye Kim**[2]**, Jihyong Oh**[3][†]**, Munchurl Kim**[1][†]

[1] KAIST, [2] Adobe Research, [3] Chung-Ang University

agusgun@kaist.ac.kr, sooyek@adobe.com, jihyongoh@cau.ac.kr, mkimee@kaist.ac.kr

Project Page: `https://kaist-viclab.github.io/omnitext-site/`

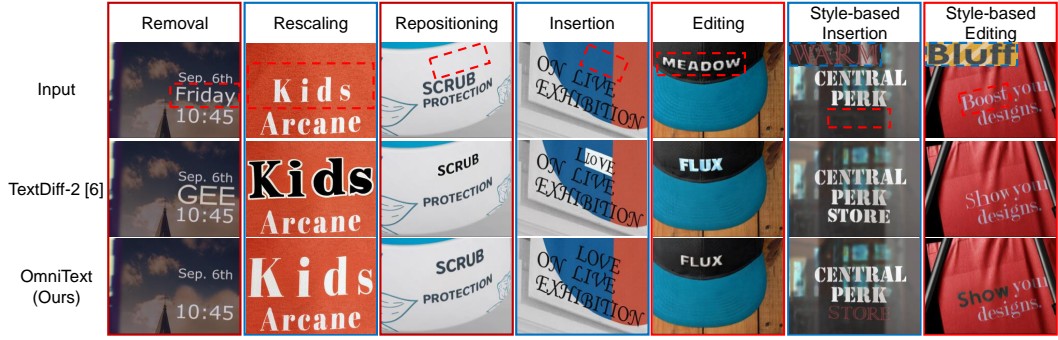

Figure 1: **Various text image manipulation applications on our OmniText-Bench using our proposed OmniText**. OmniText is a training-free generalist that can control both text content and styles during text rendering. Our OmniText allows for a wide range of applications including additional tasks (e.g., style-based insertion) that existing methods cannot achieve.

## ABSTRACT

Recent advancements in diffusion-based text synthesis have demonstrated significant performance in inserting and editing text within images via inpainting. However, despite the potential of text inpainting methods, three key limitations hinder their applicability to broader Text Image Manipulation (TIM) tasks: (i) the inability to remove text, (ii) the lack of control over the style of rendered text, and (iii) a tendency to generate duplicated letters. To address these challenges, we propose OmniText, a training-free generalist capable of performing a wide range of TIM tasks. Specifically, we investigate two key properties of cross- and self-attention mechanisms to enable text removal and to provide control over both text styles and content. Our findings reveal that text removal can be achieved by applying self-attention inversion, which mitigates the model's tendency to focus on surrounding text, thus reducing text hallucinations. Additionally, we redistribute cross-attention, as increasing the probability of certain text tokens reduces text hallucination. For controllable inpainting, we introduce novel loss functions in a latent optimization framework: a cross-attention content loss to improve text rendering accuracy and a self-attention style loss to facilitate style customization. Furthermore, we present OmniText-Bench, a benchmark dataset for evaluating diverse TIM tasks. It includes input images, target text with masks, and style references, covering diverse applications such as text removal, rescaling, repositioning, and insertion and editing with various styles. Our OmniText framework is the first generalist method capable of performing diverse TIM tasks. It achieves state-of-the-art performance across multiple tasks and metrics compared to other text inpainting methods and is comparable with specialist methods.

---

[*]Co-first authors (equal contribution). † Co-corresponding authors.

# 1 INTRODUCTION

Recent advances in large-scale text-to-image (T2I) models, particularly diffusion (Ho et al., 2020; Song et al., 2020; Dhariwal & Nichol, 2021; Rombach et al., 2022) and multi-modal models (Sun et al., 2024; Ge et al., 2024), have greatly improved image editing. These developments have led to generalist frameworks for diverse tasks, including object editing (Wei et al., 2025), style transfer (Han et al., 2024), and fine-grained editing tasks (e.g., object repositioning and rescaling) (Mou et al., 2024). However, generating accurate and coherent text within images remains challenging, even for recent closed-source models like GPT-4o-Image (Achiam et al., 2023) and Gemini 2.5 Flash-Image (Comanici et al., 2025), as well as open-source models without native text support (Labs, 2024; Tan et al., 2024; Kulikov et al., 2024; Avrahami et al., 2025) or with native text support (Wu et al., 2025) (see Appendix Section B). To address this, text-oriented T2I models (Tuo et al., 2023; 2024; Chen et al., 2023b; 2024) fine-tuned pre-trained latent diffusion models (LDMs) (Rombach et al., 2022) to generate visually appealing text that is coherent with the background.

Table 1: **Comparison of our OmniText with recent methods on TIM tasks**. **L** indicates that the method can perform the task, but with limitations such as a lack of control over the text styles.

| Methods | Venues | Text Insertion | Text Editing | Text Removal | Text Rescaling | Text Repositioning | Style Control |
|---|---|---|---|---|---|---|---|
| AnyText (Tuo et al., 2023) | ICLR '24 | L | L | ✗ | ✗ | ✗ | ✗ |
| TextDiff-2 (Chen et al., 2024) | ECCV '24 | L | L | ✗ | ✗ | ✗ | ✗ |
| DreamText (Wang et al., 2024) | CVPR '25 | L | L | ✗ | ✗ | ✗ | ✗ |
| TextSSR (Ye et al., 2024) | ICCV '25 | L | L | ✗ | ✗ | ✗ | ✗ |
| TextCtrl (Zeng et al., 2024) | NIPS '24 | ✗ | ✓ | ✗ | ✗ | ✗ | ✗ |
| VITEraser (Peng et al., 2024) | AAAI '24 | ✗ | ✗ | ✓ | ✗ | ✗ | ✗ |
| DARLING (Zhang et al., 2024) | CVPR '24 | ✗ | ✓ | ✓ | ✗ | ✗ | ✗ |
| OmniText (Ours) | - | ✓ | ✓ | ✓ | ✓ | ✓ | ✓ |

Text Image Manipulation (TIM) aims to seamlessly modify text within an image (Shu et al., 2024) and commonly consists of three tasks: (i) removal: erasing text and restoring the background; (ii) editing: modifying text while preserving its style; and (iii) insertion: adding new text that blends naturally with the background. Due to the distinct goals of each task, prior works have developed specialized methods for text removal (Liu et al., 2022; Zhang et al., 2024), editing (Zeng et al., 2024; Fang et al., 2025), and insertion (Chen et al., 2024; Tuo et al., 2023; 2024). While generalist frameworks exist for object-level image editing (Wei et al., 2025), no generalist TIM framework has been proposed to address these diverse tasks (see Table 1). Such a generalist approach is particularly valuable for applications like poster design (Gao et al., 2023; Wang et al., 2025b).

In this work, we propose OmniText, a generalist framework for diverse TIM tasks (see Fig. 1), including common tasks like removal, editing, and insertion, as well as tasks not jointly addressed in previous works like *text style transfer, rescaling, and repositioning*. Our method builds on recent text-oriented T2I models (Chen et al., 2024; Tuo et al., 2023; Zhao & Lian, 2023; Wang et al., 2024), adopting TextDiff-2 (Chen et al., 2024) as the backbone for its simplicity, its similarity to a vanilla diffusion model, and its support for text insertion and editing. TextDiff-2 combines a Language Model for layout planning with a latent diffusion model (Rombach et al., 2022) for text generation and is fine-tuned for inpainting-based text insertion within specified masks.

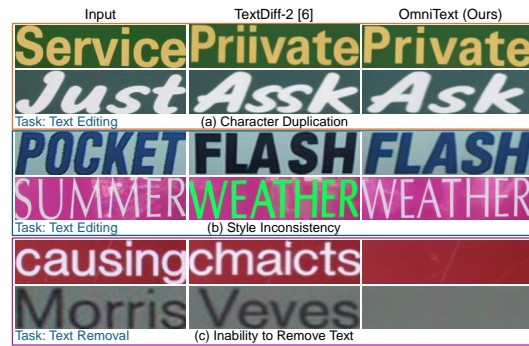

Figure 2: TextDiff-2 limitations.

Despite its strengths, TextDiff-2 (Chen et al., 2024) has key limitations that hinder its use as a generalist for diverse TIM tasks (see Fig. 2): (i) *Character Duplication*: Edited regions occasionally exhibit duplicated characters. (ii) *Uncontrollable Style Fidelity*: The method lacks explicit text style control. (iii) *Inability to Perform Text Removal*: Given a blank text as input, the model hallucinates text instead of removing it. To address these limitations, we first analyze the cross- and self-attention maps of TextDiff-2 to identify key properties essential for enabling diverse TIM tasks. First, self-attention influences style: when generating masked text, high responses to surrounding text lead to style transfer. Second, cross-attention controls content alignment: individual character tokens attend to distinct spatial regions. Third, strong self-attention responses to surrounding text in early sampling

cause text-like artifacts during text removal. In addition, we find that redistributing cross-attention probability from one text token to another reduces hallucinations in rendered text.

Building on these insights, we propose OmniText, a training-free framework capable of performing both standard TIM tasks (e.g., text removal, editing, and insertion) and less explored ones (e.g., text style transfer, repositioning, and rescaling). To achieve this, we introduce two core components: (i) Text Removal: we propose self-attention inversion and cross-attention reassignment to suppress the backbone's text generation ability, enabling effective text removal. (ii) Controllable Inpainting: we adopt a latent optimization framework guided by carefully designed loss functions to control both text content and style. For style control, we use a grid-based input (inspired by the grid trick (Kara et al., 2024)) and a self-attention style loss based on KL divergence. For content control, we propose a cross-attention content loss based on Focal Loss (Lin et al., 2017). Thanks to its modular design, OmniText supports a wide range of TIM tasks by simply adjusting its components.

Our main contributions are as follows: (i) We introduce **OmniText**, the *first training-free method* to jointly enable explicit control over style fidelity, text content, and text removal. (ii) To the best of our knowledge, this is the *first generalist method* supporting diverse TIM tasks, including style-controlled text insertion and editing, removal, repositioning, and rescaling. (iii) We present **OmniText-Bench**, a mockup-based evaluation dataset consisting of 150 sets of input images, target texts with masks, reference images, and ground-truth, covering five distinct applications.

## 2 RELATED WORK

**Text Image Manipulation.** Common tasks in Text Image Manipulation (TIM) include text removal, editing, and insertion (Shu et al., 2024). Text removal segments and removes text regions, then reconstructs the background, using GANs (Goodfellow et al., 2014) with multi-task decoders (Liu et al., 2020; Wang et al., 2023; Tang et al., 2021), transformers (Liu et al., 2022), and masked image modeling (Peng et al., 2024). Text editing modifies content while preserving style and background, with recent approaches leveraging GANs (Qu et al., 2023), diffusion models (Ji et al., 2023; Santoso et al., 2024; Zeng et al., 2024; Chen et al., 2023a), and multi-modal language models (Fang et al., 2025). With the rise of text-oriented text-to-image (T2I) models (Chen et al., 2023b; 2024; Tuo et al., 2023; 2024), text insertion (Tuo et al., 2023; 2024; Chen et al., 2023b; 2024; Zhao & Lian, 2023; Wang et al., 2024) has gained attention, aiming to generate realistic text at specified locations that blends with the background. However, unlike general image editing, TIM remains limited to text removal, editing, and insertion. Tasks such as text style transfer, repositioning, and rescaling remain underexplored due to the lack of fine-grained style control. Although prior works (Yang et al., 2020; Zeng et al., 2024) have attempted style transfer, they either alter both text and background (Yang et al., 2020), or cannot utilize a separate style reference to modify only the text style while preserving the original background (Zeng et al., 2024). To the best of our knowledge, we propose the first generalist capable of addressing diverse TIM tasks.

**Large-scale Models for Text Generation and Inpainting.** Large-scale text-oriented T2I generation has recently gained attention, driven by the limitations of diffusion models in generating accurate and coherent text (Chen et al., 2023b). Recent approaches can be broadly grouped into two categories: (i) Encoder-based (Chen et al., 2023b; 2024; Zhao & Lian, 2023; Wang et al., 2024), which train a text-encoder to learn text-level representations for accurate text insertion into specified masks; (ii) ControlNet (Zhang et al., 2023)-based architectures (Tuo et al., 2023; 2024; Yang et al., 2023), which use glyph-rendered images or multiple control conditions (e.g., glyph images, masks, fonts, colors) either within the text embedding or as control signals. These methods support both text insertion and editing, either directly (Tuo et al., 2023; 2024) or via fine-tuning for inpainting tasks (Chen et al., 2023b; 2024; Zhao & Lian, 2023; Wang et al., 2024).

**Attention Control in Training-free Image Editing with Diffusion Models**. TIM can be considered a subset of image editing (Zeng et al., 2024). Recent advances in large-scale T2I generation, particularly diffusion models (Rombach et al., 2022), have enabled diverse editing applications, including text-conditioned editing (Hertz et al., 2022) and style transfer (Chung et al., 2024). Recent works on training-free methods (Huberman-Spiegelglas et al., 2024; Wallace et al., 2023; Hertz et al., 2022; Cao et al., 2023; Tumanyan et al., 2023; Couairon et al., 2022; Mirzaei et al., 2024; Li et al., 2024) enable image editing without retraining or fine-tuning, with attention manipulation emerging as a central technique. It leverages the properties of cross- and self-attention to guide edits, either

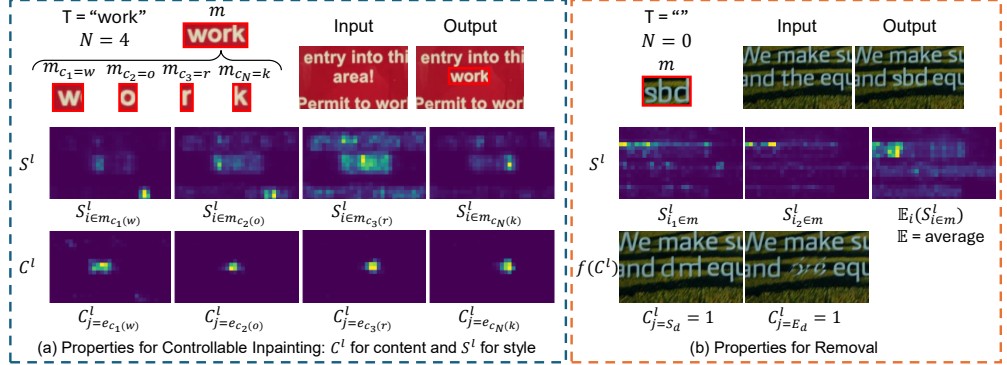

Figure 3: **Key attention properties for controllable inpainting and removal** at sampling step $t = 751$ and at Decoder Block 2, Layer 0. $f$ stands for cross-attention manipulation.

during sampling or via latent optimization, where attention serves as the reward function (Chefer et al., 2023; Phung et al., 2024). These methods manipulate self- and/or cross-attention to control various aspects of editing, including content (Hertz et al., 2022; Li et al., 2024), structure (Tumanyan et al., 2023; Parmar et al., 2023), and style (Deng et al., 2023; Chung et al., 2024). In this work, we propose a novel attention manipulation strategy for text removal, along with loss functions that use cross-attention for content control and self-attention for style control.

## 3 METHOD

In this section, we describe OmniText, which leverages the TextDiff-2 (Chen et al., 2024) inpainting model as the backbone. First, we analyze key properties of self- and cross-attention that enable text removal and controllable inpainting in Section 3.1. Based on these insights, we propose two key components, text removal and controllable inpainting, described in Section 3.2.

### 3.1 KEY ATTENTION PROPERTIES FOR TEXT REMOVAL AND CONTROLLABLE INPAINTING

Our backbone, TextDiff-2, has two key limitations: (i) inability to perform text removal and (ii) lack of direct control over text content and style. To overcome these, we analyze attention behavior during sampling, focusing on the attention probability map for layer $l$ at sampling step $t$, defined as:

$$A^{l,t} = \text{softmax}(\mathbf{Q}\mathbf{K}^\top/\sqrt{d}) \in [0,1]^{hw \times n} \tag{1}$$

where $d$ is the feature dimension, the query $\mathbf{Q} \in \mathbb{R}^{hw \times d}$ is computed from feature map $h \times w$, while the key $\mathbf{K} \in \mathbb{R}^{n \times d}$ depends on whether the layer performs cross-attention $C^{l,t}$ or self-attention $S^{l,t}$. For clarity, we omit the timestep $t$ hereafter. In cross-attention $C^l$, the key is computed from the text embedding $e$. The text input $T$, with $N$ characters $\{c_k\}_{k=1}^N$, is tokenized into three segments with start ($S$) and end ($E$) tokens for each segment: an empty description ($[S_d \ E_d]$), a text description ($[S_T \ c_1 \ ... \ c_N \ E_T]$), and padding ($[P \ ...]$). These tokens are linearly projected to obtain $e \in \mathbb{R}^{n \times d}$, where $n$ is the embedding length. The cross-attention map $C^l \in \mathbb{R}^{hw \times n}$ measures the association between spatial location $i$ in the feature map and token $j$ in the text embedding $\{e_j\}_{j=1}^n$. In self-attention $S^l \in \mathbb{R}^{hw \times hw}$, the key is computed from the feature map, allowing the model to capture relationships between spatial locations $i$ and $j$ in the feature map.

We identify two key properties of the attention maps $S^l$ and $C^l$ (Fig. 3). **First**, for controllable inpainting (Fig. 3 (a)), in certain cross-attention layers $C^l$, each character token $C^l_{j=e_{c_k}}$ attends to its corresponding spatial region $m_{c_k}$, suggesting that text content can be controlled via cross-attention. In self-attention layers $S^l$, tokens within a character region $S^l_{i \in m_{c_k}}$ attend to nearby text or similar characters, enabling style transfer or character-level replication, making self-attention effective for style control. **Second**, for text removal (Fig. 3 (b)), self-attention ($S^l_{i \in m}$) may still attend to nearby text, causing hallucinations even with an empty prompt ($T =$ ""). Moreover, increasing cross-attention of the start-of-description token ($C^l_{j=S_d}$) amplifies hallucinations and background reconstruction, while boosting the end-of-description token ($C^l_{j=E_d}$) helps suppress hallucinations.

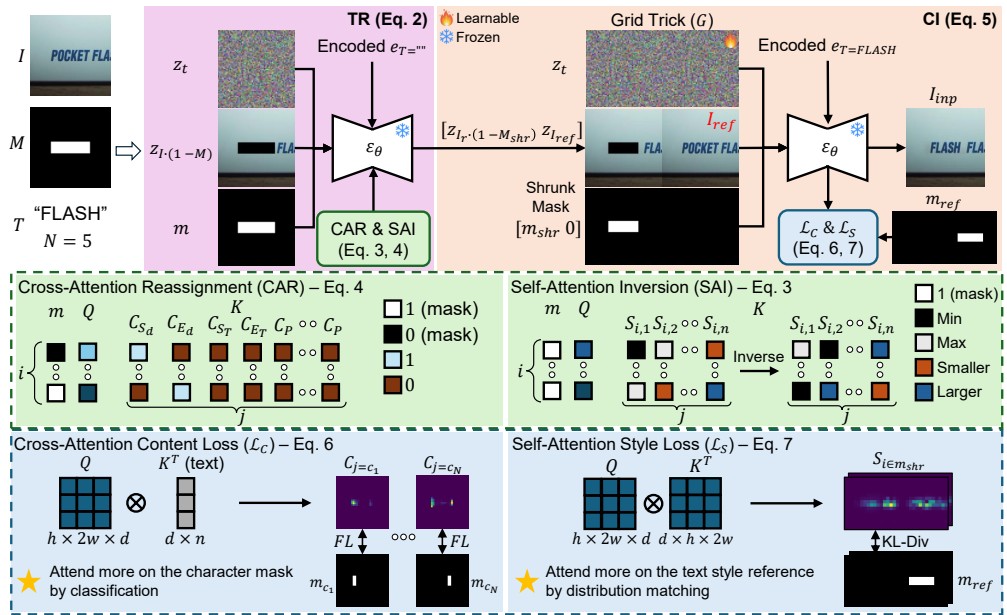

Figure 4: **Overview of OmniText for text editing**. First, we perform Text Removal (TR) by modulating attention with our proposed CAR and SAI during sampling. Then, we apply Controllable Inpainting (CI) using a latent optimization strategy with content loss $\mathcal{L}_C$ and style loss $\mathcal{L}_S$ to control content and style, respectively.

These insights form the basis of OmniText, enabling fine control over text content and style, and effective text removal.

## 3.2 TEXT REMOVAL (TR) AND CONTROLLABLE INPAINTING (CI)

Our OmniText framework is built on two core components: text removal and controllable inpainting, derived from the attention properties analyzed in Sec. 3.1. As illustrated in Fig. 4, these components are applied to tasks such as text editing. Given an input image $I$, a target mask $M$, and text $T$, the inpainting process for generating the text removal output $I_r$ is defined as:

$$I_r = TR_{\epsilon_\theta}(z_t, z_{I \cdot (1-M)}, m, e_T) \tag{2}$$

where $z_t$ is Gaussian noise, $z_{I \cdot (1-M)}$ is the latent representation of the masked image, $m$ is the target latent mask obtained via bilinear interpolation, and $e_T$ is the text embedding. For text removal, the prompt is set to blank ($T = $ ""). During sampling, as discussed in Sec. 3.1, the model's self-attention often attends to surrounding text, which leads to hallucinated text. To mitigate this, we propose Self-Attention Inversion (SAI), which inverts the self-attention values within the text mask $S_{i \in m}^l$ by linearly mapping the minimum value to the maximum and vice versa, defined as:

$$S_{i,j}^l = max_j(S_{i,j}^l) + min_j(S_{i,j}^l) - S_{i,j}^l \tag{3}$$

This encourages the model to focus on the background rather than surrounding text. However, in some cases, the network's self-attention may fail to attend to surrounding text, likely due to weak detection signals, making SAI alone insufficient for effective text removal. To address this, we propose Cross-Attention Reassignment (CAR), a complementary strategy that enhances suppression by reassigning the attention probability map using a piecewise function:

$$C_{i,j}^l = \begin{cases} 1, \text{if } (i \in m \text{ and } j = E_d) \text{ or } (i \notin m \text{ and } j = S_d) \\ 0, \text{otherwise} \end{cases} \tag{4}$$

This function reconstructs the non-inpainted region ($i \notin m$) and suppresses text generation in the inpainted region ($i \in m$). This design is based on our observation that setting $C_{i,j=S_d}^l = 1$ promotes background reconstruction, while setting $C_{i,j=E_d}^l = 1$ encourages text removal.

**Controllable Inpainting (CI)**. For controllable inpainting, we propose a method that enables control over both text content and style. Given the text removal output $I_r$, target text $T$, target mask $M$ with bilinearly interpolated latent mask $m$, reference image $I_{ref}$, and its corresponding mask $m_{ref}$ (where $I_{ref} = I$ and $m_{ref} = m$ in text editing), we construct a grid structure $G$ inspired by its use in video editing (Kara et al., 2024).

The grid structure guides self-attention toward the reference text for effective style transfer, as existing self-attention-based style transfer methods (Chung et al., 2024) are not directly applicable (see Appendix). Specifically, $G$ includes a grid latent, a grid masked latent $[z_{I_r \cdot (1-M_{shr})} \; z_{I_{ref}}]$, and a grid latent mask $[m_{shr} \; 0]$. However, when the target text (e.g., "FLASH") is shorter than the input (e.g., "POCKET"), naively copying the style may cause character duplication. To address this, we shrink the mask to obtain $M_{shr}$ and $m_{shr}$ using character-width priors (e.g., "W" is wider than "I"). Our controllable inpainting process then generates the final output $I_{inp}$, formulated as:

$$I_{inp} = CI_{\epsilon_\theta}(z_t, [z_{I_r \cdot (1-M_{shr})} \; z_{I_{ref}}], [m_{shr} \; 0], e_T) \tag{5}$$

To enable both content and style control, we perform on-the-fly optimization of latent variable $z_t$ during early sampling steps. We update $z_t$ using the Adam optimizer (Kingma & Ba, 2014) as: $z'_t \leftarrow \text{Adam}(\nabla_{z_t} \mathcal{L}(z_t))$, where the total loss $\mathcal{L}$ combines two novel terms: Cross-Attention Content Loss ($\mathcal{L}_C$) and Self-Attention Style Loss ($\mathcal{L}_S$), weighted by hyperparameters $\lambda_C$ and $\lambda_S$: $\mathcal{L} = \lambda_C \mathcal{L}_C + \lambda_S \mathcal{L}_S$.

**Cross-Attention Content Loss ($\mathcal{L}_C$)**. Cross-attention $C^l_{i,j}$ plays a key role in controlling text content during inpainting (see Sec. 3.1). To enforce correct content placement, we formulate the task as a binary classification problem, where for each character $c_k$, the cross-attention probability $C^l_{j=c_k}$ should be high in the corresponding character region $i \in m_{c_k}$ and low in other regions $i \notin m_{c_k}$. To obtain the per-character masks ($m_{c_k}$), we divide the target text mask $m$ equally among all characters in the target text. While binary cross-entropy loss could be used, it suffers from class imbalance, as positive samples ($i \in m_{c_k}$) are much fewer than negatives ($i \notin m_{c_k}$). To mitigate this, we adopt Focal Loss ($FL$) (Lin et al., 2017), which is specifically designed to handle imbalanced-classification problems. We define our Cross-Attention Content Loss $\mathcal{L}_C$ with focusing parameter $\gamma$ as:

$$\mathcal{L}_C = \Sigma_{k=1}^{N} FL(C^l_{i,j=c_k}, m_{c_k}); FL(p,l) = (1 - (p \cdot l))^\gamma \cdot -(l \cdot \log(p) + (1 - l) \cdot \log(1 - p)) \tag{6}$$

**Self-Attention Style Loss ($\mathcal{L}_S$)**. We observe that self-attention plays a key role in controlling the style of rendered text during inpainting. To leverage this, we propose a Self-Attention Style loss that encourages the network to attend to the text mask of reference image $m_{ref}$. We formulate this as a distribution matching problem using KL-divergence, aligning the self-attention probability map $S^l_{i \in m,j}$ within the target latent mask $m$ with a ground-truth (GT) derived from normalized $m_{ref}$:

$$\mathcal{L}_S = D_{KL}(GT, S^l_{i \in m}); GT = m_{\text{ref}}/\Sigma_{j=1}^{N}(m_{\text{ref}})_j \tag{7}$$

To address resolution mismatches, we bilinearly interpolate $m_{ref}$ to match the resolution of the self-attention map. By aligning self-attention with $m_{ref}$, the network is guided to copy the style of the reference text, ensuring stylistic consistency in the inpainted result.

## 4 Experiments and applications

### 4.1 Dataset and metrics

**OmniText-Bench (Ours).** Currently, no existing benchmark covers the full range of Text Image Manipulation (TIM) tasks. To address this, we introduce OmniText-Bench, a mockup-based dataset for evaluating text style fidelity and rendering accuracy across five tasks: insertion, editing, removal, repositioning, and rescaling. To assess style transfer, we also provide alternate style references for insertion and editing. The dataset includes 150 mockups on diverse surfaces (e.g., print, apparel, packaging, devices), each manually edited to support all tasks, with ground-truth for precise evaluation of both style fidelity and content accuracy. Additional details are provided in the Appendix Section E.

**Evaluation Dataset.** To evaluate standard TIM tasks, we use SCUT-EnsText (Liu et al., 2020) for text removal (following Zhang et al. (2024)) and ScenePair (Zeng et al., 2024) for text editing (following (Fang et al., 2025)). For text removal, we replace the unedited regions with the original image. To evaluate additional applications beyond removal and editing, we use our proposed OmniText-Bench.

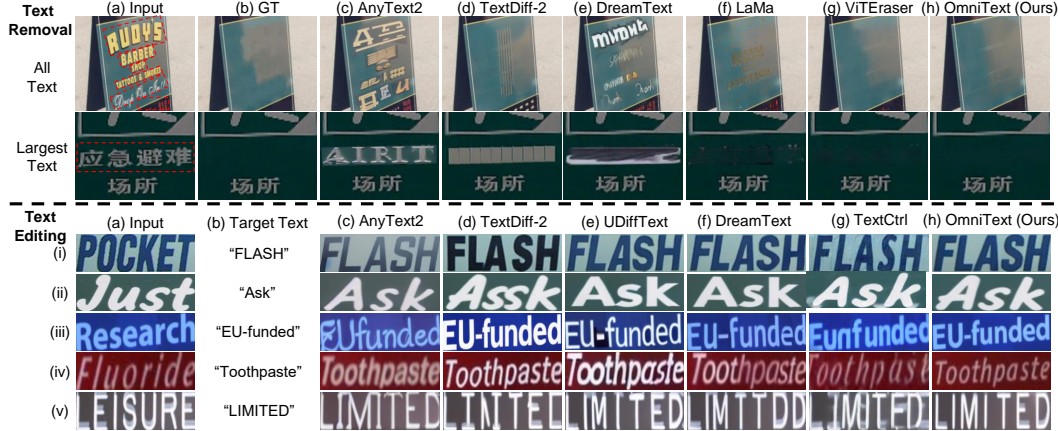

Figure 5: **Qualitative comparison on standard benchmark**. We present visual comparisons for both all text and largest text settings, with some baselines excluded due to their poor performance. A complete comparison is provided in the Appendix Section J.1.

Table 2: **Quantitative comparison on standard text removal benchmark (SCUT-EnsText)**. **Bold** and underline denote the best and second-best performance in each category. **Red** denotes the best performance across all categories.

| Metrics
Methods | All Text Removal | | | | Largest Text Removal | | | |
|---|---|---|---|---|---|---|---|---|
| | MS-SSIM ↑ ($\times 10^{-2}$) | PSNR ↑ | MSE ↓ | FID ↓ | MS-SSIM ↑ ($\times 10^{-2}$) | PSNR ↑ | MSE ↓ | FID ↓ |
| AnyText (Tuo et al., 2023) | 92.11 | 23.38 | 9.70 | 69.55 | 95.93 | 26.32 | 5.23 | 29.63 |
| AnyText2 (Tuo et al., 2024) | 88.64 | 21.42 | 12.91 | 84.58 | 95.75 | 25.66 | 6.10 | 30.13 |
| TextDiff-2 (Chen et al., 2024) | 92.73 | 25.42 | 9.51 | 52.46 | 96.27 | 28.52 | 5.19 | 21.64 |
| UDiffText (Zhao & Lian, 2023) | 88.12 | 18.74 | 27.48 | 75.25 | 94.89 | 21.66 | 12.58 | 34.44 |
| DreamText (Wang et al., 2024) | 85.80 | 19.56 | 20.65 | 73.84 | 94.82 | 23.37 | 9.09 | 29.69 |
| OmniText (Ours) | 95.71 | 29.52 | 3.44 | 39.06 | 98.21 | 33.90 | 1.61 | 15.33 |
| (Specialist) LaMa (Suvorov et al., 2022) | 93.93 | 29.37 | 2.91 | 43.67 | 97.40 | 32.80 | 1.62 | 15.83 |
| (Specialist) ViTEraser (Peng et al., 2024) | 96.55 | 34.12 | 1.14 | 28.35 | 98.83 | 40.36 | 0.44 | 8.59 |

**Evaluation Metrics.** For text removal, we follow (Zhang et al., 2024) and evaluate full images with PSNR, MS-SSIM ($\times 10^{-2}$), and FID (Heusel et al., 2017). For text editing, following (Zeng et al., 2024), we evaluate cropped regions for text style fidelity (MSE $\times 10^{-2}$, PSNR, MS-SSIM, FID) and rendering accuracy using ACC (word accuracy) and NED (normalized edit distance) from a text recognition model (Baek et al., 2019).

**Baselines.** We group baselines into generalist and specialist methods. For generalists, we compare OmniText with recent diffusion-based text synthesis models: AnyText (Tuo et al., 2023), AnyText2 (Tuo et al., 2024), TextDiff-2 (Chen et al., 2024), UDiffText (Zhao & Lian, 2023), and DreamText (Wang et al., 2024). For specialists, we include task-specific models: LaMa (Suvorov et al., 2022) and the recent SOTA ViTEraser (Peng et al., 2024) for text removal, and TextCtrl (Zeng et al., 2024) for text editing, serving as upper bounds. We exclude other underperforming text synthesis models (see Appendix Section B.2) and RS-STE (Fang et al., 2025), because part of ScenePair was used in its training.

## 4.2 PERFORMANCE COMPARISON ON TEXT REMOVAL AND EDITING

In this comparison, we evaluate OmniText on standard benchmarks for text removal (SCUT-EnsText (Liu et al., 2020)) and text editing (ScenePair (Zeng et al., 2024)). Hyperparameter details of OmniText are provided in Appendix Section F, where we also include best practices for hyperparameter selection in Appendix Section G. We additionally provide a runtime analysis in Appendix Section I.

**Text Removal.** We compare our text removal ($TR_{\theta_\epsilon}$) with recent text synthesis methods under two settings: 'all text' (removing all visible text) and 'largest text' (removing only the largest text). Because latent diffusion models cannot faithfully reconstruct unmasked regions, we follow Zeng et al. (2024) and replace unedited regions with the original input.

As shown in Table 2, OmniText outperforms other generalists in both settings. Qualitative results in Fig. 5 further show that generalist methods often fail to remove text effectively and instead hallucinate

Table 3: **Quantitative comparison on standard text editing benchmark (ScenePair).** **Bold** and underline denote the best and second-best performance in each category. **Red** denotes the best performance across all categories.

| | Rendering Accuracy | | Style Fidelity | | | |
|---|---|---|---|---|---|---|
| | ACC (%) ↑ | NED ↑ | MSE ↓ ($\times 10^{-2}$) | MS-SSIM ↑ ($\times 10^{-2}$) | PSNR ↑ | FID ↓ |
| AnyText (Tuo et al., 2023) | 60.70 | 0.878 | 5.87 | 30.40 | 13.83 | 40.89 |
| AnyText2 (Tuo et al., 2024) | 71.72 | 0.918 | 5.71 | 30.47 | 13.80 | 41.97 |
| TextDiff-2 (Chen et al., 2024) | 76.41 | 0.944 | 6.49 | 35.56 | 13.62 | 29.86 |
| UDiffText (Zhao & Lian, 2023) | 78.36 | 0.954 | 7.55 | 29.51 | 12.58 | 32.87 |
| DreamText (Wang et al., 2024) | 66.88 | 0.921 | 6.24 | 29.46 | 13.59 | 28.33 |
| OmniText (Ours) | 78.44 | 0.951 | 4.79 | 40.11 | 14.85 | 31.69 |
| (Specialist) TextCtrl (Zeng et al., 2024) | 78.98 | 0.917 | 4.58 | 37.93 | 14.92 | 31.95 |

text or textures, whereas OmniText produces cleaner output with fewer artifacts. We also compare with specialist methods: LaMa, a general inpainting model, and ViTEraser, a recent SOTA model trained specifically for text removal. As expected, ViTEraser outperforms all generalist models, including ours. Nevertheless, OmniText surpasses the specialist LaMa in MS-SSIM, PSNR, and FID, and qualitatively delivers more precise, context-aware removal, whereas LaMa often introduces artifacts from contextual confusion when nearby text is unmasked (see Fig. 5 (f)). Unlike these specialist methods, OmniText is a generalist that also supports other tasks, e.g., text editing.

**Text Editing.** We compare OmniText ($TR_{\theta_\epsilon}$ and $CI_{\theta_\epsilon}$ with $\lambda_C = 5$ and $\lambda_S = 10$, and the input image as the CI reference) against other text synthesis baselines (generalists). As shown in Table 3, OmniText outperforms all generalist models across accuracy metrics, remaining competitive on NED. For style fidelity, it achieves the best results on all metrics except FID, which is less reliable as it relies on features from networks trained on natural rather than text-specific images. Qualitative results in Fig. 5 support these findings: OmniText renders text more accurately and preserves style consistency, while baselines often distort style characteristics. In challenging cases (Fig. 5 iv, v), OmniText maintains subtle font attributes, including orientation (slight tilt) in (iv) and shadow effects in (v), whereas baselines introduce artifacts or fail to replicate the style.

We further compare OmniText with TextCtrl (Zeng et al., 2024), a specialist method trained for text editing. OmniText outperforms TextCtrl in NED, MS-SSIM and FID, while achieving comparable accuracy, MSE, and PSNR. OmniText's higher NED but slightly lower accuracy suggest occasional duplicated characters, likely due to the backbone's limitations in handling large fonts and spacing between characters (see Appendix Section O). In terms of structural fidelity (MS-SSIM), OmniText surpasses TextCtrl in preserving font styles, as shown in Fig. 5 (i), (ii), (iv), and (v). By contrast, TextCtrl introduces noticeable artifacts in Fig. 5 (i), (ii), and (v), and struggles with style replication, partly due to its self-attention manipulation strategy, which fuses key and value features from the reconstruction and editing branches, limiting flexibility in style transfer. Moreover, in longer-text cases (e.g., Fig. 5 iv), OmniText adaptively adjusts character sizes to fit the mask, whereas TextCtrl often generates text that exceeds the mask boundaries.

## 4.3 Additional applications and comparison on OmniText-Bench

We present four additional applications of Text Image Manipulation (TIM): text rescaling, repositioning, style-based insertion, and style-based editing. Full comparisons with additional baselines and quantitative results are available in the Appendix Section J. In addition, we include further results on challenging cases, such as long-text manipulation, manipulation on complex textures, and low-resolution input images, in Appendix Section M. For rescaling, repositioning, and style-based editing, we use both $TR_{\theta_\epsilon}$ and $CI_{\theta_\epsilon}$; for style-based insertion, only $CI_{\theta_\epsilon}$ is used. We set $\lambda_C = 5$ and $\lambda_S = 10$, and use the provided style image as the reference. Since UDiffText does not support text removal, we apply LaMa (Suvorov et al., 2022) for tasks that require it (e.g., repositioning). As there are no existing baselines for these tasks, we evaluate the generalizability of our grid trick, which guides self-attention to focus on the text reference, by integrating it into UDiffText. Due to UDiff-Text's specialized architecture, latent optimization is not applicable. Instead, we apply self-attention modulation during sampling, denoted as UDiffText + Mod (Ours). We include additional discussion on generalization to other backbones in Appendix Section D.

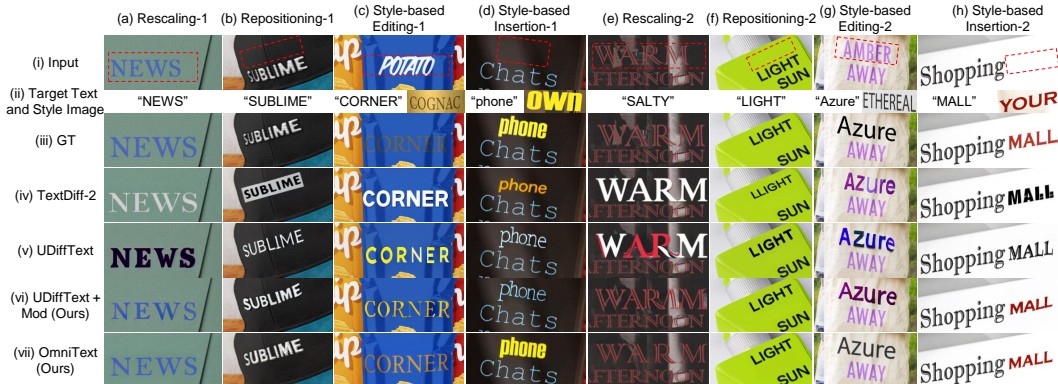

Figure 6: **Qualitative comparison on additional applications with OmniText-Bench**. The additional applications include text rescaling, repositioning, and style-based editing and insertion.

As shown in Fig. 6, text synthesis baselines such as TextDiff-2 and UDiffText lack style control and instead rely on surrounding text (Fig. 6 f, g, h) to maintain style consistency. This leads to noticeable font style changes when no surrounding text is present (Fig. 6 a, b, c). Our modulation improves UDiffText's ability to transfer font type and color in simpler cases with flat surfaces and orientations (Fig. 6 a, b, c, e, h), demonstrating the generalizability of our grid trick and self-attention for style control. However, despite the improvement, UDiffText + Mod still suffers from color (Fig. 6 c) and structural distortions (Fig. 6 e, h), and struggles to replicate style when the reference font significantly differs from the input (Fig. 6 d, g). In contrast, OmniText consistently transfers both font color and type across simple and challenging cases (Fig. 6 a-h), highlighting the effectiveness of our grid trick combined with latent optimization and style loss.

## 4.4 ABLATION STUDY

Table 4: **Ablation study** on Text Removal components using SCUT-EnsText.

|  | All Text | | | Largest Text | | |
|---|---|---|---|---|---|---|
|  | SSIM ↑ | PSNR ↑ | FID ↓ | SSIM ↑ | PSNR ↑ | FID ↓ |
| TextDiff-2 | 92.73 | 25.42 | 52.46 | 96.27 | 28.52 | 21.64 |
| + SAI | 95.56 | 30.02 | 37.31 | 97.58 | 33.12 | 16.70 |
| + SAI + CAR | 95.71 | 29.52 | 39.06 | 98.21 | 33.90 | 15.33 |

Table 5: **Ablation study** on Controllable Inpainting components using ScenePair.

|  | Rendering Accuracy | | Style Fidelity | | | |
|---|---|---|---|---|---|---|
|  | ACC (%) ↑ | NED ↑ | MSE ↓ | MS-SSIM ↑ | PSNR ↑ | FID ↓ |
| TextDiff-2 | 76.41 | 0.944 | 6.49 | 35.56 | 13.62 | 29.86 |
| + $\mathcal{L}_C$ | 88.52 | 0.970 | 8.75 | 29.90 | 12.01 | 38.85 |
| + G + $\mathcal{L}_S$ | 78.28 | 0.949 | 4.78 | 40.19 | 14.86 | 31.64 |
| + $\mathcal{L}_C$ + G + $\mathcal{L}_S$ | 78.44 | 0.951 | 4.79 | 40.11 | 14.85 | 31.69 |

We analyze the contribution of OmniText's components, text removal and controllable inpainting, using the SCUT-EnsText (Liu et al., 2020) and ScenePair (Zeng et al., 2024) benchmarks. Additionally, we include further analysis of the potential interactions between the text removal and controllable inpainting components in Appendix Section L.

**Text Removal.** As shown in Table 4, in the 'all text' setting, Self-Attention Inversion (SAI) alone suppresses text hallucinations and achieves strong removal performance. Adding Cross-Attention Reassignment (CAR) can slightly reduce performance, as it may introduce minor color shifts that lower PSNR and FID (see Appendix Fig. 19). However, masking all text for removal is often impractical. In the more realistic 'largest text' setting, SAI alone is insufficient, as seen in both quantitative (Table 4) and qualitative (Fig. 7) results. Combining SAI with CAR enables effective selective removal, improving both metrics and visual quality.

**Controllable Inpainting.** As shown in Table 5 and Fig. 7, the backbone alone struggles with accurate text rendering and style consistency. Adding latent optimization with the cross-attention content loss $\mathcal{L}_C$ improves accuracy, but often distorts style due to its sole focus on accuracy. Meanwhile, applying the grid trick ($G$) with the self-attention style loss $\mathcal{L}_S$ enhances style fidelity but reduces accuracy. Combining

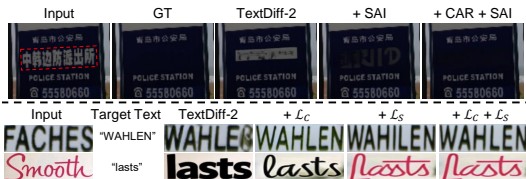

Figure 7: **Ablation study** for various components of our text removal and controllable inpainting.

both losses balances content and style, improving rendering accuracy while preserving style consistency. Since our goal is style-consistent text rendering, over-weighting $\mathcal{L}_C$ can cause undesirable style distortion in real-world applications.

## 5 CONCLUSION

We propose OmniText, a training-free method that manipulates self- and cross-attention for effective text removal and fine-grained control over text style and content during inpainting. Thanks to its modular design, OmniText is the first generalist method to tackle diverse Text Image Manipulation (TIM) tasks, including text insertion, editing, removal, repositioning, rescaling, and style-based insertion and editing. To evaluate performance across these tasks, we introduce OmniText-Bench, a mockup-based dataset with 150 image sets and corresponding ground truth. Experiments show that OmniText outperforms general text synthesis methods and achieves results comparable to specialist models trained for individual tasks. Moreover, our grid trick and self-attention manipulation for style control generalize well to other text synthesis backbones.

**Limitations.** The limitations of our work are discussed in the Appendix Section O.

ACKNOWLEDGMENTS

This work was supported by Institute of Information & communications Technology Planning & Evaluation (IITP) grant funded by the Korean Government [Ministry of Science and ICT (Information and Communications Technology)] (Project Number: RS-2022-00144444, Project Title: Deep Learning Based Visual Representational Learning and Rendering of Static and Dynamic Scenes, 100%).

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

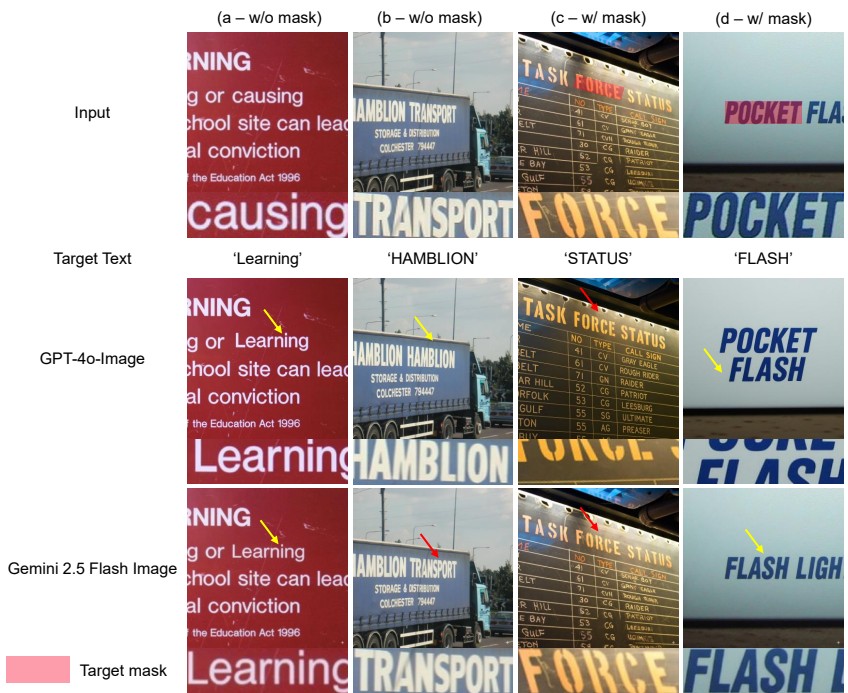

Figure 8: **Problems with existing closed-source models.** Closed-source models often fail to edit text correctly in tilted cases, merely reconstructing the input image (red arrow). In other cases, they generate text that extends beyond the intended region when no mask is provided (yellow arrows a, b), and even spill outside the boundaries when a mask is provided (yellow arrow d). Moreover, GPT-4o-Image often alters the entire image, such as enhancing the text (c) or causing color shifts (a).

# APPENDIX

## A    ETHICS STATEMENT

Our proposed method, OmniText, enables high-quality text manipulation within images, which is beneficial for a variety of image editing tools. In particular, OmniText demonstrates strong performance in text style fidelity, effectively replicating text styles from a reference image without requiring explicit font information. This capability is especially useful in applications such as poster design (Gao et al., 2023), clothing, and packaging design, where designers often rely on visual references without access to the original font specifications.

However, the strength of OmniText in accurately replicating text styles also introduces potential risks of misuse. First, it could be exploited for design plagiarism, where users replicate and modify copyrighted visual designs by merely changing the textual content. This may lead to violations of intellectual property rights. Second, OmniText could be used to alter textual content in images that convey information (e.g., infographics, advertisements, or screenshots), enabling the spread of misinformation through deceptively edited visuals. To address these concerns, we emphasize the importance of establishing appropriate legal and ethical frameworks to ensure responsible use of technologies like OmniText.

**Usage of Large Language Model (LLM).** We employed an LLM during the preparation of this manuscript to check grammar and spelling, and to provide recommendations for improving clarity.

Table 6: **Quantitative comparison of closed-source models on 20 images from the standard text editing benchmark (ScenePair).** We select 20 images to cover diverse cases (tilted vs. non-tilted and various text styles) and run each closed-source model using its respective service. **Bold** and underline denote the best and second-best performance in each category. **Red** denotes the best performance across all categories.

| | Rendering Accuracy | | Style Fidelity | | | |
|---|---|---|---|---|---|---|
| | ACC (%) ↑ | NED ↑ | MSE ↓ ($\times 10^{-2}$) | MS-SSIM ↑ ($\times 10^{-2}$) | PSNR ↑ | FID ↓ |
| GPT-4o-Image | 25.00 | 0.485 | 10.29 | 13.35 | 11.15 | 177.73 |
| Gemini 2.5 Flash Image | 40.00 | 0.640 | 6.77 | 16.88 | 13.08 | 140.92 |
| OmniText (Ours) | **95.00** | **0.994** | **4.63** | **40.15** | **14.79** | **119.64** |

Figure 9: **Qualitative comparison of closed-source models on the standard text editing benchmark (ScenePair).** Closed-source models often fail to edit text properly (red box) or fail to constrain the edits within the mask when no mask is provided (yellow boxes a, b, c), and they may still fail even when a mask is provided (yellow box d).

# B COMPARISON OF CLOSED-SOURCE AND ADDITIONAL OPEN-SOURCE MODELS

## B.1 COMPARISON AGAINST CLOSED-SOURCE MODELS

Recently, several closed-source models from industry, such as GPT-4o and Gemini 2.5 Flash Image, have demonstrated promising performance in image editing. However, there is currently no benchmark that evaluates the text editing capabilities of these models. To address this gap, we conducted a benchmark study by running inference experiments. We designed two different approaches:

- Without a mask: using the prompt 'Replace the text <SOURCE> with <TARGET> while preserving the original text style', where <SOURCE> is the source text and <TARGET> is the target replacement text.
- With a mask: using the prompt 'Replace the text of the image inside the given mask with <TARGET> while preserving the original text style'.

We also experimented with other prompts, but the two above consistently produced the best results. To benchmark these closed-source models, we selected 20 images from ScenePair (Zeng et al., 2024), covering diverse cases such as tilted versus non-tilted text and various text styles. Masks were provided for 10 images, as the models sometimes performed better without masks than with them. Since inference was conducted through the vendors' front-facing applications and we need to verify each image is edited properly until several trials, we could not evaluate the full dataset due to the impracticality of uploading all images manually. As shown in Fig. 8, two main issues emerged. First, the models often failed to edit tilted text (red arrows) while successfully handling simpler non-tilted cases. Second, the generated text frequently extended beyond the intended region when no mask was provided (yellow arrows a, b). Even with masks (yellow arrows d), the models did not reliably constrain text within the boundaries, suggesting poor utilization of positional guidance. Overall, these limitations indicate that precise text image manipulation is not feasible with current closed-source models. Notably, GPT-4o-Image tended to shift text substantially when masks were used and occasionally altered the entire image, introducing unwanted color shifts and enhancements (Fig. 8 a–d).

The impacts of failing to faithfully edit text and the positional shifts in generated text are reflected in Table 6, where these models achieve significantly lower rendering accuracy even in simple cases.

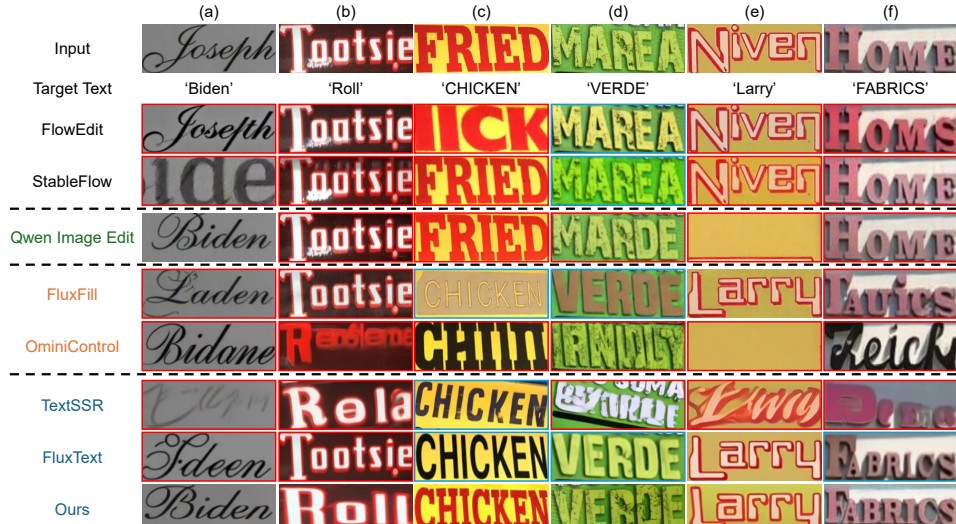

Figure 10: **Qualitative comparison of other open-source models on the standard text editing benchmark (ScenePair).** We group the models into four categories: (i) general image editing (black text), (ii) general image editing with native text support (green text), (iii) general inpainting models (orange text), and (iv) text inpainting models (blue text). Recent open-source models face two main issues: failure to edit text faithfully (red box) and failure to preserve the original text style (blue box).

Table 7: **Quantitative comparison on standard text editing benchmark (ScenePair) for various open-source models**. **Bold** and underline denote the best and second-best performance in each category. **Red** denotes the best performance across all categories.

| Category | Method | Rendering Accuracy | | Style Fidelity | | | |
|---|---|---|---|---|---|---|---|
| | | ACC (%) ↑ | NED ↑ | MSE ↓ ($\times 10^{-2}$) | MS-SSIM ↑ ($\times 10^{-2}$) | PSNR ↑ | FID ↓ |
| General Image Editing (GIE) | FlowEdit | **9.45** | **0.262** | 7.49 | 23.31 | 12.52 | **26.20** |
| | StableFlow | 7.50 | 0.231 | **7.31** | **23.39** | **12.75** | 36.14 |
| GIE with Text Support | QwenImageEdit | 27.42 | 0.459 | 6.91 | 25.51 | 13.46 | 25.69 |
| General Inpainting | FluxFill | **35.08** | **0.613** | **5.78** | **26.60** | **13.95** | **29.79** |
| | OminiControl | 12.97 | 0.313 | 9.30 | 19.35 | 11.91 | 61.06 |
| Text Inpainting | TextSSR | 53.20 | 0.773 | 6.73 | 26.47 | 13.20 | 48.22 |
| | FluxText | 54.53 | 0.815 | 5.60 | 33.88 | 14.22 | 38.43 |
| | OmniText (Ours) | **78.44** | **0.951** | **4.79** | **40.11** | **14.85** | **31.69** |

This is further illustrated in Fig. 9: without a mask, positional shifts often produce incorrect outputs (e.g., b – "AUF", c – "Fran"), causing otherwise faithful generations to be classified as errors. Even with masks, shifting still occurs (e.g., d – "FLASH I"), leading to misclassification. These results indicate that precise text image manipulation, even with positional masks, remains infeasible for current closed-source models.

## B.2 COMPARISON AGAINST OPEN-SOURCE MODELS

In addition to closed-source models, we compare OmniText with recent methods across several categories: general image editing (FlowEdit (Kulikov et al., 2024) and StableFlow (Avrahami et al., 2025)), general image editing with text support (QwenImageEdit (Wu et al., 2025)), general inpainting (FluxFill (Labs, 2024) and OminiControl (Tan et al., 2024)), and text inpainting (TextSSR (Ye et al., 2024) and FluxText (Lan et al., 2025)), using the ScenePair dataset (Zeng et al., 2024). Although some models, such as FlowEdit, StableFlow, FluxFill, and OminiControl, are designed for general editing, their results demonstrate partial capability in text editing. For evaluation, we follow the prompt format specified in each method's examples. For instance, QwenImageEdit allows the use of an additional bounding box (derived from dataset-provided masks) to guide text editing. In contrast, for general image editing methods, we provide only the source and target text as the prompt. We also experimented with prompt fine-tuning for each model. However, this did not yield noticeable improvements.

As shown in Table 7 and Fig. 10, general image editing methods such as FlowEdit and StableFlow struggle with text editing, achieving rendering accuracy below 10%. With native text support, QwenImageEdit performs better than general image editing methods but still fails to edit text faithfully, with rendering accuracy remaining below 30%. These findings are further illustrated in Fig. 10, where general image editing methods often produce no noticeable changes compared to the input image. In contrast, QwenImageEdit successfully edits the text in Fig. 10 (a), but fails on other examples.

In addition to general image editing models, we also compare our method with recent general inpainting models that show promising performance in text editing. However, as shown in Table 7 and Fig. 10, these models also fail to faithfully edit the text with rendering accuracy below 40% for FluxFill and below 15% for OminiControl. In some cases, it sometimes edits the text into content that differs from the target text Fig. 10 (a, f). It also removes the text or even changes the original text style Fig. 10 (c, d).

Lastly, we compare OmniText with recent text inpainting models: TextSSR (Ye et al., 2024), which uses a U-Net backbone, and FluxText (Lan et al., 2025), which adopts a Diffusion Transformer (DiT) backbone. As shown in Table 7, these models achieve low rendering accuracy (below 60%) and are therefore excluded from the main comparison. FluxText, however, shows a promising result by achieving high style fidelity (measured by MSE, MS-SSIM, and PSNR) despite its low accuracy. This observation is supported by Fig. 10, where FluxText generally preserves style consistency and fails only in one case (c), but struggles to edit text with intricate styles (a, b). TextSSR performs worse, failing to handle intricate styles and frequently introducing undesired tilts in the text (a, b, c, d, f), even when the original text is not tilted.

In summary, OmniText outperforms both methods by a large margin in terms of rendering accuracy and style fidelity. As shown in Fig. 10, OmniText is able to edit text with intricate styles (a, b, d) while preserving the original style across all cases (a–f).

## B.3 Discussion on the Connection and Divergence between General Image Editing and Specialized Text Image Manipulation

One related work, DesignEdit (Jia et al., 2024), attempts to combine general image editing with some text image manipulation tasks such as insertion and removal. Specifically, it supports a range of object-oriented edits including removal, resizing, swapping, duplication, and repositioning, as well as text-oriented applications such as text insertion and removal. DesignEdit is built on the SDXL backbone and performs text removal and editing by segmenting text regions with a pretrained multi-modal model. For insertion, however, it requires a glyph-level (exact) text mask from another image. This setup is incompatible with our problem setting, where only a target text and target mask are provided. Generating a glyph-level mask from text alone remains an open challenge in the field of text generation in images. For this reason, general image editing methods such as DesignEdit cannot be directly applied to our specialized text image manipulation setting.

As shown in Table 7, general image editing methods also perform poorly on text editing, achieving significantly lower rendering accuracy. This highlights the limitations of general T2I models for specialized text manipulation and motivates our adoption of text-oriented T2I models, which are better suited for this task.

Finally, faithfully editing both text (high rendering accuracy) and objects remains a challenging direction for future work, as text-generation backbones often struggle with object generation due to their distinct architectures (e.g., character-level text encoders) and training strategies tailored specifically for text rendering.

## C Additional Details on Proposed Method

### C.1 Analysis of Key Attention Properties

Our analysis is primarily based on empirical evaluation and attention visualization to uncover control mechanisms in generative models, similar to (Chefer et al., 2023; Chung et al., 2024; Phung et al., 2024). Beyond the performance gains in text removal achieved by combining CAR and SAI, and

---

**Algorithm 1** PyTorch-style pseudocode for Text Removal component of OmniText.

---

```
# attention_probs: precomputed attention probabilities,
#                   i.e., softmax(QK^T / sqrt(d)), for this attention layer (B x HW x C)
# is_self_attn: boolean indicating whether this attention layer is self-attention
# text_mask: resized target mask indicating the text region to be removed (HW)

if is_self_attn: # Eq. (3)
    masked_tensor = attention_probs[:, text_mask == 1] # B x seq_len x C
    max_value = masked_tensor.max(-1)[0].unsqueeze(-1) # B x seq_len x 1
    min_value = masked_tensor.min(-1)[0].unsqueeze(-1) # B x seq_len x 1
    flipped = max_value + min_value - masked_tensor
    flipped = flipped.softmax(dim=-1)
    attention_probs[:, text_mask == 1] = flipped
else: # Eq. (4)
    # inside text mask, start-of-description token
    attention_probs[:, text_mask == 1, [0]] = 0
    # inside text mask, end-of-description token
    attention_probs[:, text_mask == 1, [1]] = 1

    # outside text mask, start-of-description token
    attention_probs[:, text_mask == 0, [0]] = 1
    # outside text mask, end-of-description token
    attention_probs[:, text_mask == 0, [1]] = 0

    attention_probs[:, :, 2:] = 0 # zero out other text tokens
```

---

the improvements in text rendering accuracy and style fidelity from our latent optimization with the proposed cross-attention content loss and self-attention style loss, we also conduct an approximate statistical analysis.

Precise statistical evaluation is limited by the lack of ground truth data (e.g., exact text and generated character positions, or exact positions of style-matched text). To approximate this, we analyze cross- and self-attention in Decoder Block 2, Layer 0, the same layer used in our visualization, to better understand their behavior in controllable inpainting.

First, to understand the properties of cross-attention for controllable inpainting, we use ScenePair (text editing). We approximate character-level masks by evenly dividing the text mask. We compute attention values for each character token inside its corresponding spatial region (character mask) and outside the text mask, where each token's attention values are normalized so that the sum is one. We average attention responses across characters, the first 50% of sampling steps, and all images (excluding those with inaccurate text rendering). The average attention value inside the character mask is 0.5645, while it is 0.0772 outside the text mask. This strong spatial alignment (high attention value inside the character mask) supports our claim.

Second, to understand the properties of self-attention for controllable inpainting, we cannot perform additional analysis since the analysis will require information on the spatial position of characters or text with similar styles. One possible approach is to use an OCR model to detect characters. However, OCR can only identify identical characters and cannot evaluate whether two characters share a similar style. Since such data is not currently available, we leave this analysis for future work, due to the lack of a model capable of evaluating style similarities between characters.

### C.2 TEXT REMOVAL PSEUDOCODE

To enhance the reproducibility of our text removal component, we provide PyTorch-style pseudocode in Algorithm 1.

## D DISCUSSION ON GENERALIZATION OF THE METHOD

### D.1 DISCUSSION ON GENERALIZATION TO A DIFFUSION TRANSFORMER-BASED BACKBONE

Our analysis results for U-Net-based backbones differ from those for DiT-based models. This discrepancy arises from a fundamental difference in tokenization: DiT uses token-level tokenization, encoding the target text as one or a few tokens based on a tokenizer dictionary, whereas U-Net-based models use character-level tokenization, encoding each character individually. Because of these

differences, our method cannot be directly applied to DiT-based models. These distinctions also affect our text removal analysis, rendering it inapplicable to DiT-based models.

Regarding controllable inpainting, the attention behavior in FluxFill (a DiT-based backbone) (Labs, 2024) exhibits key differences. In FluxFill, self-attention within the target text mask tends to focus on other text regions equally, rather than on characters or text with similar styles as seen in our U-Net-based models. Similarly, the cross-attention of text tokens in FluxFill attends to the entire text region as a whole, rather than focusing on the spatial positions of individual characters. These differences prevent us from applying our method to DiT-based models.

Table 8: **Generalization of our method on additional U-Net-based backbones for different tasks in OmniText-Bench. Bold** denotes the best performance.

| **Text Insertion** | | | | | | |
|---|---|---|---|---|---|---|
| | Rendering Accuracy | | Style Fidelity | | | |
| | ACC (%) ↑ | NED ↑ | MSE ↓ ($\times 10^{-2}$) | MS-SSIM ↑ ($\times 10^{-2}$) | PSNR ↑ | FID ↓ |
| DreamText (Wang et al., 2024) | **85.33** | **0.965** | 6.44 | 36.23 | 13.18 | **54.48** |
| (Style-control) DreamText + Mod. (Ours) | 84.00 | 0.946 | **5.65** | **38.05** | **13.96** | 54.87 |
| UDiffText (Zhao & Lian, 2023) | **94.67** | **0.985** | 7.94 | 34.83 | 12.03 | 62.00 |
| (Style-control) UDiffText + Mod. (Ours) | 89.33 | 0.975 | **6.74** | **36.68** | **13.08** | **58.03** |
| **Style-based Text Insertion** | | | | | | |
| | Rendering Accuracy | | Style Fidelity | | | |
| | ACC (%) ↑ | NED ↑ | MSE ↓ ($\times 10^{-2}$) | MS-SSIM ↑ ($\times 10^{-2}$) | PSNR ↑ | FID ↓ |
| DreamText (Wang et al., 2024) | **78.67** | **0.941** | 10.06 | 24.11 | 10.67 | **79.71** |
| (Style-control) DreamText + Mod. (Ours) | 74.67 | 0.929 | **7.75** | **28.66** | **12.19** | 94.00 |
| UDiffText (Zhao & Lian, 2023) | **88.00** | **0.973** | 11.79 | 24.43 | 9.98 | 82.21 |
| (Style-control) UDiffText + Mod. (Ours) | 86.67 | 0.968 | **9.98** | **26.64** | **10.80** | **81.21** |
| **Text Editing** | | | | | | |
| | Rendering Accuracy | | Style Fidelity | | | |
| | ACC (%) ↑ | NED ↑ | MSE ↓ ($\times 10^{-2}$) | MS-SSIM ↑ ($\times 10^{-2}$) | PSNR ↑ | FID ↓ |
| DreamText (Wang et al., 2024) | **84.00** | **0.954** | 7.99 | 34.17 | 12.23 | 65.83 |
| (Style-control) DreamText + Mod. (Ours) | 80.67 | 0.946 | **5.28** | **40.11** | **14.26** | **50.27** |
| UDiffText (Zhao & Lian, 2023) | **96.00** | **0.984** | 9.78 | 31.22 | 11.17 | 68.67 |
| (Style-control) UDiffText + Mod. (Ours) | 90.67 | 0.981 | **6.33** | **38.82** | **13.34** | **55.23** |
| **Style-based Text Editing** | | | | | | |
| | Rendering Accuracy | | Style Fidelity | | | |
| | ACC (%) ↑ | NED ↑ | MSE ↓ ($\times 10^{-2}$) | MS-SSIM ↑ ($\times 10^{-2}$) | PSNR ↑ | FID ↓ |
| DreamText (Wang et al., 2024) | **79.33** | **0.950** | 9.98 | 24.78 | 10.63 | **86.09** |
| (Style-control) DreamText + Mod. (Ours) | 72.67 | 0.920 | **7.14** | **30.25** | **12.65** | 106.23 |
| UDiffText (Zhao & Lian, 2023) | **94.67** | **0.986** | 11.97 | 24.96 | 9.86 | 87.70 |
| (Style-control) UDiffText + Mod. (Ours) | 81.33 | 0.964 | **10.22** | **26.32** | **10.58** | **84.78** |
| **Text Rescaling** | | | | | | |
| | Rendering Accuracy | | Style Fidelity | | | |
| | ACC (%) ↑ | NED ↑ | MSE ↓ ($\times 10^{-2}$) | MS-SSIM ↑ ($\times 10^{-2}$) | PSNR ↑ | FID ↓ |
| DreamText (Wang et al., 2024) | **72.00** | **0.931** | 9.22 | 26.82 | 11.32 | 72.84 |
| (Style-control) DreamText + Mod. (Ours) | 45.33 | 0.887 | **6.45** | **29.04** | **13.02** | **65.55** |
| UDiffText (Zhao & Lian, 2023) | **88.67** | **0.978** | 11.11 | **28.56** | 10.28 | 76.51 |
| (Style-control) UDiffText + Mod. (Ours) | 56.00 | 0.918 | **7.98** | 28.30 | **11.94** | **62.69** |
| **Text Repositioning** | | | | | | |
| | Rendering Accuracy | | Style Fidelity | | | |
| | ACC (%) ↑ | NED ↑ | MSE ↓ ($\times 10^{-2}$) | MS-SSIM ↑ ($\times 10^{-2}$) | PSNR ↑ | FID ↓ |
| DreamText (Wang et al., 2024) | **83.33** | **0.958** | 7.87 | 32.44 | 12.03 | 69.15 |
| (Style-control) DreamText + Mod. (Ours) | 82.67 | 0.956 | **5.22** | **39.67** | **14.15** | **54.61** |
| UDiffText (Zhao & Lian, 2023) | **92.00** | **0.985** | 10.35 | 30.15 | 10.84 | 74.13 |
| (Style-control) UDiffText + Mod. (Ours) | 88.67 | 0.978 | **6.49** | **38.89** | **13.11** | **58.51** |

### D.2 DISCUSSION ON OTHER U-NET-BASED BACKBONES

The framework consists of two components: text removal and controllable inpainting. The text removal component can, in principle, be applied to any backbone that is trained on some removal cases, such as the TextDiff-2 training data (MARIO-10M (Chen et al., 2023b)). However, to the best of our knowledge, no recent backbone has been trained on this dataset, and thus we are currently unable to evaluate our text removal component on other models.

In contrast, the controllable inpainting component can be applied to any U-Net-based backbone. Recent methods (Ye et al., 2024) still predominantly adopt U-Net-based architectures, as they achieve better accuracy than DiT-based backbones, so our method remains directly beneficial to current approaches. Our content-control mechanism, however, cannot be applied into recent models (Tuo et al., 2023; 2024; Zhao & Lian, 2023; Wang et al., 2024; Ye et al., 2024), because these methods rely on glyph-based conditioning that fundamentally changes the behavior of the cross-attention layers and in some cases even remove the text encoder e.g., TextSSR (Ye et al., 2024). By contrast, our style-control mechanisms, implemented either via a style loss or via self-attention modulation, depending on whether the backbone has a specialized design, are compatible with recent U-Net-based backbones such as UDiffText (Zhao & Lian, 2023), DreamText (Wang et al., 2024), and even the most recent TextSSR (Ye et al., 2024).

As shown in Table 8, we apply our self-attention modulation to UDiffText (Zhao & Lian, 2023) and DreamText (Wang et al., 2024) (termed as Backbone + Mod.). We have already demonstrated the generalization ability of our method to UDiffText in Table 12 and Table 13. Overall, our method improves style fidelity across all tasks and almost all metrics. The only exception occurs in a few tasks where the FID of DreamText becomes worse after we apply our modulation. We attribute this to a trade-off: in some cases, while our method improves other metrics related to style and content consistency, the resulting images may deviate slightly from the natural-image distribution captured by FID. As discussed in the main paper, FID is not an appropriate metric for evaluating style fidelity in text-centric images, because it is trained on natural (non text-centric) images and primarily measures overall naturalness rather than style consistency.

In addition, we apply our style-loss-based variant to the TextSSR backbone (Ye et al., 2024), enabling style control in *Chinese* text generation (see Section N). We do not report the TextSSR results on OmniText-Bench, because this backbone exhibits relatively low accuracy, as shown in Table 7.

The key consideration when applying our method to other backbones is that each backbone's self-attention layers behave differently. For example, to apply our method to UDiffText, we perform our modulation at Block 1, Layer 0 and Block 2, Layer 1 during the middle phase of sampling, specifically from 40% to 80% of the total steps. In contrast, for DreamText, we apply our method at Decoder Block 2, Layer 2 over the same sampling range as in UDiffText. Therefore, for a new backbone, one must first identify the layer that most strongly encodes the style prior and then determine the sampling interval over which to apply our method. Note that this adjustment also happens in other training-free works Phung et al. (2024); Chefer et al. (2023); Si et al. (2024), where these methods usually apply their methods in some variants of latent diffusion model with no specialized architecture design. Then, when applied to different backbones, some adjustments, such as empirically determined hyperparameters, are needed for the method to be effective.

## E DETAILS ABOUT OUR PROPOSED OMNITEXT-BENCH

Currently, text image manipulation datasets primarily cover three tasks: text generation (Chen et al., 2023b), editing (Zeng et al., 2024), and removal (Liu et al., 2020). However, no existing work has explored other important applications such as text rescaling, repositioning, style-based insertion, and style-based editing. To address this gap, we propose OmniText-Bench, a mockup-based evaluation dataset designed to support five key text image manipulation (TIM) applications: text rescaling, repositioning, removal, and both insertion and editing with an additional style reference image. Specifically, we collect 150 mockups from two free mockup platforms: Freebiesbug[1] and Mockupfree[2]. These platforms allow redistribution of derivative works under the condition that the

---

[1]Freebiesbug Link
[2]Mockupfree Link

original mockup files are not redistributed. We adhere to these terms by only distributing our edited results (images), not the original assets. Moreover, we provide proper attribution and include the license terms and credits for each mockup. Since Freebiesbug acts as an aggregator, we also credit the original designers of the mockups used.

The 150 mockups are categorized as follows: 50 print-related (e.g., posters, books, magazines, banners), 40 apparel-related (e.g., clothing, hats, bags), 40 packaging-related (e.g., boxes, bottles, mugs, cans, perfume containers), and 20 device-related (e.g., billboards, phones, laptops). For each mockup, we zoom into the region containing the target text until the shortest side of the text area reaches at least 64 pixels. We then randomly place the target text within a $512 \times 512$ patch and crop the image accordingly. This patch is used as the base for manipulation. The target text is then modified according to each TIM application:

- Text Removal: The text is masked and removed, followed by background inpainting, similar to SCUT-EnsText (Liu et al., 2020).
- Text Rescaling and Repositioning: The original text is removed and reinserted at a new scale or position. Both the removal and new insertion regions are annotated with corresponding masks.
- Text Editing: The content of the original text is modified while preserving its position. We annotate both the removal and new insertion masks since the new text can be bigger or smaller than the original text.
- Text Insertion: New text is added to a previously empty region, and the corresponding target mask is generated.
- Style-based Insertion and Editing: The inserted or edited text adopts a font style and color sampled from a separate reference image.

For each application, we provide a complete set of annotated images, including the input image, target mask, reference image (if applicable), reference mask, and the ground truth output. This structure allows systematic evaluation of a broad range of TIM tasks.

## F  ADDITIONAL IMPLEMENTATION DETAILS

**Component Configurations for Text Removal (TR) and Controllable Inpainting (CI).** Our OmniText framework consists of two main components: Text Removal ($TR_{\epsilon_\theta}$) and Controllable Inpainting ($CI_{\epsilon_\theta}$). These components serve as the foundation for a wide range of text image manipulation applications. The Controllable Inpainting module ($CI_{\epsilon_\theta}$) leverages an additional reference image $I_{\text{ref}}$ through our proposed Grid Trick ($G$), which is adapted according to the specific application. We summarize the configurations used for each application as follows:

- **Text Removal:** We use only the text removal module, $TR_{\epsilon_\theta}$.
- **Text Rescaling, Editing, and Repositioning:** We apply $TR_{\epsilon_\theta}$ followed by $CI_{\epsilon_\theta}$, using the input image and input text mask as the reference image and mask, i.e., $I_{\text{ref}} = I$ and $m_{\text{ref}} = m$.
- **Text Insertion:** We use only $CI_{\epsilon_\theta}$, with the input image and the provided text mask as the reference: $I_{\text{ref}} = I$, $m_{\text{ref}}$.
- **Style-Controlled Insertion:** We use only $CI_{\epsilon_\theta}$, using the given reference image and text mask: $I_{\text{ref}}$ and $m_{\text{ref}}$.
- **Style-Controlled Editing:** We first apply $TR_{\epsilon_\theta}$, followed by $CI_{\epsilon_\theta}$ using the given reference image and text mask: $I_{\text{ref}}$ and $m_{\text{ref}}$.

**Lambda Weights for CI Across Datasets.** In the latent optimization process of our Controllable Inpainting (CI) module, we minimize a loss function composed of two terms: the Cross-Attention Content Loss ($\mathcal{L}_C$) and the Self-Attention Style Loss ($\mathcal{L}_S$), weighted by hyperparameters $\lambda_C$ and $\lambda_S$, respectively:

$$\mathcal{L} = \lambda_C \mathcal{L}_C + \lambda_S \mathcal{L}_S.$$

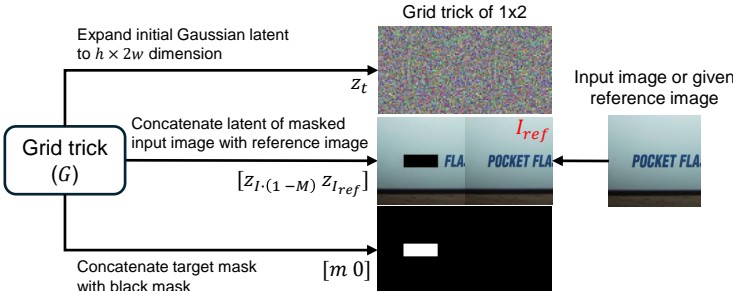

Figure 11: **Detailed operations of the Grid Trick** $G$. The Grid Trick constructs a grid structure for the input by performing: (i) expansion of the initial Gaussian latent, (ii) concatenation of the latents from the masked input image and the reference image, and (iii) concatenation of the target mask with a black mask.

We employ the same configurations for all of the benchmarks (ScenePair and OmniText-Bench). We set $\lambda_C = 5$ and $\lambda_S = 10$ for all experiments unless stated otherwise.

**Details of TR.** Our Text Removal (TR) module is composed of two key contributions: Self-Attention Inversion (SAI) and Cross-Attention Reassignment (CAR). We perform 20 sampling steps in total, where SAI is applied during the first 50% of sampling steps, while CAR is applied throughout all 20 steps. SAI is only applied in the early stages because its primary purpose is to suppress the self-attention activation around existing text, thereby mitigating text hallucination. Extending SAI beyond the early steps may lead to artifacts: once the model has shifted focus to the background, further inversion can introduce artifacts. On the other hand, CAR is performed across the full sampling process to maintain consistent suppression of hallucinated text. Applying CAR only partially would weaken this effect, potentially allowing artifacts to emerge. It is important to note that the application of these modules can be adjusted on a case-by-case basis due to the modularity of our framework. For instance, if the backbone model reliably detects the surrounding text when performing inpainting, SAI alone may suffice. Conversely, when the backbone fails to localize the surrounding text properly, both SAI and CAR are necessary. In terms of architecture, we apply SAI in the decoder of the U-Net, specifically at Block 2, Layer 1 of the self-attention layers. CAR is applied to the cross-attention layers in the decoder at Block 1, Layer 2 and Block 2, Layer 0.

**Details of CI.** Our Controllable Inpainting (CI) module incorporates two loss components: the Cross-Attention Content Loss ($\mathcal{L}_C$) and the Self-Attention Style Loss ($\mathcal{L}_S$). For $\mathcal{L}_C$, we extract cross-attention maps from the decoder at Block 1, Layer 2 and Block 2, Layer 0. For $\mathcal{L}_S$, we extract self-attention maps from Block 1, Layer 0 and Block 1, Layer 1 of the decoder. During latent optimization, we perform iterative updates at three stages of the denoising process: the initial step (0%), the 20% step, and the 40% step. Each optimization stage runs for 20 iterations.

When optimizing with the Cross-Attention Content Loss ($\mathcal{L}_C$) alone, we observe that content control is often unsatisfactory. Further analysis reveals that the gradient signal from $\mathcal{L}_C$ does not propagate effectively, i.e., increasing the loss value does not meaningfully impact content alignment in the generated output. To address this limitation, we introduce a technique called Self-Attention Manipulation (SAM). Specifically, we enforce the self-attention within each character region to behave as an identity matrix, that is:

$$S^l_{i \in m_{c_k}, j \in m_{c_k}} = I$$

where $m_{c_k}$ denotes the spatial mask corresponding to the $k$-th character. This modification strengthens the gradient flow within each character region and improves alignment between the generated content and the desired target. However, applying SAM may lead to a reduction in style fidelity, as the imposed identity structure disrupts the natural style representation encoded in self-attention. Despite this trade-off, we adopt SAM in all results (including main results) in the main paper to ensure reliable content control. Specifically, we apply SAM to the self-attention layer of the decoder in Block 1, Layer 2.

**Details of the Grid Trick.** Our Grid Trick is inspired by the recent work in video editing (Kara et al., 2024), where it is used to enforce temporal consistency across frames. Similarly, this technique, often referred to as a "character sheet" in image generation, is employed to maintain consistent visual

style across multiple outputs (Kara et al., 2024). In our work, we adopt the Grid Trick to enhance style control and to support the computation of the Self-Attention Style Loss ($\mathcal{L}_S$). However, since our method is based on an inpainting backbone, we adapt the Grid Trick to suit this architecture. Specifically, we apply the Grid Trick to the input of the inpainting model, which consists of the initial latent, the latent mask, and the masked latent representation of the input image. An illustration of this process is provided in Fig. 11. Because the model input forms a grid-structured representation, the output is also a grid. To obtain the final result, we crop the target region from the grid-based output.

**Details of Initial Latent.** For all components in our framework, we compute the initial latent using the DDPM forward process (Ho et al., 2020):

$$z_t = \sqrt{\bar{\alpha}_t}\, z_0 + \sqrt{1 - \bar{\alpha}_t}\, \epsilon, \quad \epsilon \sim \mathcal{N}(0, I)$$

where $z_0$ is the latent representation of the input image obtained from the VAE encoder. This noisy latent $z_t$ serves as the starting point for our TR and CI.

## G  BEST PRACTICE FOR HYPERPARAMETER SELECTION

In this section, we recommend best practices for hyperparameter selection. For text removal, we recommend using SAI on 50% of the sampling steps and CAR on 100% of the sampling steps. This configuration is the most robust for both "largest text" removal and "all text" removal, as shown in Table 4.

For controllable inpainting, the choice of hyperparameters depends on whether the user prioritizes style consistency or is willing to accept style hallucination. If style consistency is not critical, one can simply optimize with the content loss while disabling both the grid trick and the self-attention style loss. In this case, the recommended loss weight for content loss $\lambda_C$ is 1. However, if style control is important, a more careful hyperparameter search is recommended.

The results in Table 16 and Table 18 show that our method is robust to both the choice of loss weights and the number of iterations. This robustness allows us to use the same loss weights for both OmniText-Bench and ScenePair (Zeng et al., 2024). Note that, we do not perform hyperparameter search for OmniText-Bench. If a dedicated hyperparameter search is needed, one can first use a small number of iterations (e.g., 5 iterations) to explore candidate settings efficiently and later scale to improve the style fidelity (see Table 18). We recommend first tuning the style loss weight $\lambda_S$, since it has a larger impact on performance and needs to be sufficiently strong for $\mathcal{L}_S$ to handle the styles present in the dataset. The recommended starting point for $\lambda_S$ is 5, which is sufficiently large without being excessive. After that, one can tune the content loss weight $\lambda_C$, wheret the recommended starting point is 1. However, as shown in Table 16, the results are quite robust to $\lambda_C$, so using a default value of 1 is often adequate, with further refinement only if necessary.

Note that our method is particularly well suited for interactive scenarios (changing the weight depending on the sample instead of dataset) because style and rendering accuracy can already be assessed during the first few optimization steps, allowing users to adjust the loss weights based on the observed results after, for example, the first five sampling steps.

## H  ADDITIONAL EXPERIMENTAL SETTING DETAILS

**Computing Resources.** As our method is entirely training-free, it requires only a single GPU for inference. Specifically, we use an NVIDIA RTX 4090 (24GB) to run the Text Removal (TR) component. For the Controllable Inpainting (CI) module, which involves both the Grid Trick ($G$) and latent optimization, we utilize an NVIDIA A6000 (48GB) due to the increased GPU memory requirements.

**Preprocessing of Images in the Evaluation Dataset for Standard Benchmarks.** TextCtrl (Zeng et al., 2024) reports low accuracy for text inpainting baselines in their evaluation. However, this underperformance is largely due to the evaluation protocol used. Inpainting baselines based on latent diffusion models typically require a minimum mask size, as they are unable to reliably inpaint small regions, specifically, regions smaller than 64 pixels on the shortest side. TextCtrl conducts text editing by cropping the target text region and resizing it to a fixed resolution of 64×256 pixels. While this is suitable for their model, such preprocessing removes crucial contextual information required by

inpainting-based baselines. To ensure a fair comparison, we modify the preprocessing protocol. Instead of fixed resizing, we zoom into the target text region such that the shortest side of the text mask is at least 64 pixels, while preserving the surrounding context. This allows inpainting-based baselines to function properly and significantly improves their editing performance.

Specifically, for the text removal evaluation dataset (i.e., the test set of SCUT-EnsText (Liu et al., 2020)), we apply different preprocessing strategies based on the evaluation setting: For the "all text" removal setting, we zoom in as far as possible while ensuring that all text regions remain within the cropped image. For the "largest text" removal setting, we zoom in such that the shortest side of the target text mask is greater than 64 pixels, ensuring compatibility with inpainting-based methods. For the text editing benchmark using ScenePair (Zeng et al., 2024), we zoom in so that the shortest side of the target text mask exceeds 96 pixels. This provides sufficient context for editing and ensures that the inpainting-based models can operate effectively. By applying these preprocessing strategies, we enable a more fair and representative comparison across different methods, particularly those that rely on latent diffusion-based inpainting.

**Details of Baseline Methods.** In our main paper, we compare OmniText against two categories of baselines: generalist methods that perform text inpainting or synthesis, and specialist methods trained for specific tasks such as text removal or editing. The generalist baselines include state-of-the-art (SOTA) text inpainting and synthesis methods that accept an input mask to render text at a specified location. These methods include AnyText (Tuo et al., 2023), AnyText2 (Tuo et al., 2024), TextDiff-2 (Chen et al., 2024), UDiffText (Zhao & Lian, 2023), and DreamText (Wang et al., 2024). For the specialist baselines in text removal, we evaluate against the recent SOTA method ViTEraser (Peng et al., 2024), which is capable of removing all text from an image but does not support selective removal. In addition, we include the LaMa model (Suvorov et al., 2022), a general-purpose image inpainting method. For text editing, we compare with the state-of-the-art TextCtrl (Zeng et al., 2024), which is specifically designed for style-preserving text editing. Unfortunately, we are unable to compare against the newest method RS-STE (Fang et al., 2025), as its implementation was not publicly available at the time of submission. Currently, there are no existing baseline methods capable of handling the full range of applications included in our OmniText-Bench. To address this, we adopt LaMa (Suvorov et al., 2022) as the first step for all tasks that involve text removal, including rescaling, repositioning, and editing.

**Details of Metrics**. For text removal, we follow standard evaluation protocols established in prior work (Peng et al., 2024), and compute metrics over the entire image. The evaluation metrics include Mean Squared Error (MSE), Multi-Scale Structural Similarity (MS-SSIM), Peak Signal-to-Noise Ratio (PSNR), and Fréchet Inception Distance (FID), which measures the statistical distance between image distributions in the feature space of a pre-trained network. Due to space constraints in the main paper, we report only MS-SSIM, PSNR, and FID, omitting MSE. As PSNR is derived from MSE, reporting both is redundant in practice.

For text editing, we adopt the standard evaluation setup used in recent works (Zeng et al., 2024; Fang et al., 2025), where metrics are computed over the cropped text region to assess both style fidelity and rendering accuracy. Style fidelity is evaluated using MSE, PSNR, MS-SSIM, and FID. Rendering accuracy is assessed using Word Accuracy (ACC), defined as word-level accuracy, and Normalized Edit Distance (NED), where higher denotes better performance. To compute these accuracy-based metrics, we use a case-sensitive text recognition model from (Baek et al., 2019).

We exclude background preservation from the metrics since evaluation of text editing uses cropped images (without background). Meanwhile, for text removal, we replace the unedited regions of the output with the original input, thus background preservation is not an accurate metric for our settings.

# I   RUNTIME COMPARISON

We provide several runtime comparisons for a more thorough analysis. Specifically, we run each method/configuration fifteen times on a single RTX A6000 48GB GPU (which can handle all configurations), then average the runtimes and report the per-sample runtime in seconds. Note that this additional analysis is mainly intended as a reference for future work: in this paper, OmniText focuses primarily on enabling text generation models to tackle diverse TIM tasks and to perform effective removal and controllable inpainting, rather than on optimizing efficiency. A promising way

Table 9: **Runtime comparison per sample** for our Text Removal components.

| Method | Runtime (s) |
|---|---|
| TextDiff-2 ((Chen et al., 2024)) | 1.954 |
| + SAI | 2.083 |
| + SAI + CAR | 2.335 |

Table 10: **Runtime comparison per sample** for our Controllable Inpainting components.

| Method | Runtime (s) |
|---|---|
| TextDiff-2 ((Chen et al., 2024)) | 2.521 |
| + $\mathcal{L}_C$ | 24.946 |
| + G + $\mathcal{L}_S$ | 68.279 |
| + $\mathcal{L}_C$ + G + $\mathcal{L}_S$ | 69.461 |

Table 11: **Runtime comparison per sample** against other methods. Runtime for other generalist are the same for both text editing and removal, thus we do not show them in text removal.

| Text Editing | |
|---|---|
| Method | Runtime (s) |
| AnyText ((Tuo et al., 2023)) | 1.526 |
| AnyText2 ((Tuo et al., 2024)) | 3.106 |
| TextDiff-2 ((Chen et al., 2024)) | 2.521 |
| UDiffText ((Zhao & Lian, 2023)) | 8.257 |
| DreamText ((Wang et al., 2024)) | 5.987 |
| OmniText (Ours) | 69.461 |
| (Specialist) TextCtrl | 6.782 |
| **Text Removal** | |
| Method | Runtime (s) |
| TextDiff-2 ((Chen et al., 2024)) | 1.954 |
| OmniText (Ours) | 2.335 |
| (Specialist) LaMa ((Suvorov et al., 2022)) | 1.396 |
| (Specialist) ViTEraser ((Peng et al., 2024)) | 0.061 |

to improve efficiency is to distill our latent optimization procedure into a new attention modulation technique that can jointly handle content and style control during sampling without latent optimization, which we leave to future work.

**Runtime Comparison on Text Removal Components.** We first analyze the additional runtime introduced by each component of our text removal pipeline. As shown in Table 9, the overhead is negligible for both SAI (run for 50% of the sampling steps) and CAR (run for 100% of the sampling steps). Both components are applied as inference-time modulation and do not involve computationally heavy operations, which explains the minimal increase in runtime.

**Runtime Comparison on Controllable Inpainting Components.** We next analyze the runtime increase introduced by each component of our text editing pipeline. As shown in Table 10, the added runtime is substantial. First, latent optimization with our simple cross-attention content loss ($\mathcal{L}_C$) increases runtime by about 22 seconds, indicating that latent optimization is inherently expensive. Note that the expensive runtime of latent optimization has been known in many works (Chefer et al., 2023; Phung et al., 2024), but is still being used even in many editing tasks including more expensive video editing (Pondaven et al., 2025) due to its effectiveness. Second, when we apply the grid trick (G) and change the input to a grid-based representation with twice the original resolution, the runtime further increases. This increase is mainly due to the higher-resolution input, rather than the self-attention style loss ($\mathcal{L}_S$) itself, since the computation of $\mathcal{L}_S$ is similar to $\mathcal{L}_C$, where $\mathcal{L}_C$ adds only about 1 second of overhead. Despite the higher computational cost, this trade-off is necessary due to the difficulty of style control in text generation architectures. Our method is, to our knowledge, the first effective approach that enables style control in text generation/inpainting while supporting diverse TIM tasks.

**Runtime Comparison against Other Methods.** We also compare the runtime of our method with existing baselines including both generalists and specialists. As shown in Table 11, OmniText has the longest runtime for text editing. This is mainly due to the combination of the grid trick (G) and latent optimization. Latent optimization methods such as (Chefer et al., 2023; Phung et al., 2024) are known to be computationally expensive but offer stronger control over specific aspects. Since our domain is text, additional mechanisms, e.g., Grid trick, are required to enable fine-grained control, which further increases runtime. Compared to a specialist model that operates on cropped inputs ($64 \times 256$), such as TextCtrl (Zeng et al., 2024), other generalist models that run on full-resolution inputs ($512 \times 512$) tend to be faster. This is likely due to the additional inference-time techniques

employed by TextCtrl to improve style consistency. For text removal, our method is still the slowest among diffusion-based methods, but remains competitive overall, while performing effective text removal. Note that, for other generalists, the runtime is similar to their editing runtime, thus we omit them from the text removal table. Specialist models that use non-diffusion architectures, such as LaMa (Suvorov et al., 2022) and ViTEraser (Peng et al., 2024), are faster because they do not require 20 diffusion sampling steps, while diffusion model tends to be slower.

Importantly, for text editing, OmniText is more effective than both generalist and specialist methods, and for text removal, it outperforms other generalists as well as LaMa. Moreover, unlike specialist and other generalist approaches, our method can handle a wide range of TIM tasks with a single framework, while other generalists lag in terms of style fidelity performance.

**Runtime Comparison against Alternative Methods.** We compare our approach to alternative methods along two axes: content control and style control. For content control, we compare our classification-loss-based method with Attend-and-Excite (Chefer et al., 2023) and Attention Refocusing (Phung et al., 2024). As shown in Table 20, all methods have very similar runtimes. Attend-and-Excite is about 0.4 seconds faster than our method, while Attention Refocusing is about 0.1 seconds slower; our method lies in between. Despite comparable runtimes, our approach provides stronger and more reliable content control (higher rendering accuracy), with a favorable trade-off between performance and runtime.

For style control, we compare our self-attention style loss with an image style transfer method (StyleID (Chung et al., 2024)) and with our proposed self-attention modulation. As shown in Table 21, StyleID achieves the fastest runtime because it operates on non–grid-based inputs. Adding grid-based inputs with self-attention modulation increases runtime by approximately 15 seconds, and latent optimization with the self-attention style loss increases it further. However, StyleID yields suboptimal style transfer. With the grid trick and self-attention modulation, style fidelity improves noticeably, and combining the grid trick with the self-attention style loss achieves the best style fidelity, showing that our approach is the most effective among the compared alternatives.

**Runtime Comparison on the Number of Iterations for Latent Optimization.** Finally, we analyze the impact of the number of latent optimization iterations on performance and runtime in Table 18. As shown in the table, reducing the number of iterations keeps rendering accuracy stable, but gradually decreases style fidelity. At the same time, fewer iterations reduce runtime, where decreasing the iteration count by one reduces runtime by approximately 2 seconds. Since most performance changes (other than style fidelity) are minor, this observation provides a useful guideline for hyperparameter search: one can start with a smaller number of iterations for efficiency, and later increase the iteration count when stronger style control is required.

All in all, OmniText increases runtime only slightly for text removal but more substantially for text editing. Despite this, OmniText achieves the best overall performance among generalist methods and several specialist baselines. Because this work focuses primarily on effective removal and controllable inpainting for diverse TIM tasks, our efficiency analysis can serve as a foundation for future work on improving the efficiency of text image manipulation (TIM).

## J    ADDITIONAL EXPERIMENTS

### J.1    FULL COMPARISON ON STANDARD BENCHMARK

In this section, we show full comparison on standard benchmark consisting of text removal using SCUT-EnsText (Liu et al., 2020) and text editing using ScenePair (Zeng et al., 2024).

#### J.1.1    TEXT REMOVAL (SCUT-ENSTEXT BENCHMARK)

Latent Diffusion Models (LDMs) (Rombach et al., 2022) are inherently unable to perfectly reconstruct the input image due to the compression of the image into the latent space. As a result, LDM-based text inpainting methods often produce noticeable shifts in the unmasked regions of the image. To ensure a fair comparison across different methods, we use the evaluation protocol from TextCtrl (Zeng et al., 2024), where the unmasked regions in the output are replaced with the corresponding parts from the original input image. This evaluation protocol is illustrated in Fig. 12.

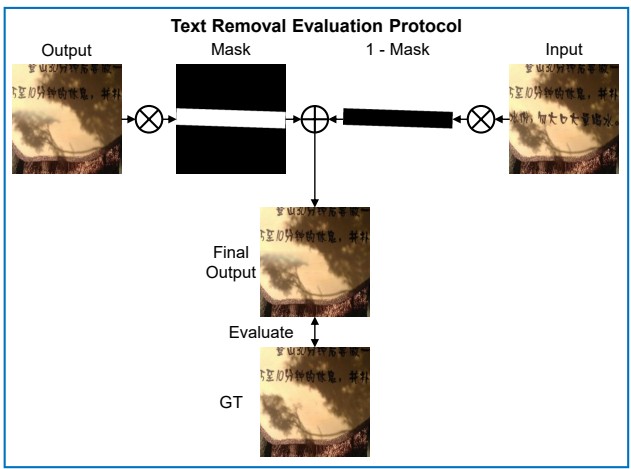

Figure 12: **Overview of evaluation protocol for text removal.** We follow the evaluation protocol of TextCtrl (Zeng et al., 2024), where the composite image is formed by combining the masked region from the text removal output with the unmasked region from the original input. This ensures fair comparison across methods, since LDM-based approaches cannot perfectly reconstruct unedited regions.

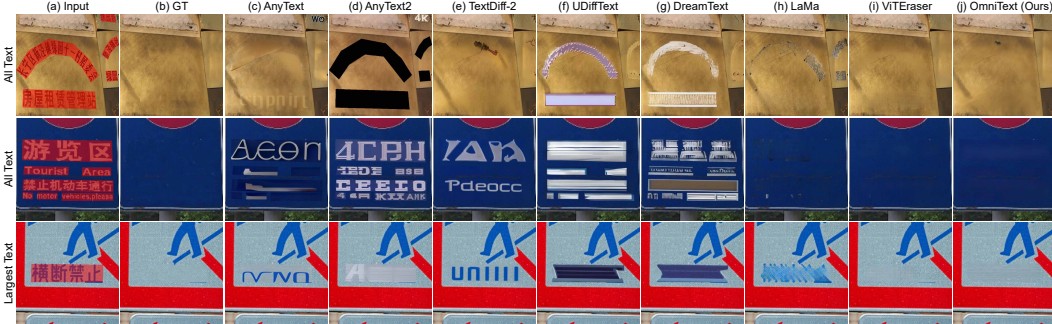

Figure 13: **Additional qualitative comparison on the standard benchmark for text removal.** Compared to other generalist baselines, our OmniText significantly outperforms existing text synthesis methods, which often generate text hallucinations or texture artifacts instead of effectively removing the text. Notably, in the 'largest text' removal setting, our method achieves better removal in rows 1 and 4 compared to LaMa (specialist).

In this section, we present a full comparisons of text removal in two settings: "all text" removal and "largest text" removal. Compared to generalist methods, as shown in Fig. 13, similar to the analyses in the main paper, our method outperforms other generalist text synthesis baselines, where text synthesis baselines often produce text hallucinations or texture artifacts. Compared to specialist methods, as shown in Fig. 13 (rows 1 and 3), LaMa sometimes produces artifacts, while our method achieves better removal. Meanwhile, our method does not outperform ViTEraser as ViTEraser is specifically trained for text removal. However, we note that our method is a generalist that can perform not only removal but also synthesis, editing, and other text image manipulation applications, all within one unified framework.

### J.1.2 TEXT EDITING (SCENEPAIR BENCHMARK)

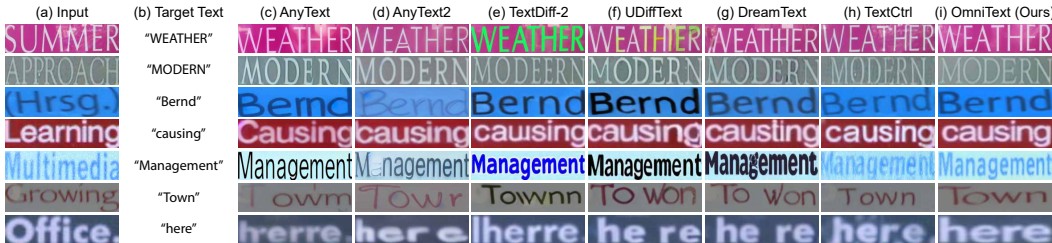

Figure 14: **Qualitative comparison of cropped outputs on the standard benchmark for text editing (ScenePair (Zeng et al., 2024)).** Our OmniText consistently outperforms other methods in terms of text style fidelity. Additionally, it produces fewer artifacts compared to other baselines.

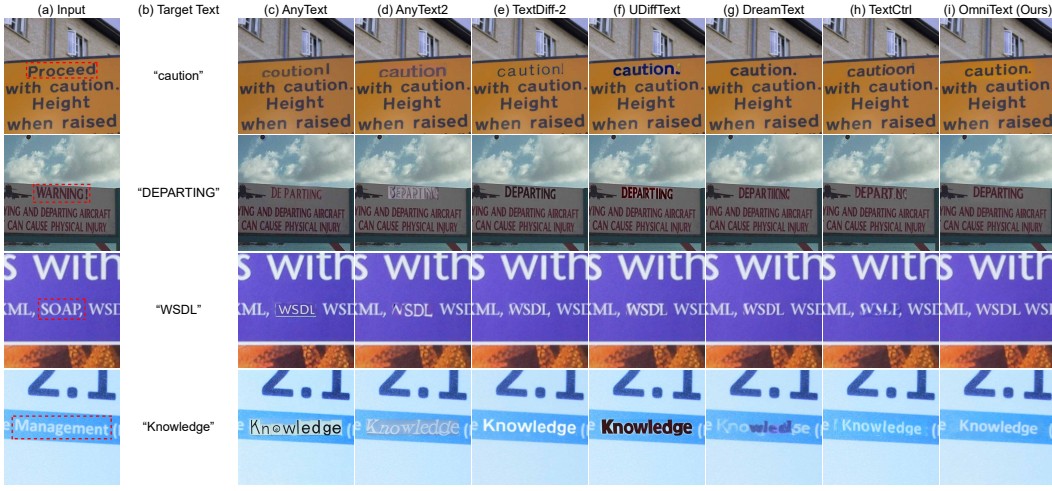

Figure 15: **Qualitative comparison of full images on the standard benchmark for text editing (ScenePair (Zeng et al., 2024)).** The dashed red box highlights the target region for editing. Our OmniText consistently outperforms other methods in both text style fidelity and content accuracy, while also producing fewer artifacts than other baselines.

As shown in Fig. 14 for the cropped image and Fig. 15 for the full image, our OmniText outperforms other methods in text style fidelity on the ScenePair dataset similar to the results in the main paper. The style fidelity of our OmniText is even superior to that of the specialist method, TextCtrl (Zeng et al., 2024), as seen in Fig. 14 (rows 1–2 and 5–7), without producing artifacts such as duplicated letters or inaccurate rendering (e.g., Fig. 15 - 1st to 3rd rows). Furthermore, compared to the specialist method, our OmniText is able to cover the full range of the mask, as shown in Fig. 14 (rows 2, 6, and 7).

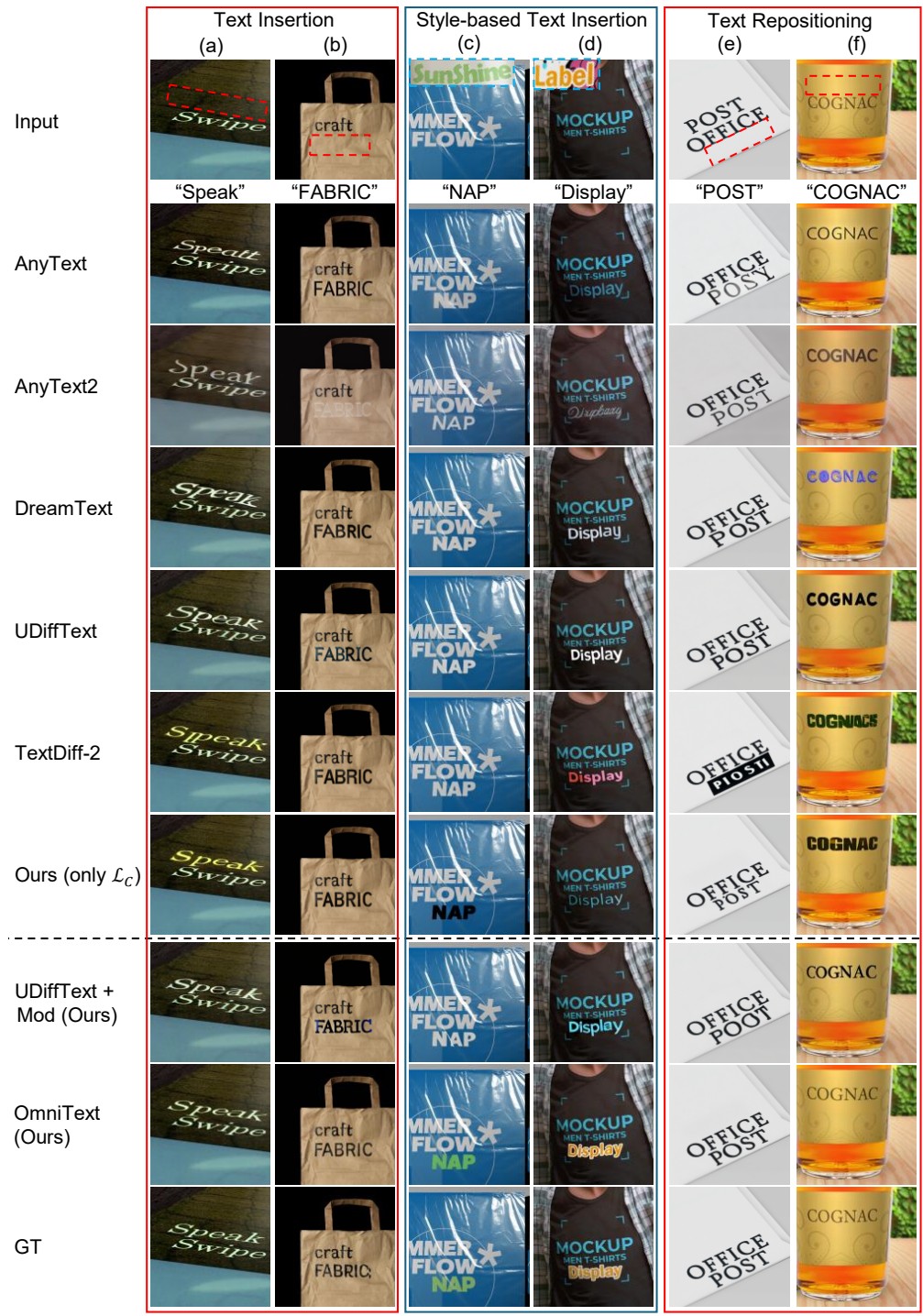

Figure 16: **Qualitative comparison on additional applications using our OmniText-Bench dataset.** We present qualitative comparisons for text insertion, style-based text insertion, and text repositioning. The red dashed box denotes the target region for manipulation, while the blue dashed box highlights the text reference used in style-based tasks. Note that other baselines rely on LaMa (a specialist model) for the removal step in text repositioning, whereas our OmniText performs removal using its own unified framework. OmniText significantly outperforms other baselines in terms of text style fidelity. Specifically, examples (b) and (e) demonstrate that our method can reliably preserve the original font structure and type. Additionally, examples (c), (d), and (f) show that OmniText can accurately transfer the reference style, including font structure, type, and color.

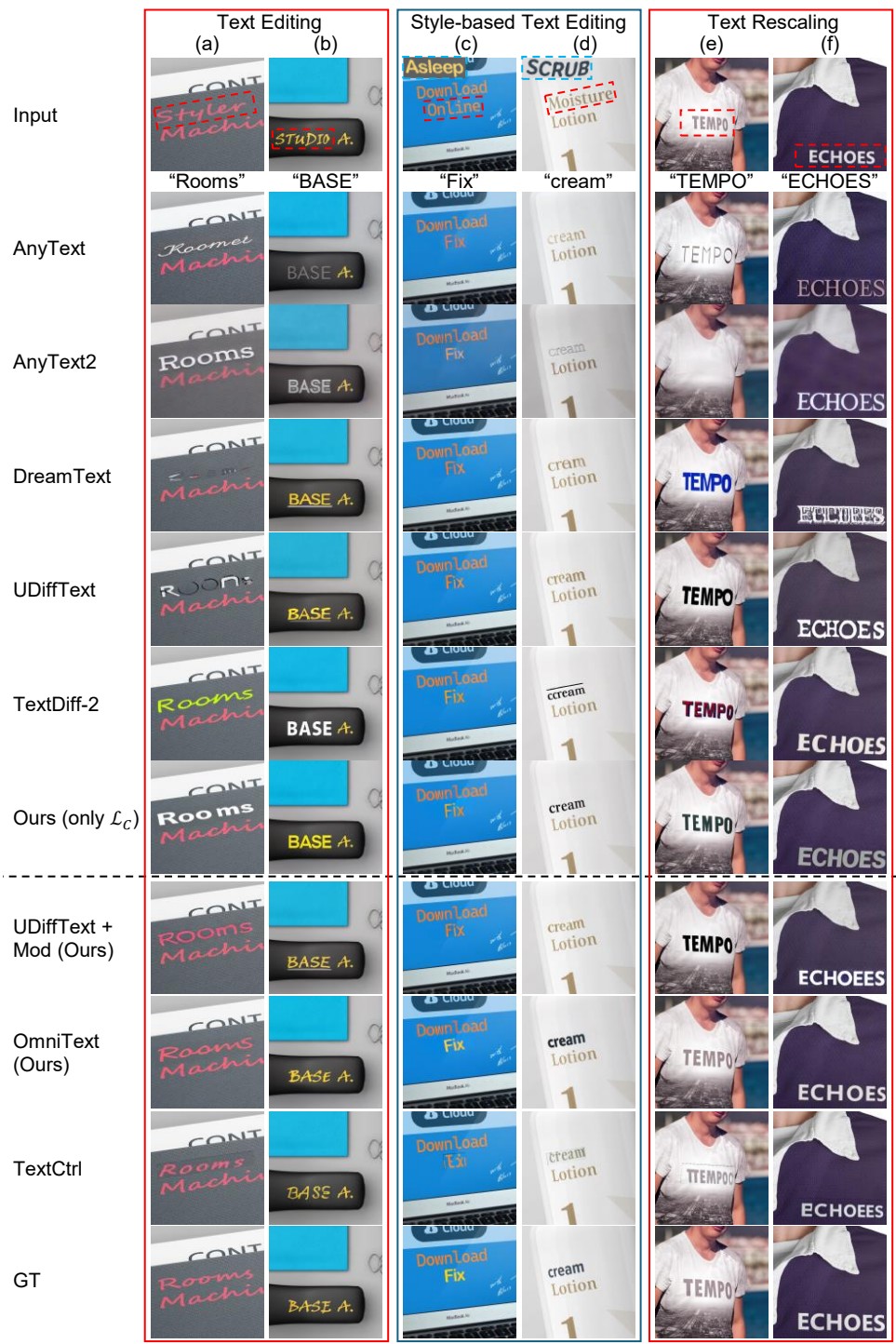

Figure 17: **Qualitative comparison on additional applications using our OmniText-Bench dataset.** We present qualitative comparisons for text editing, style-based text editing, and text rescaling. The red dashed box denotes the target region for manipulation, while the blue dashed box highlights the text reference used in style-based tasks. Note that other baselines rely on LaMa (a specialist model) for the removal step in all three tasks, whereas our OmniText performs removal using its own unified framework. We also include an additional specialist baseline, TextCtrl, which is heavily trained for text editing. OmniText significantly outperforms other baselines in terms of text style fidelity. Specifically, examples (a), (b), (e), and (f) demonstrate that our method reliably preserves the original font structure, type, and color of the input text. Additionally, examples (c) and (d) show that OmniText accurately transfers the text reference style, including font structure, type, and color.

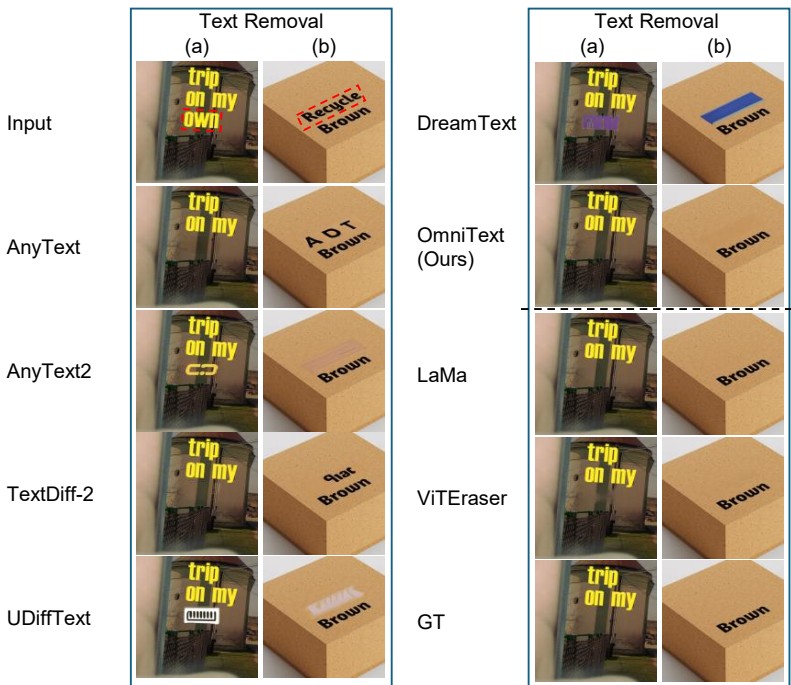

Figure 18: **Qualitative comparison on text removal using our OmniText-Bench dataset.** Our method consistently outperforms other generalist (text synthesis) baselines, which often generate text hallucinations or texture artifacts. Compared to specialist methods such as LaMa and ViTEraser, OmniText is comparable qualitatively. In scenes with highly structured backgrounds (e.g., poles), OmniText demonstrates superior performance in reconstructing fine structures after text removal compared to the specialist baseline ViTEraser. In addition to text removal, our OmniText can also perform other text image manipulation tasks, such as text editing, within a single unified framework.

Table 12: **Full comparison against baselines across text insertion, style-based text insertion, text editing, and style-based text editing in our OmniText-Bench.** It is worth noting that this comparison may be somewhat biased, as UDiffText and DreamText are trained on the synthetic text dataset SynthText (Gupta et al., 2016), which may share similar characteristics with our OmniText-Bench. This likely contributes to their higher accuracy in most cases. In contrast, our backbone (TextDiff-2) is not trained with SynthText. Despite this, our OmniText generally outperforms other baselines in terms of text style fidelity. Although its performance in style-based text insertion and style-based text editing is sometimes lower in terms of pixel-level metrics, this is likely due to reduced text accuracy, which affects these quantitative scores. **Bold** and underline denote the best and second-best performance in each category, respectively. **Red** highlights the best performance across all categories.

**Text Insertion**

| | Rendering Accuracy | | Style Fidelity | | | |
| --- | --- | --- | --- | --- | --- | --- |
| | ACC (%) ↑ | NED ↑ | MSE ↓ ($\times 10^{-2}$) | MS-SSIM ↑ ($\times 10^{-2}$) | PSNR ↑ | FID ↓ |
| AnyText (Tuo et al., 2023) | 72.00 | 0.903 | **6.19** | 33.70 | **13.27** | 71.95 |
| AnyText2 (Tuo et al., 2024) | 80.00 | 0.926 | 6.23 | 32.55 | 12.89 | 89.70 |
| DreamText (Wang et al., 2024) | 85.33 | 0.965 | 6.44 | 36.23 | 13.18 | 54.48 |
| UDiffText (Zhao & Lian, 2023) | 94.67 | 0.985 | 7.94 | 34.83 | 12.03 | 62.00 |
| TextDiff-2 (Chen et al., 2024) | 76.67 | 0.899 | 8.40 | 35.48 | 12.23 | 72.56 |
| Ours - Backbone: TextDiff-2 (only $\mathcal{L}_C$) | 85.33 | 0.962 | 8.19 | **39.15** | 12.47 | 63.78 |
| (Style-control) UDiffText + Mod (Ours) | **89.33** | **0.975** | 6.74 | 36.68 | 13.08 | 58.03 |
| (Style-control) Ours - Backbone: TextDiff-2 | 75.33 | 0.943 | 5.93 | 40.45 | 13.96 | **55.90** |

**Style-based Text Insertion**

| | Rendering Accuracy | | Style Fidelity | | | |
| --- | --- | --- | --- | --- | --- | --- |
| | ACC (%) ↑ | NED ↑ | MSE ↓ ($\times 10^{-2}$) | MS-SSIM ↑ ($\times 10^{-2}$) | PSNR ↑ | FID ↓ |
| AnyText (Tuo et al., 2023) | 70.00 | 0.900 | 9.01 | 25.33 | 11.48 | 89.32 |
| AnyText2 (Tuo et al., 2024) | 80.67 | 0.918 | 7.59 | **25.79** | 12.12 | 101.22 |
| DreamText (Wang et al., 2024) | 78.67 | 0.941 | 10.06 | 24.11 | 10.67 | **79.71** |
| UDiffText (Zhao & Lian, 2023) | 88.00 | 0.973 | 11.79 | 24.43 | 9.98 | 82.21 |
| TextDiff-2 (Chen et al., 2023b) | 66.67 | 0.856 | 11.80 | 25.02 | 10.39 | 93.52 |
| Ours - Backbone: TextDiff-2 (only $\mathcal{L}_C$) | 78.00 | 0.946 | 11.63 | 24.61 | 10.12 | 84.32 |
| (Style-control) UDiffText + Mod (Ours) | **86.67** | **0.968** | 9.98 | 26.64 | 10.80 | 81.21 |
| (Style-control) Ours - Backbone: TextDiff-2 | 59.33 | 0.904 | **9.82** | 26.98 | **11.08** | 77.37 |

**Text Editing**

| | Rendering Accuracy | | Style Fidelity | | | |
| --- | --- | --- | --- | --- | --- | --- |
| | ACC (%) ↑ | NED ↑ | MSE ↓ ($\times 10^{-2}$) | MS-SSIM ↑ ($\times 10^{-2}$) | PSNR ↑ | FID ↓ |
| AnyText (Tuo et al., 2023) | 81.33 | 0.928 | **6.68** | **35.24** | **13.05** | 77.07 |
| AnyText2 (Tuo et al., 2024) | 86.00 | 0.951 | 6.94 | 30.31 | 12.53 | 98.14 |
| DreamText (Wang et al., 2024) | 84.00 | 0.954 | 7.99 | 34.17 | 12.23 | 65.83 |
| UDiffText (Zhao & Lian, 2023) | 96.00 | 0.984 | 9.78 | 31.22 | 11.17 | 68.67 |
| TextDiff-2 (Chen et al., 2023b) | 76.00 | 0.932 | 10.57 | 31.04 | 11.28 | 70.70 |
| Ours - Backbone: TextDiff-2 (only $\mathcal{L}_C$) | 82.00 | 0.958 | 9.49 | 32.08 | 11.46 | 68.95 |
| (Style-control) UDiffText + Mod (Ours) | **90.67** | **0.981** | 6.33 | 38.82 | 13.34 | 55.23 |
| (Style-control) Ours - Backbone: TextDiff-2 | 76.00 | 0.947 | 5.51 | 41.09 | 14.25 | 49.44 |
| (Specialist) TextCtrl (Zeng et al., 2024) | 87.33 | 0.963 | 6.02 | 34.47 | 13.56 | 56.22 |

**Style-based Text Editing**

| | Rendering Accuracy | | Style Fidelity | | | |
| --- | --- | --- | --- | --- | --- | --- |
| | ACC (%) ↑ | NED ↑ | MSE ↓ ($\times 10^{-2}$) | MS-SSIM ↑ ($\times 10^{-2}$) | PSNR ↑ | FID ↓ |
| AnyText (Tuo et al., 2023) | 72.00 | 0.901 | 8.70 | 27.40 | 11.51 | 96.60 |
| AnyText2 (Tuo et al., 2024) | 82.00 | 0.947 | 7.65 | 27.57 | 12.06 | 110.57 |
| DreamText (Wang et al., 2024) | 79.33 | 0.950 | 9.98 | 24.78 | 10.63 | 86.09 |
| UDiffText (Zhao & Lian, 2023) | 94.67 | 0.986 | 11.97 | 24.96 | 9.86 | 87.70 |
| TextDiff-2 (Chen et al., 2023b) | 68.67 | 0.898 | 12.66 | 23.06 | 9.86 | 89.72 |
| Ours - Backbone: TextDiff-2 (only $\mathcal{L}_C$) | 76.67 | 0.946 | 12.18 | 24.34 | 9.95 | **85.48** |
| (Style-control) UDiffText + Mod (Ours) | **81.33** | **0.964** | 10.22 | 26.32 | 10.58 | 84.78 |
| (Style-control) Ours - Backbone: TextDiff-2 | 65.33 | 0.930 | 9.89 | 26.50 | 11.10 | 78.03 |
| TextCtrl (Zeng et al., 2024) | 77.33 | 0.929 | **9.28** | 26.81 | **11.28** | 79.54 |

Table 13: **Full comparison against baselines across text rescaling and repositioning in our OmniText-Bench.** Note that other baselines rely on LaMa (a specialist model) for the removal step in text repositioning and rescaling, whereas our OmniText performs removal using its own unified framework. It is worth noting that this comparison may be somewhat biased, as UDiffText and DreamText are trained on the synthetic text dataset SynthText (Gupta et al., 2016), which may share similar characteristics with our OmniText-Bench. This likely contributes to their higher accuracy in most cases. In contrast, our backbone (TextDiff-2) is not trained with SynthText. Despite this, our OmniText generally outperforms other baselines in terms of text style fidelity. Although its performance is sometimes lower in terms of pixel-level metric (i.e., MS-SSIM), this is likely due to reduced text accuracy, which affects these quantitative scores. **Bold** and underline denote the best and second-best performance in each category, respectively. **Red** highlights the best performance across all categories.

| | Rendering Accuracy | | Style Fidelity | | | |
|---|---|---|---|---|---|---|
| **Text Rescaling** | ACC (%) ↑ | NED ↑ | MSE ↓ (×10⁻²) | MS-SSIM ↑ (×10⁻²) | PSNR ↑ | FID ↓ |
| AnyText (Tuo et al., 2023) | 59.33 | 0.870 | 7.75 | 24.95 | **12.04** | 85.84 |
| AnyText2 (Tuo et al., 2024) | 74.67 | 0.912 | **7.41** | 25.53 | 12.00 | 105.45 |
| DreamText (Wang et al., 2024) | 72.00 | 0.931 | 9.22 | 26.82 | 11.32 | 72.84 |
| UDiffText (Zhao & Lian, 2023) | **88.67** | **0.978** | 11.11 | **28.56** | 10.28 | 76.51 |
| TextDiff-2 (Chen et al., 2023b) | 59.33 | 0.907 | 11.38 | 23.17 | 10.60 | 79.21 |
| Ours - Backbone: TextDiff-2 (only $\mathcal{L}_C$) | 66.67 | 0.917 | 10.59 | 24.35 | 10.87 | 75.38 |
| (Style-control) UDiffText + Mod (Ours) | 56.00 | **0.918** | 7.98 | **28.30** | 11.94 | 62.69 |
| (Style-control) Ours - Backbone: TextDiff-2 | 33.33 | 0.868 | **7.30** | 25.90 | **12.40** | **59.68** |
| TextCtrl (Zeng et al., 2024) | **64.00** | 0.907 | 7.47 | 24.85 | 12.31 | 68.55 |

| | Rendering Accuracy | | Style Fidelity | | | |
|---|---|---|---|---|---|---|
| **Text Repositioning** | ACC (%) ↑ | NED ↑ | MSE ↓ (×10⁻²) | MS-SSIM ↑ (×10⁻²) | PSNR ↑ | FID ↓ |
| AnyText (Tuo et al., 2023) | 72.67 | 0.905 | 6.97 | 28.73 | **12.54** | 86.60 |
| AnyText2 (Tuo et al., 2024) | 81.33 | 0.948 | **6.59** | 27.11 | 12.50 | 108.17 |
| DreamText (Wang et al., 2024) | 83.33 | 0.958 | 7.87 | **32.44** | 12.03 | **69.15** |
| UDiffText (Zhao & Lian, 2023) | **92.00** | **0.985** | 10.35 | 30.15 | 10.84 | 74.13 |
| TextDiff-2 (Chen et al., 2023b) | 64.00 | 0.906 | 11.68 | 27.18 | 10.66 | 81.06 |
| Ours - Backbone: TextDiff-2 (only $\mathcal{L}_C$) | 74.67 | 0.944 | 9.83 | 31.21 | 11.37 | 77.90 |
| (Style-control) UDiffText + Mod (Ours) | **88.67** | **0.978** | 6.49 | **38.89** | 13.11 | 58.51 |
| (Style-control) Ours - Backbone: TextDiff-2 | 69.33 | 0.943 | **5.96** | 38.83 | **13.85** | **52.17** |

Table 14: **Full comparison against baselines in text removal applications using our OmniText-Bench.** MSE is reported on a scale of $10^{-3}$. Our OmniText significantly outperforms other generalist (text synthesis) baselines. As expected, it performs comparably to specialist methods that are extensively trained for general inpainting (LaMa) and dedicated text removal (ViTEraser).

| | Text Removal | | | |
|---|---|---|---|---|
| Metrics
Methods | MS-SSIM ↑ (×10⁻²) | PSNR ↑ | MSE ↓ | FID ↓ |
| AnyText (Tuo et al., 2023) | 96.83 | 30.19 | 2.59 | 31.18 |
| AnyText2 (Tuo et al., 2024) | 96.01 | 26.87 | 4.05 | 44.12 |
| TextDiff-2 (Chen et al., 2024) | 97.06 | 34.99 | 3.16 | 20.75 |
| UDiffText (Zhao & Lian, 2023) | 95.13 | 21.84 | 8.69 | 52.13 |
| DreamText (Wang et al., 2024) | 95.31 | 22.92 | 8.01 | 47.57 |
| OmniText (Ours) | **98.16** | **43.48** | **0.39** | **11.14** |
| (Specialist) LaMa (Suvorov et al., 2022) | 98.22 | 46.15 | 0.26 | 5.39 |
| (Specialist) ViTEraser (Peng et al., 2024) | **98.78** | **48.34** | **0.23** | **5.30** |

## J.2 FULL COMPARISON ON OMNITEXT-BENCH

The quantitative results of our method compared to other baselines using our OmniText-Bench can be seen in Table 12, Table 13, and Table 14. The qualitative results are shown in Fig. 16, Fig. 17, and Fig. 18. Based on these results, several key observations can be made.

**Generalization of OmniText.** As current baseline is incapable of performing style control, we further apply our OmniText method to UDiffText. Specifically, we modify the inpainting backbones by incorporating our proposed Grid Trick, which guides self-attention to focus on the style features

from a reference image, to handle tasks that require style reference such as style-based text insertion and editing. However, we do not perform latent optimization due to the specialized architecture of these backbones. Instead, we implement self-attention modulation during sampling, resulting in a modified baseline denoted as UDiffText + Mod (Ours). This self-attention modulation shifts the self-attention weights within the target text mask toward the reference attention distribution. The reference attention is obtained by normalizing the reference mask to sum to one (details provided in the main paper).

UDiffText uses 50 sampling steps, where we apply our self-attention modulation during the middle phase of sampling, specifically from 40% to 80% of the total steps. The modulation is applied to the decoder of UDiffText, specifically at Block 1, Layer 0 and Block 2, Layer 1, aligning the self-attention behavior with the reference style.

The different configurations for our backbone and UDiffText show that each backbone has different layers in controlling the style. Thus, one can extend our method to another backbone, but additional analyses are needed to choose the correct layers and sampling steps.

**Bias of Certain Text Synthesis Baselines.** As shown in Table 12 and Table 13, UDiffText achieves the best performance in rendering accuracy across all text manipulation tasks. Upon careful analysis, we found that UDiffText is optimized for accurate text rendering, regardless of style, and was trained on the SynthText (Gupta et al., 2016) dataset. This training causes the method to perform well in rendering accurate text on our OmniText-Bench dataset, as our dataset contains synthetic text generated using image editing tools. As a result, UDiffText performs better across all tasks that require precise text rendering.

However, due to the nature of its architecture and training, this baseline is less general, as it performs worst in text removal (Table 14). Additionally, it synthesizes only accurate text without regard to style, as seen in Fig. 16 and Fig. 17, particularly in Fig. 16 (b and d). When selecting our backbone, we considered its performance across all tasks and on real datasets, where only TextDiff-2 consistently performs well across all tasks.

**Performance Across All Tasks.** As shown in Table 12, Table 13, and Table 14, our OmniText outperforms other generalist text synthesis baselines in text style fidelity across all tasks, particularly in text insertion, editing, rescaling, and repositioning. For the removal task, only our method is capable of performing text removal, while other text synthesis baselines fail. This is evident in Fig. 18, where other baselines tend to produce texture artifacts or text hallucinations.

For text synthesis-related tasks, our OmniText achieves only comparable performance in text accuracy, as it depends on the performance of the TextDiff-2 backbone. As seen in Table 12 and Table 13, the TextDiff-2 backbone performs worse in text accuracy. However, it achieves strong text accuracy performance on real datasets like ScenePair (Zeng et al., 2024) compared to the baselines. The lower accuracy of TextDiff-2 is due to the nature of its training data, which consists mostly of real images with real text, while OmniText-Bench is closer to synthetic dataset. In contrast, other backbones, which are trained with additional synthetic data, perform better on synthetic data, including our OmniText-Bench.

**Strong Text Style Fidelity Performance.** As shown in Table 12 and Table 13, our OmniText outperforms other generalist text synthesis baselines in terms of text style fidelity across all tasks, particularly in text insertion, editing, rescaling, and repositioning. This is further supported by Fig. 16 and Fig. 17, where OmniText successfully matches the reference text style in style-based text insertion and style-based text editing, compared to other baselines, including the modified version of UDiffText ("UDiffText + Mod"). Notably, OmniText can match challenging font styles, structures, and colors, as demonstrated in Fig. 16 (b, c, d, f) and Fig. 17 (a, b, c, d, e).

However, despite its superior qualitative performance, OmniText achieves competitive performance in MS-SSIM scores for the text repositioning task and comparable MS-SSIM scores for the text rescaling task, as shown in Table 13. The reason for this is that MS-SSIM is also related to rendering accuracy, where OmniText performs lower compared to "UDiffText + Mod", resulting in a lower MS-SSIM score. This observation is also seen in style-based text insertion and style-based text editing tasks in Table 12, where the lower accuracy of our method leads to worse MSE, MS-SSIM, and PSNR scores compared to the baselines. Notably, AnyText2 achieves better MSE, MS-SSIM, and PSNR in these tasks because it is more accurate than our baselines. Additionally, AnyText2

frequently uses the "Times" and "Arial" fonts during synthesis (see Fig. 16 b, c, e, f and Fig. 17 a, b, c, d, f), which are commonly employed in design. This contributes to its better performance in these tasks. This highlights the limitation of the style fidelity metric, as it cannot fully capture text style performance without considering accuracy. One potential solution is to use the TextCtrl (Zeng et al., 2024) text style embedding as a style similarity metric. However, due to the lack of a user study and the method's limited robustness (being trained only for text without curves or significant perspective), we cannot currently use it in its present form. Since style-based text insertion and editing are particularly challenging, the style fidelity scores tend to be lower overall.

**Trade-off Between Text Accuracy and Style Fidelity.** As shown in Table 12 and Table 13, there is a trade-off between style-optimized and accuracy-optimized methods. Specifically, the first group in each table represents methods that either disregard the text style or only copy the style from surrounding text. Our content loss (Ours with only $\mathcal{L}_C$) improves rendering accuracy of the TextDiff-2 backbone across all tasks. However, when we optimize for text style fidelity, rendering accuracy drops for challenging tasks such as text insertion, style-based text insertion, text rescaling, and style-based text editing, with a significant performance drop observed in text rescaling.

This drop is due to the nature of the TextDiff-2 backbone, which tends to produce duplicated letters even with our content control. When optimizing for style, the method copies the style of the reference text. If the reference text has enough spacing, this will be reflected as well. However, if the reference text is tightly spaced, the backbone produces duplicated letters to fill the remaining space in the mask. A potential solution is to fine-tune the backbone to handle this issue. Similarly, a drop in performance can be observed when optimizing for style with another backbone (UDiffText), where text accuracy drops across all cases, with the most significant decline seen in text rescaling tasks. This shows the general issue of this trade-off that also impacts other text synthesis baselines. Nonetheless, our method successfully preserves the accuracy of the original backbone and even improves it in standard tasks like text editing and text repositioning, where the mask does not change significantly. Further analysis of this issue can be found in Section O.

**Comparison against specialists.** We compare our method against specialists across various tasks. For text removal, we compare our method to the general inpainting backbone (LaMa) and the state-of-the-art text removal method (ViTEraser). As shown in Table 14, our OmniText achieves comparable results to the specialists, which is expected since these methods are specifically trained to remove regions (LaMa) or text (ViTEraser).

However, as shown in Fig. 18, ViTEraser shows artifacts (e.g., pole) in the removed text compared to LaMa, despite its better quantitative performance in Table 14. Notably, our OmniText achieves competitive performance compared to the specialists, as shown in Table 14, and significantly outperforms other text synthesis baselines in the removal task.

For other applications involving text editing (text editing, style-based text editing, and text rescaling), we compare our OmniText with the specialist TextCtrl (Zeng et al., 2024). As shown in Table 12, OmniText outperforms the specialist in text style fidelity for text editing, despite having lower accuracy. This highlights the advantage of our style control contribution. For text rescaling, OmniText also outperforms the specialist, as TextCtrl was not trained for this task. This demonstrates the significance of OmniText as a generalist method, which does not require retraining to perform tasks compared to specialists that need to be retrained for additional tasks. However, in style-based text editing, our method achieves competitive results against the specialist. This is likely due to lower rendering accuracy, as style fidelity scores are also related to rendering accuracy. Our method has lower accuracy compared to TextCtrl, which impacts the style fidelity score. Overall, OmniText is a generalist method that covers a wide range of applications without the need for retraining.

### J.3 ADDITIONAL ABLATION STUDIES

#### J.3.1 TEXT REMOVAL (TR)

We present both quantitative and qualitative results when varying the number of steps used for applying each of our techniques, including Cross-Attention Reassignment (CAR) and Self-Attention Inversion (SAI). As shown in Table 15, the performance of text removal drops significantly without SAI. When SAI is added to more sampling steps, the performance improves. However, the performance increase plateaus and then declines when SAI is applied beyond 50% of the sampling steps. This can be

Table 15: **Ablation study on the number of steps at which CAR and SAI are applied. Bold** and underline denote the best and second-best performance in each category. When CAR is fixed at 100%, removing SAI leads to a significant drop in performance. Increasing the number of steps at which SAI is applied tends to improve performance, with gains plateauing around 50% of the steps. Conversely, when SAI is fixed at 50% steps, applying CAR improves performance in the 'largest text' removal setting, with performance gains plateauing after 50% steps.

| | Text Removal (SCUT-EnsText (Liu et al., 2020)) | | | | | | | |
|---|---|---|---|---|---|---|---|---|
| Metrics | All Text Removal | | | | Largest Text Removal | | | |
| Methods | MS-SSIM $\uparrow (\times 10^{-2})$ | PSNR $\uparrow$ | MSE $\downarrow$ | FID $\downarrow$ | MS-SSIM $\uparrow (\times 10^{-2})$ | PSNR $\uparrow$ | MSE $\downarrow$ | FID $\downarrow$ |
| CAR (100%) + SAI (0%) | 94.02 | 26.09 | 8.18 | 48.86 | 97.27 | 30.02 | 4.14 | 21.44 |
| CAR (100%) + SAI (25%) | 95.19 | 28.61 | 3.96 | 42.63 | 97.87 | 32.68 | 1.94 | 17.54 |
| CAR (100%) + SAI (50%) - Ours | 95.71 | **29.52** | **3.44** | **39.06** | 98.21 | **33.90** | **1.61** | 15.33 |
| CAR (100%) + SAI (100%) | **95.81** | 28.91 | 4.58 | 40.42 | **98.42** | **33.90** | 1.74 | **14.17** |
| SAI (50%) + CAR (0%) | 95.56 | **30.02** | **3.01** | **37.31** | 97.58 | 33.12 | 1.89 | 16.70 |
| SAI (50%) + CAR (25%) | 95.54 | 29.81 | 3.67 | 37.69 | **98.24** | **34.02** | 1.64 | **15.25** |
| SAI (50%) + CAR (50%) | 95.36 | 29.53 | 3.85 | 38.81 | **98.24** | **34.02** | 1.64 | **15.25** |
| SAI (50%) + CAR (100%) - Ours | **95.71** | 29.52 | 3.44 | 39.06 | 98.21 | 33.90 | **1.61** | 15.33 |

(a) SAI Ablation - CAR (100%)

(b) CAR Ablation - SAI (50%)

Figure 19: **Ablation study on the number of steps at which CAR and SAI are applied.** When CAR is fixed at 100%, removing SAI leads to text hallucinations. (a) Increasing the number of steps at which SAI is applied tends to improve performance, but applying too much SAI (100%) introduces texture artifacts (indicated by the red arrow). (b) Conversely, when SAI is fixed at 50% of the steps, applying CAR improves performance and plateaus at 50%. Applying excessive CAR (100%) results in color shifting artifacts in gray regions (marked by the red arrow).

explained by Fig. 19 - a, where adding SAI after successful text removal can introduce texture artifacts (as seen in the SAI 100% case).

Adding CAR degrades the performance of "all text" removal but improves the performance of "largest text" removal up to 50% of the sampling steps. Beyond this point, the performance starts to degrade slightly. Since our goal is to achieve robust text removal that can handle a wide range of cases, we choose to use CAR at 100%. The reason for the lower performance can be seen in Fig. 19 - b (red arrow), where adding CAR removes the text but introduces slight color shifting (gray region). Increasing CAR further tends to amplify this effect. Overall, the chosen configuration for our text removal technique provides robust performance in both "all text" and "largest text" removal scenarios.

### J.3.2 CONTROLLABLE INPAINTING (CI)

**Ablation Study on Loss Weights.** We report the results of loss weight ablations for our Controllable Inpainting (CI) method in Table 16. The first group focuses on ablation of the style loss weight, $\lambda_S$. As shown, performing optimization without the Grid Trick ($G$) and using only our content loss ($\mathcal{L}_C$) improves rendering accuracy. However, once we include $G$, rendering accuracy drops significantly, but style fidelity starts to improve. This is likely because $G$ imposes strong style control, even without the style loss. Increasing the weight factor for $\mathcal{L}_S$ also significantly improves style fidelity but only increases the accuracy up to the point where $\lambda_S = 10$. Beyond this point, accuracy starts to decrease, and style fidelity improvement begins to plateau. This demonstrates the complex relationship between content loss ($\mathcal{L}_C$) and style loss ($\mathcal{L}_S$) when $G$ is integrated into our framework. Note that there is a trade-off between accuracy and style fidelity: when optimizing for style fidelity,

Table 16: **Ablation study of loss weight hyperparameters ($\lambda_C$ and $\lambda_S$) on ScenePair (Zeng et al., 2024).** When $\lambda_C$ is fixed, introducing the Grid Trick ($G$) reduces rendering accuracy due to the dominance of style control. Adding the style loss $\mathcal{L}_S$ and increasing $\lambda_S$ generally improves both rendering accuracy and style fidelity, with rendering accuracy plateauing at $\lambda_S = 10$. Further increases in $\lambda_S$ slightly improve style fidelity but lead to a decline in rendering accuracy. In contrast, when $\lambda_S$ is fixed, adding the content loss $\mathcal{L}_C$ and increasing $\lambda_C$ tends to improve rendering accuracy, which plateaus between $\lambda_C = 2.5$ and $5$.

| | Text Editing (ScenePair (Zeng et al., 2024)) | | | | | |
|---|---|---|---|---|---|---|
| | Rendering Accuracy | | Style Fidelity | | | |
| | ACC (%) ↑ | NED ↑ | MSE ↓ ($\times 10^{-2}$) | MS-SSIM ↑ ($\times 10^{-2}$) | PSNR ↑ | FID ↓ |
| $\mathcal{L}_C$ ($\lambda_C = 5$) | **88.52** | **0.970** | 8.75 | 29.90 | 12.01 | 38.85 |
| $\mathcal{L}_C$ ($\lambda_C = 5$) + G | 75.47 | 0.949 | 6.86 | 33.94 | 13.30 | 34.76 |
| $\mathcal{L}_C$ ($\lambda_C = 5$) + G + $\mathcal{L}_S$ ($\lambda_S$=1) | 77.03 | 0.949 | 5.48 | 37.21 | 14.23 | 32.86 |
| $\mathcal{L}_C$ ($\lambda_C = 5$) + G + $\mathcal{L}_S$ ($\lambda_S = 10$) - Ours | 78.44 | 0.951 | 4.79 | 40.11 | 14.85 | 31.69 |
| $\mathcal{L}_C$ ($\lambda_C = 5$) + G + $\mathcal{L}_S$ ($\lambda_S$=15) | 78.13 | 0.950 | **4.78** | **40.18** | **14.86** | 31.59 |
| $\mathcal{L}_C$ ($\lambda_C = 5$) + G + $\mathcal{L}_S$ ($\lambda_S$=20) | 78.13 | 0.951 | **4.78** | 40.17 | **14.86** | **31.56** |
| G + $\mathcal{L}_S$ ($\lambda_S = 10$) | 78.28 | 0.949 | 4.78 | 40.19 | 14.86 | 31.64 |
| G + $\mathcal{L}_S$ ($\lambda_S = 10$) + $\mathcal{L}_C$ ($\lambda_C = 1$) | 78.28 | 0.951 | 4.78 | 40.17 | 14.86 | 31.60 |
| G + $\mathcal{L}_S$ ($\lambda_S = 10$) + $\mathcal{L}_C$ ($\lambda_C = 2.5$) | 78.44 | 0.951 | 4.78 | 40.16 | 14.86 | 31.59 |
| G + $\mathcal{L}_S$ ($\lambda_S = 10$) + $\mathcal{L}_C$ ($\lambda_C = 5$) | 78.44 | 0.951 | 4.79 | 40.11 | 14.85 | 31.69 |
| G + $\mathcal{L}_S$ ($\lambda_S = 10$) + $\mathcal{L}_C$ ($\lambda_C = 10$) | 77.81 | 0.951 | 4.83 | 39.84 | 14.80 | 31.64 |

Table 17: **Ablation study of Self-Attention Manipulation (SAM) on ScenePair (Zeng et al., 2024).** SAM improves rendering accuracy but slightly reduces style fidelity. The improvement is more pronounced when only the content loss $\mathcal{L}_C$ is used during optimization. However, when the Grid Trick ($G$) and style loss $\mathcal{L}_S$ are also applied, the gain in accuracy reduces, likely due to the strong style control imposed by $G$ and $\mathcal{L}_S$.

| | Text Editing (ScenePair (Zeng et al., 2024)) | | | | | |
|---|---|---|---|---|---|---|
| | Rendering Accuracy | | Style Fidelity | | | |
| | ACC (%) ↑ | NED ↑ | MSE ↓ ($\times 10^{-2}$) | MS-SSIM ↑ ($\times 10^{-2}$) | PSNR ↑ | FID ↓ |
| $\mathcal{L}_C$ | 87.19 | 0.969 | **6.66** | **35.14** | **13.27** | **34.50** |
| $\mathcal{L}_C$ + SAM | **88.52** | **0.970** | 8.75 | 29.90 | 12.01 | 38.85 |
| $\mathcal{L}_C$ + G + $\mathcal{L}_S$ | 78.20 | **0.952** | **4.77** | **40.15** | **14.87** | **31.44** |
| $\mathcal{L}_C$ + G + $\mathcal{L}_S$ + SAM | **78.44** | 0.951 | 4.79 | 40.11 | 14.85 | 31.69 |

the text font, including spacing and font size, is fully copied. Thus, accuracy only depends on whether the backbone can create sufficient spacing between letters to maintain good accuracy.

In the second group (second row) of Table 16, increasing $\lambda_C$ generally improves rendering accuracy but reduces style fidelity. The performance improves up to $\lambda_C = 2.5$ and $\lambda_C = 5$, but further increases in $\lambda_C$ begin to degrade both rendering accuracy and style fidelity. However, note that our method is training-free and works on a case-by-case basis. Therefore, these results serve as guides, and the hyperparameters can still be adjusted based on the specific case at hand.

**Ablation Study on Self Attention Manipulation (SAM).** In Table 17, we present the results of using Self-Attention Manipulation (SAM) under two settings: (1) when optimizing with $\mathcal{L}_C$ alone, and (2) when optimizing with both $\mathcal{L}_C$ and $\mathcal{L}_S$. The results show that SAM improves rendering accuracy when $\mathcal{L}_C$ is used alone, although this comes at the cost of reduced text style fidelity, highlighting its effectiveness in enhancing content control.

However, when both content and style losses are used, the accuracy improvement from SAM is diminished. We attribute this to the strong style control enforced by the Grid Trick, which prioritizes style features over content, as also indicated in Table 16. As a result, the relative contribution of SAM, which is primarily designed to enhance content control, is reduced.

**Ablation Study on Number of Optimization Steps.** As shown in Table 18, the performance is *robust* to the number of optimization iterations: ACC fluctuates by only 0–1%, and NED varies by at most 0.001 across different iteration counts. In contrast, reducing the number of iterations consistently degrades style fidelity, but only slightly. For the lowest number of iterations, MSE worsens by $\sim 0.3$, MS-SSIM by $\sim 0.8$, PSNR by $\sim 0.2$, and FID by $\sim 0.2$. Note that the results indicate a tendency for rendering accuracy to increase when style fidelity decreases. This reflects the content–style fidelity

Table 18: **Ablation study of number of iteration steps of our latent optimization on ScenePair (Zeng et al., 2024) using best loss configuration of** $\mathcal{L}_C$ ($\lambda_C = 5$) **+ G +** $\mathcal{L}_S$ ($\lambda_S = 10$)**.** We apply optimization att three stages of the denoising process: the initial step (0%), the 20% step, and the 40% step, while varying the number of iterations per optimization. **Bold** and underline denote the best and second-best performance. Results show that the performance is robust to the number of optimization iterations.

| | Text Editing (ScenePair (Zeng et al., 2024)) | | | | | | |
|---|---|---|---|---|---|---|---|
| | Rendering Accuracy | | Style Fidelity | | | | Runtime (s) |
| | ACC (%) ↑ | NED ↑ | MSE ↓ (×10⁻²) | MS-SSIM ↑ (×10⁻²) | PSNR ↑ | FID ↓ | |
| 20 Iterations | 78.44 | 0.951 | **4.79** | **40.11** | **14.85** | **31.69** | 69.461 |
| 15 Iterations | **79.30** | **0.953** | 4.81 | 40.09 | 14.83 | 32.54 | 58.803 |
| 10 Iterations | 78.98 | **0.953** | 4.93 | 39.74 | 14.74 | 33.05 | 46.910 |
| 5 Iterations | 78.36 | 0.952 | 5.08 | 39.29 | 14.61 | 33.49 | 34.531 |

Figure 20: **Qualitative comparison between our OmniText and diffusion-based inpainting methods.** Diffusion-based text inpainting methods often produce text hallucinations, even when the prompt is explicitly modified to exclude any text. Among them, only one example using SDXL successfully removes the text, but still introduces texture artifacts.

trade-off discussed in the main paper. Although using 15 iterations leads to a minor drop in style fidelity, it yields the best rendering accuracy overall. This suggests that 15 iterations achieves the most balanced trade-off between rendering accuracy and style fidelity.

Overall, the ablation studies on the number of optimization steps and loss weights demonstrate that our method is *robust* to both the choice of iteration count and the loss-weight configuration, with only minor fluctuations in the results.

### J.4 COMPARISON AGAINST ALTERNATIVE METHODS

#### J.4.1 DIFFUSION-BASED INPAINTING FOR REMOVAL

In this comparison, we evaluate our text removal method against current diffusion-based inpainting methods. For the diffusion-based inpainting methods, we use the "background image" as the prompt and "text, letter, font" as the negative prompt. We experimented with other prompts, but this combination proved to be the most effective in most cases. As shown in Table 19, our OmniText outperforms other diffusion-based inpainting methods across all settings, both in "all text" and "largest text" removal, in all metrics.

The superior performance of our method in text removal is further supported by Fig. 20, where diffusion-based inpainting baselines tend to insert text into the target mask, even though we use prompts specifically designed to prevent text appearance. This issue is likely due to the nature of the training data, which often includes images of signs, posters, or boxes that contain text. The only successful case is the SDXL baseline, which successfully removes text from the sign in Fig. 20 (row 1). This highlights why we do not consider diffusion-based inpainting methods as specialists, as traditional (non-diffusion-based) methods like LaMa tend to perform better.

Table 19: **Quantitative comparison between OmniText and other diffusion-based text inpainting methods.** **Bold** and underline indicate the best and second-best performance in each category, respectively. **Red** highlights the best performance across all categories. MSE is reported on a scale of $10^{-3}$. We include LaMa, a non-diffusion-based inpainting method, as a specialist baseline. Our OmniText outperforms other diffusion-based methods in both the "all text" and "largest text" removal settings. Notably, our method also surpasses LaMa, demonstrating the strong potential of OmniText for text removal.

| | Text Removal (SCUT-EnsText (Liu et al., 2020)) | | | | | | | |
|---|---|---|---|---|---|---|---|---|
| Metrics | All Text Removal | | | | Largest Text Removal | | | |
| Methods | MS-SSIM ↑ ($\times 10^{-2}$) | PSNR ↑ | MSE ↓ | FID ↓ | MS-SSIM ↑ ($\times 10^{-2}$) | PSNR ↑ | MSE ↓ | FID ↓ |
| SD2 (Rombach et al., 2022) | 88.28 | 22.91 | 15.12 | 59.44 | 94.65 | 25.31 | 7.15 | 25.76 |
| SDXL (Podell et al., 2023) | 90.26 | 23.60 | 11.00 | 60.26 | 95.46 | 26.64 | 5.34 | 24.81 |
| FluxFill (Labs, 2024) | 89.51 | 24.65 | 8.29 | 62.84 | 94.69 | 25.54 | 5.63 | 27.07 |
| OmniText (Ours) | **95.71** | **29.52** | **3.44** | **39.06** | **98.21** | **33.90** | **1.61** | **15.33** |
| (Specialist) LaMa (Suvorov et al., 2022) | 93.93 | 29.37 | **2.91** | 43.67 | 97.40 | 32.80 | 1.62 | 15.83 |

Table 20: **Quantitative comparison against alternative methods for content control on ScenePair (Zeng et al., 2024).** Our classification loss outperforms alternative approaches based on maximization and minimization losses in terms of rendering accuracy, the primary objective of content control.

| | Text Editing (ScenePair (Zeng et al., 2024)) | | | | | | |
|---|---|---|---|---|---|---|---|
| | Rendering Accuracy | | Style Fidelity | | | | Runtime (s) |
| | ACC (%) ↑ | NED ↑ | MSE ↓ ($\times 10^{-2}$) | MS-SSIM ↑ ($\times 10^{-2}$) | PSNR ↑ | FID ↓ | |
| Attend-and-Excite (Chefer et al., 2023) | 86.72 | 0.964 | 7.32 | 33.65 | 12.87 | 36.28 | 24.542 |
| Attention Refocusing (Phung et al., 2024) | 85.55 | 0.960 | **6.68** | **34.83** | **13.24** | **35.64** | 25.044 |
| Classification Loss (Ours) | **88.20** | **0.969** | 8.77 | 29.92 | 12.00 | 38.64 | 24.946 |

Compared to LaMa, our method outperforms the specialist for both settings. This is likely due to LaMa's tendency to introduce artifacts, as shown in Fig. 13 (row 2). The only exception is the MSE score in the "all text" removal setting, where OmniText is competitive.

### J.4.2 CONTENT CONTROL ALTERNATIVES

To perform content control, specifically through latent optimization, we compare our proposed method, which uses a classification loss, with other maximization and minimization losses, including Attend-and-Excite (Chefer et al., 2023) and Attention Refocusing (Phung et al., 2024). As shown in Table 20, our method outperforms the other alternatives in terms of rendering accuracy, the primary goal of content control. This is supported by Fig. 21, which demonstrates that our classification loss provides better content control, reducing the tendency for duplicated letters across all cases (a-e). Notably, when we optimize using classification loss, the optimization successfully generates the necessary spacing, as seen in Fig. 21 (c).

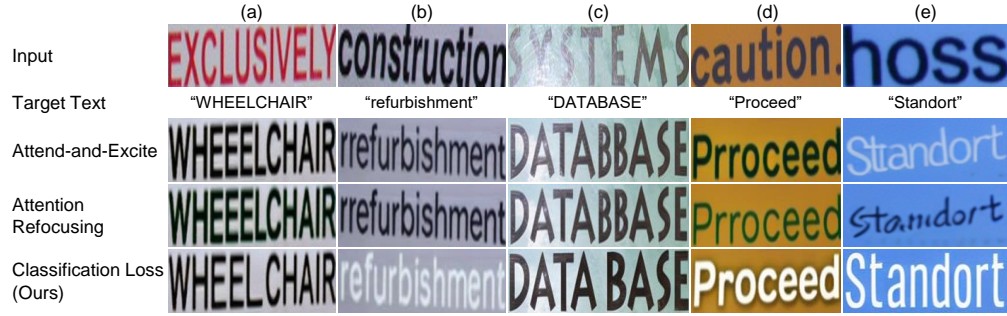

Figure 21: **Qualitative comparison of our classification loss against alternative loss functions.** Our method consistently achieves more accurate text rendering, the primary goal of content control, compared to other loss alternatives.

Table 21: **Quantitative comparison against alternative methods for style control on ScenePair (Zeng et al., 2024).** Our proposed Grid Trick ($G$) combined with style loss consistently outperforms other alternatives across all style fidelity metrics. In addition, the proposed method also achieves the best performance in terms of NED.

| | | | | Text Editing (ScenePair (Zeng et al., 2024)) | | | | |
|---|---|---|---|---|---|---|---|---|
| | Rendering Accuracy | | Style Fidelity | | | | Runtime (s) | |
| | ACC (%) ↑ | NED ↑ | MSE ↓ (×10⁻²) | MS-SSIM ↑ (×10⁻²) | PSNR ↑ | FID ↓ | | |
| StyleID (Chung et al., 2024) | **80.23** | 0.944 | 5.60 | 34.76 | 14.00 | 36.03 | 7.485 | |
| Self-Attention Modulation | 75.16 | 0.947 | 5.54 | 37.48 | 14.24 | 34.56 | 22.700 | |
| Self-Attention Style Loss (Ours) | 78.28 | **0.949** | **4.78** | **39.71** | **14.86** | **31.71** | 68.279 | |

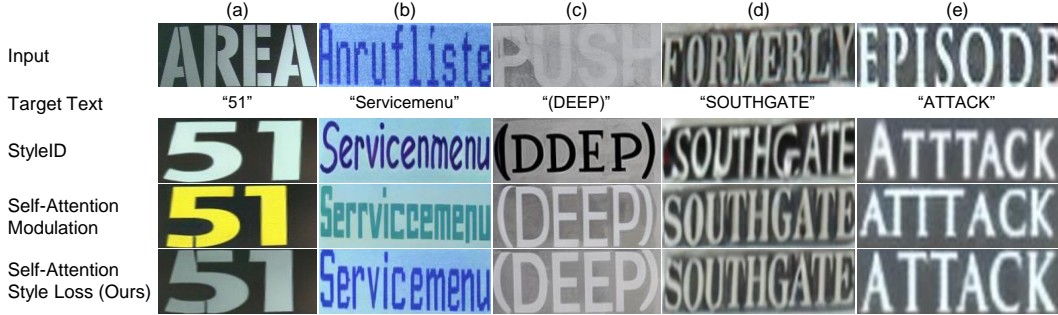

Figure 22: **Qualitative comparison against alternative methods for style control.** Our proposed method consistently preserves and accurately transfers the style of the input text, while other alternatives fail to replicate the style effectively and produce noticeable artifacts in challenging cases (d) and (e).

For each control mechanism, such as content control in this case, the goal is to choose the one that most effectively improves that specific aspect. In this scenario, our classification loss is the most appropriate option, even though it results in a reduction of style fidelity. The observed increase in rendering accuracy and the decrease in style fidelity across all methods highlight the inherent trade-off between rendering accuracy and style fidelity.

### J.4.3 STYLE CONTROL ALTERNATIVES

In this comparison, we evaluate our latent optimization with style loss against alternative methods, including the state-of-the-art diffusion-based style transfer method, StyleID (Chung et al., 2024), and self-attention modulation, where we modulate the self-attention to focus on the text reference within the grid input. As shown in Table 21, our latent optimization with the proposed style loss $\mathcal{L}_S$ outperforms the other alternatives in terms of style fidelity. This is further supported by Fig. 22 (a-e), where our method better copies the style of the input text compared to the other alternatives, even in challenging cases where the input has significant effects or uses an uncommon font. Our self-attention modulation also performs better in terms of style fidelity compared to StyleID (Chung et al., 2024), demonstrating the benefits of the Grid Trick and the use of self-attention manipulation within this framework.

Interestingly, in contrast to the results for content control, there is less trade-off between rendering accuracy and style fidelity in style control. As shown in Table 21, our method outperforms the other methods in terms of Normalized Edit Distance (NED), despite falling behind StyleID in word accuracy. This highlights the more complex relationship when optimizing for style fidelity, as also shown in Table 16, where adjusting the weighting factor $\lambda_S$ for our style loss can also improve accuracy.

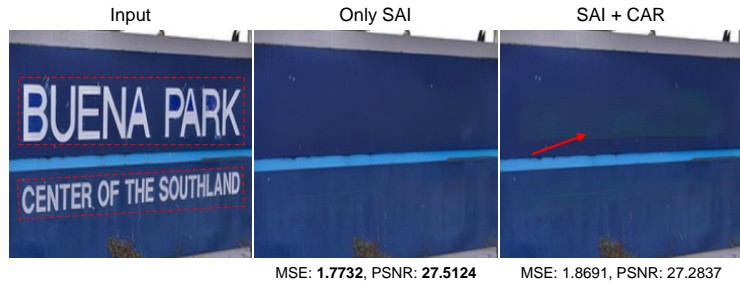

Figure 23: **Adding CAR when SAI alone is sufficient tends to introduce color shifting artifacts, which in turn degrades performance in terms of MSE and PSNR.**

Table 22: **Mean and standard deviation of text removal experiments on SCUT-EnsText (Liu et al., 2020) across 6 runs with different random seeds.** The standard deviation across all metrics is relatively low, indicating that our OmniText produces stable results regardless of the random seed.

| Text Removal (SCUT-EnsText (Liu et al., 2020)) | | | | |
|---|---|---|---|---|
| Metrics
Methods | All Text | | | |
| | MSE $\downarrow (\times 10^{-3})$ | MS-SSIM $\uparrow (\times 10^{-2})$ | PSNR $\uparrow$ | FID $\downarrow$ |
| OmniText (Reported) | 3.44 | 95.71 | 29.52 | 39.06 |
| OmniText (Mean ± Std) | 3.5764 ± 0.0968 | 95.7250 ± 0.0288 | 29.5230 ± 0.0673 | 39.7967 ± 0.5436 |
| Metrics
Methods | Largest Text | | | |
| | MSE $\downarrow (\times 10^{-3})$ | MS-SSIM $\uparrow (\times 10^{-2})$ | PSNR $\uparrow$ | FID $\downarrow$ |
| OmniText (Reported) | 1.6148 | 98.2100 | 33.8967 | 15.3331 |
| OmniText (Mean ± Std) | 1.5364 ± 0.0679 | 98.2400 ± 0.0210 | 34.1013 ± 0.1210 | 15.1820 ± 0.1299 |

### J.5 Text Removal - Cross Attention Reassignment (CAR) on Top of Self-Attention Inversion (SAI) on SAI's Successful Case

As shown in Fig. 23 and Fig. 19, adding CAR can introduce minor color shifts when SAI alone is sufficient, leading to a drop in MSE and PSNR scores. This is also reflected in Table 15, particularly in the "all text" removal scenario.

### J.6 Error Bars

In this section, we conduct additional experiments for text removal using the SCUT-EnsText (Liu et al., 2020) dataset and text editing with the ScenePair (Zeng et al., 2024) dataset. We report the results presented in the main paper, along with the mean and standard deviation. Due to limited computing resources, we only run multiple experiments for our method to assess the uncertainty of our results. Specifically, we run five additional experiments for text removal using different random seeds. Additionally, we run two more experiments for text editing, also with different random seeds.

#### J.6.1 Error Bar for Removal

As shown in Table 22, the standard deviation of the text removal experiments is relatively low for each metric. This indicates that OmniText tends to produce stable results regardless of the initial random seed. Additionally, the reported values in the main paper are generally lower than the mean, except for the FID in the "all text" removal scenario, where the difference is not significant.

#### J.6.2 Error Bar for Editing

As shown in Table 23, the standard deviation of the text editing experiments is relatively low across each metric. This indicates that OmniText tends to produce stable results regardless of the initial random seed. The only metric with a notably larger difference is ACC (Word Accuracy), while the

Table 23: **Mean and standard deviation of text editing experiments on ScenePair (Zeng et al., 2024) across 3 runs with different random seeds.** The standard deviation across all metrics is relatively low, indicating that our OmniText produces stable results regardless of the random seed.

| | Text Editing (ScenePair (Zeng et al., 2024)) | | | | | |
|---|---|---|---|---|---|---|
| | Rendering Accuracy | | Style Fidelity | | | |
| | ACC (%) ↑ | NED ↑ | MSE ↓ ($\times 10^{-2}$) | MS-SSIM ↑ ($\times 10^{-2}$) | PSNR ↑ | FID ↓ |
| OmniText (Reported) | 78.4375 | 0.9510 | 4.7900 | 40.1100 | 14.8469 | 31.6936 |
| OmniText (Mean ± Std) | 78.9584 ± 0.4511 | 0.9509 ± 0.0002 | 4.7933 ± 0.0252 | 39.8200 ± 0.2762 | 14.8295 ± 0.0154 | 31.6975 ± 0.1652 |

NED (Normalized Edit Distance) remains quite stable. This suggests that word accuracy is more sensitive to the initial random seed (see Section O for more discussion).

# K  ADDITIONAL DISCUSSION ON THE TRADE-OFF BETWEEN RENDERING ACCURACY AND STYLE FIDELITY

The trade-off between rendering accuracy (or content fidelity) and style fidelity (shown in Table 5) is not inherent to our method. Rather, this trade-off between content and style fidelity has been extensively studied in the image style transfer literature. However, to the best of our knowledge, our work is the first to explicitly demonstrate this trade-off in the *text domain*.

In image style transfer, it has been shown that content and style cannot be completely disentangled (Gatys et al., 2016; Huang & Belongie, 2017; Park & Lee, 2019), and thus a method cannot simultaneously achieve both content and style fidelity. This trade-off remains evident even in recent style transfer methods (Chung et al., 2024). In our case of text generation, this trade-off currently cannot be fully eliminated. In practice, the best we can do is to substantially improve style fidelity while minimizing the loss in rendering accuracy. Uniquely to our problem, one potential way to mitigate this issue is by adjusting the mask to follow the style of the input. For example, if the input font is larger, a larger mask is required. Conversely, a smaller font requires a smaller mask. The mask also needs to be adaptive based on the number of characters. With such mask adjustments, the rendering accuracy can be improved and the trade-off can be alleviated. However, designing an adaptive masking strategy that jointly accounts for both the input style and the number of characters remains an open problem, which we leave for future work. One possible direction is to train a model based on ViT (Dosovitskiy, 2020) or the Segment Anything Model (Kirillov et al., 2023) on a dataset constructed from SynthText (Gupta et al., 2016), which can render text on full images. Such a model would then be tasked with predicting appropriate mask given the target text and the input image.

# L  ADDITIONAL DISCUSSION ON THE INTERACTION BETWEEN TEXT REMOVAL AND CONTROLLABLE INPAINTING COMPONENTS

Note that our text removal components cannot be applied to controllable inpainting, and vice versa. Moreover, these two components cannot be combined into a single process. As illustrated in Fig. 24, this limitation arises from two main problems. First, these two components have conflicting objectives: the goal of text removal is to *remove* text, whereas the goal of controllable inpainting is to *generate* text under content and style constraints, in order to improve both rendering accuracy and style fidelity. Because of these conflicting objectives, jointly applying both components would be contradictory, and using the removal components during controllable inpainting would in fact degrade the generation performance.

Second, beyond this conceptual difference, these two components also use different operation that conflict each other. The text removal components *modify* cross-attention and self-attention values during sampling, while latent optimization for controllable inpainting *extracts* cross-attention and self-attention values during the optimization loop. If we were to perform latent optimization while simultaneously modifying these attention values on-the-fly, it would lead to inconsistencies and backpropagation errors, where the results becomes *NaN*. On the positive side, both components are modular and can be turned on or off independently, allowing users to select either text removal or controllable inpainting as needed.

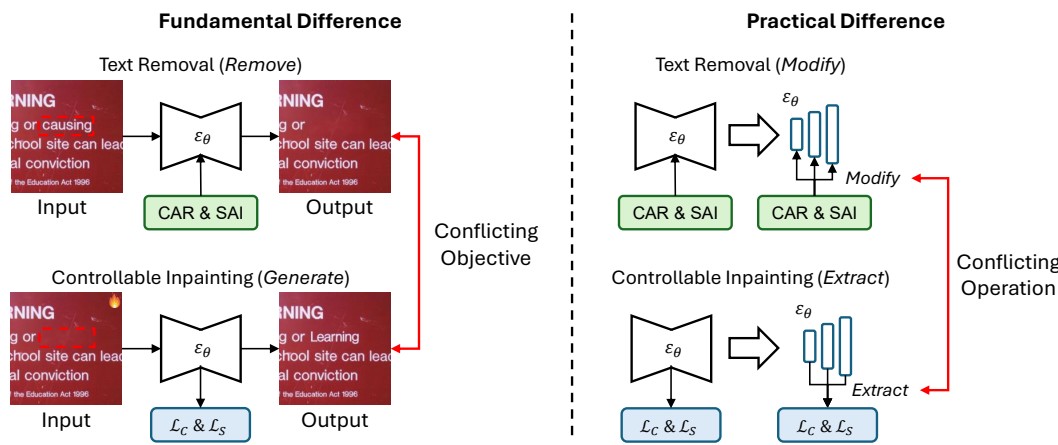

Figure 24: **Illustration of two main problems preventing the combination between text removal and controllable inpainting components.**

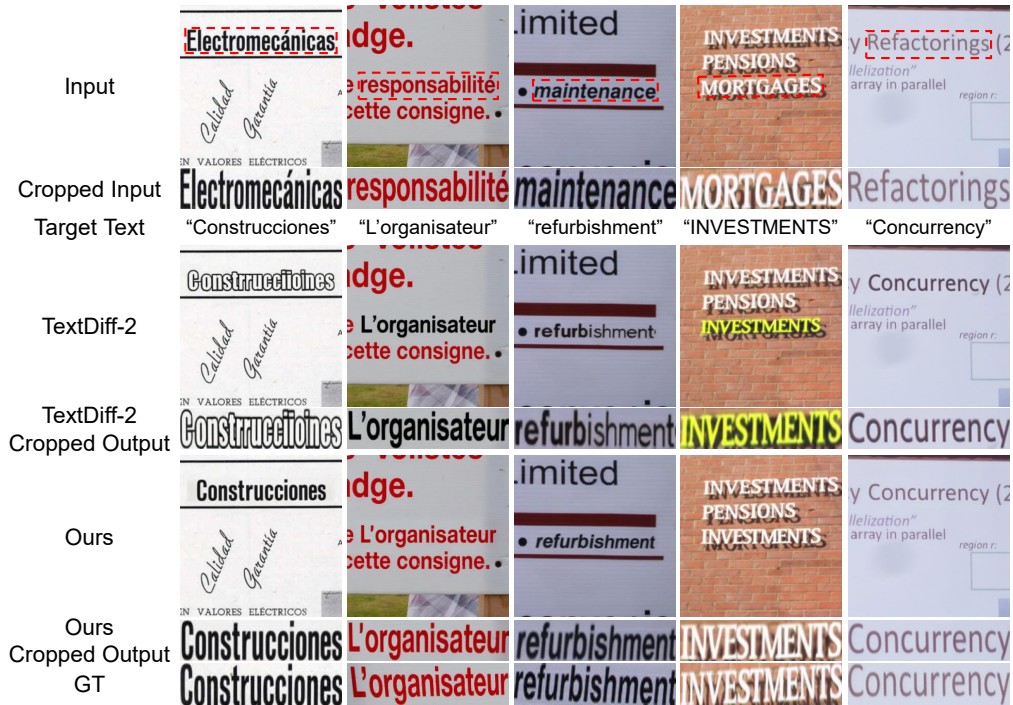

Figure 25: **Qualitative results on long-text manipulation.** We present qualitative results of OmniText compared to the backbone used in our method (TextDiff-2 (Chen et al., 2024)).

## M ADDITIONAL RESULTS ON CHALLENGING CASES

We provide additional qualitative results on challenging cases using the ScenePair dataset (Zeng et al., 2024) and copyright-free real-world images from the Internet. We analyze three main scenarios: (i) long-text manipulation, (ii) text manipulation on complex textures, and (iii) low-resolution input images.

**Long-Text Manipulation.** As shown in Fig. 25, OmniText can handle challenging settings where the text contains more than 10 characters. In the first column, our method provides better content and style control, improving both rendering accuracy and style fidelity, while in the other columns our method mainly improves style fidelity. Note that for longer text at the sentence or paragraph level

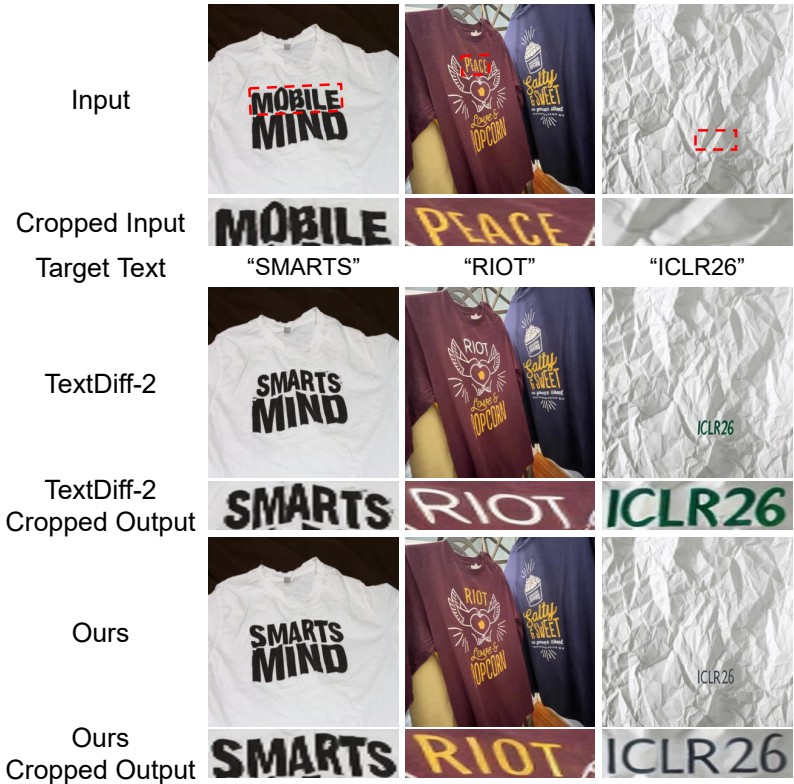

Figure 26: **Qualitative results on complex texture surfaces.** We present qualitative results of OmniText compared to the backbone used in our method (TextDiff-2 (Chen et al., 2024)).

(long, multi-line text), OmniText, as well as other text generation and editing methods (Wang et al., 2024; Fang et al., 2025; Zeng et al., 2024; Ye et al., 2024; Lan et al., 2025), does not support such cases, which are outside the scope of this work. This limitation arises because current training data for text generation are mostly at the word level or short phrase level (fewer than 3 words), which suggests an interesting direction for future work. One possible way to address longer text is to split the text into individual words and process them sequentially. A concurrent work (Wang et al., 2025a) attempts to address long-text generation, but does not support or analyze the broader set of TIM tasks that are central to our work.

**Text Manipulation on Complex Textures.** We also show qualitative results of our method when inserting or editing text on complex surfaces (e.g., curved surfaces). As shown in Fig. 26, both our backbone (TextDiff-2) and OmniText can handle natural wrinkles on clothing, as illustrated in the first column. However, TextDiff-2 tends to produce slight artifacts around the text boundaries, whereas our method produces a cleaner output. Both methods follow the curvature of the surface, and the characters "MAR" exhibit a wavy texture consistent with the underlying wrinkles.

In the second column, both our method and TextDiff-2 follow the surface curvature, particularly for the characters "OT," which exhibit noticeable depth. Our method achieves better style fidelity and curvature (for example, the orientation of "E" in the input and "T" in the output, which are similar). Lastly, for an unnatural complex surface such as crumpled paper, our method can partially follow the curvature with a more natural text style, while TextDiff-2 tends to simply overlay the text on top of the surface with an unnatural color. However, in this complex case, the fine-grained texture is not fully captured. This is likely due to limitations of the 3D prior in the backbone because its training data mainly consist of books and simpler textures such as shirts, and do not include highly complex textured surfaces like crumpled paper.

**Low-Resolution Input Images.** Lastly, we compare our method and the backbone in low-resolution settings. Many inputs from the ScenePair dataset (Zeng et al., 2024) are low-resolution images that

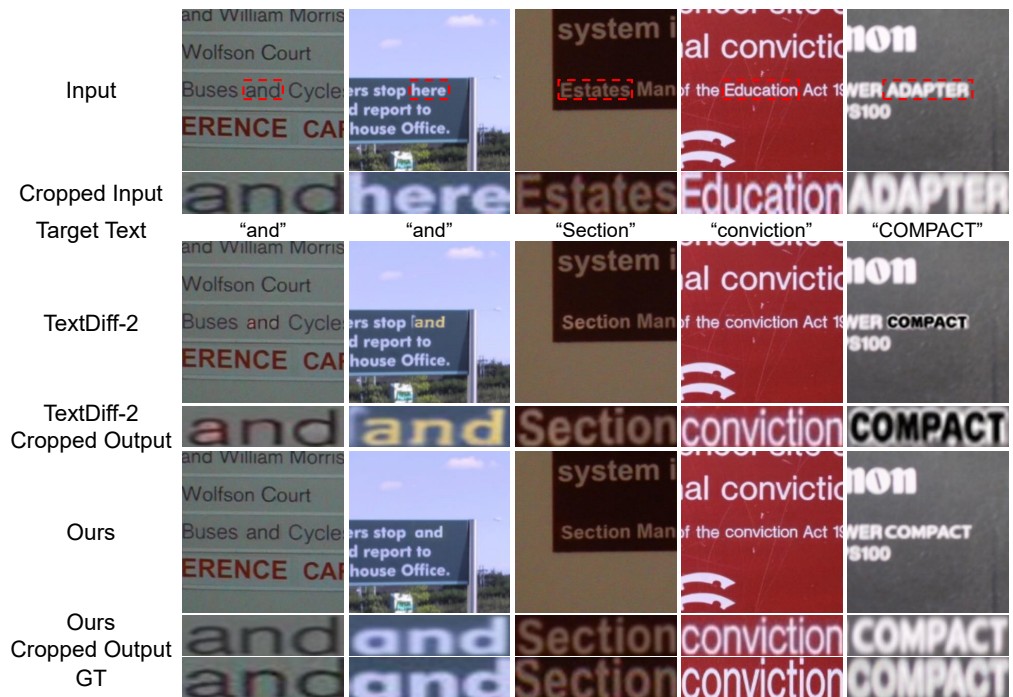

Figure 27: **Qualitative results on low-resolution input.** We present qualitative results of OmniText compared to the backbone used in our method (TextDiff-2 (Chen et al., 2024)).

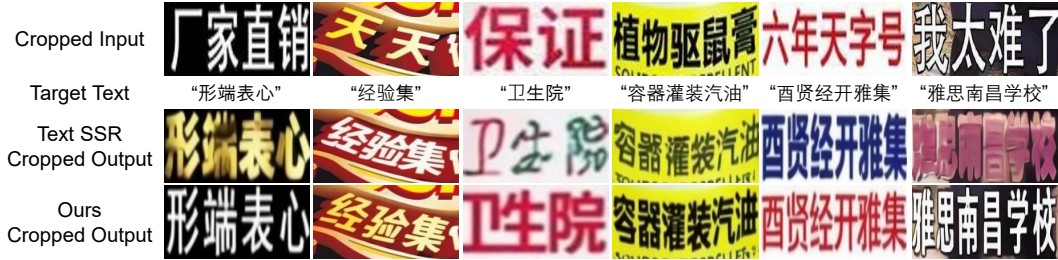

Figure 28: **Qualitative results of our method applied to the TextSSR backbone on Chinese text using the Wukong set from AnyText-benchmark (Tuo et al., 2023).**

we resize to $512 \times 512$, which introduces typical artifacts such as blur, noise, JPEG compression artifacts, and resizing artifacts. Despite this, both our method and the backbone consistently generate accurate text.

However, our method produces text whose style is more consistent with the input, whereas TextDiff-2 often generates text with different colors and fonts (last column), altered colors (second and third columns), and visible artifacts (first column), with only one clearly successful case in the fourth column, since many similar text instances appear around the target region. Therefore, our method is robust against low-resolution input images. This robustness is partly due to the backbone's training data, which already include text images of various resolutions that are preprocessed to $512 \times 512$.

# N    ADDITIONAL RESULTS ON NON ALPHANUMERIC CHARACTERS

Recently, one backbone has modified the latent diffusion model by removing the text encoder and replacing it with a glyph-based encoder, training it on both Latin and Chinese characters. Despite the removal of the text encoder (which prevents us from applying our cross-attention content loss), our method, including the grid trick and latent optimization with self-attention style loss, can still be

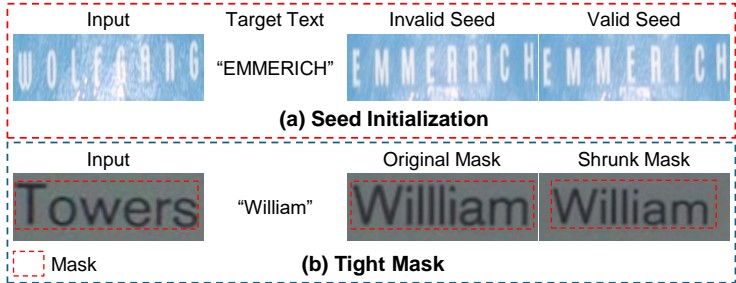

Figure 29: **Failure cases of our OmniText.** We present two failure cases of OmniText: (a) results from the inherent limitations of training-free diffusion methods, and (b) stems from the use of TextDiff-2 as the backbone. Both cases can be mitigated through user intervention, e.g., by running the model with a different random seed in (a), or by shrinking the input mask in (b).

applied to the TextSSR backbone (Ye et al., 2024). Specifically, we extract self-attention probabilities from Decoder Block 3, Layer 0, which encodes style information in the TextSSR architecture and use it for our self-attention style loss. We use 20 sampling steps and perform latent optimization for 20 iterations at each optimization step, applying optimization at timesteps 601, 551, and 501. Note that each backbone has its own unique properties, such as which self-attention layers encode style and at which timesteps the style can be modified with minimal artifacts.

As shown in Fig. 28, our method generalizes well to Chinese characters, with our outputs consistently achieving the best style fidelity across all examples, while the backbone hallucinates the text style. The improved styles also lead to better rendering accuracy, for instance, in the first column the character is more clearly visible, in the third column the strokes are sharper, and in the last column the TextSSR backbone produces hard-to-read text whereas our method significantly improves readability. Despite these strong results, minor failure cases remain, such as a single missing stroke in the second column.

Overall, these qualitative results demonstrate that our method generalizes to non-alphanumeric characters. Moreover, the concept of style encoded in self-attention does not depend on the language of the text, making our style-control mechanism broadly applicable to text generation in general regardless of the language.

## O    DISCUSSION OF LIMITATIONS

Our limitations can be categorized into two main points:

**Seed Initialization Validity.** Similar to other training-free methods that utilize latent optimization (e.g., Attend-and-Excite (Chefer et al., 2023)), the initial latent can impact the performance of our OmniText. As shown in Fig. 29 (a), using an invalid initial latent may reduce the accuracy of our method. This issue can be mitigated by running the method with different initial latents. However, this is a case-by-case issue, as the results from running with different random seeds across the full dataset tend to have low standard deviation (see Table 23), demonstrating performance stability. This issue was first explored in InitNO (Guo et al., 2024). We attempted to integrate their method to address this problem, but it did not show any improvement. The lack of improvement is likely due to the nature of the task, as InitNO is focused on image generation, while our task is primarily text editing. InitNO is designed for initial latent optimization at the object level, whereas our task requires more fine-grained, text-level manipulation, limiting the applicability of InitNO in our context. This highlights an opportunity for future work to explore the influence of the initial latent space in text editing tasks more deeply.

**Rendering Accuracy Depends on Mask Tightness.** As shown in Fig. 29 (b), the performance of OmniText depends on the backbone due to the nature of training-free method. Therefore, the limitations of the backbone are inherited by OmniText. In this case, our backbone, TextDiff-2 (Chen et al., 2024), tends to produce duplicated letters when given a mask larger than the target text. This issue arises because the backbone is not trained to account for spacing between letters in such cases.

Although we attempt to mitigate this by using our content control, the simultaneous goal of performing style control means that the problem of duplicated letters cannot be entirely eliminated. Furthermore, when optimizing for style, the spacing between letters is copied, which exacerbates the duplicated letter issue.

This limitation can be alleviated through additional fine-tuning strategies for the backbone, specifically by training it to handle cases where spacing needs to be added. Another potential solution is to involve users in the process by allowing them to reduce the target mask, which is a simple and effective way to alleviate the issue, as demonstrated in Fig. 29 (b).

Besides these two limitations of our method, similar to other recent text editing works (Wang et al., 2024; Fang et al., 2025; Zeng et al., 2024), our OmniText does not support other languages that use non-alphanumeric characters. However, note that this is because our OmniText depends on the backbone's capabilities. To extend our method to other languages, we can fine-tune the backbone for the additional languages we want to support. In addition, like other recent text editing works (Wang et al., 2024; Fang et al., 2025; Zeng et al., 2024; Ye et al., 2024; Lan et al., 2025), our OmniText does not support long, multi-line text, which falls outside the scope of this work. All models presented in this work are optimized for short text rendering. Supporting long text would require splitting the text into individual words and processing them sequentially. We also note that while a concurrent work (Wang et al., 2025a) addresses long-text generation, it does not support or analyze other TIM tasks, which are central to our work.

