# OpenReview forum: "OmniText: A Training-Free Generalist for Controllable Text-Image Manipulation"
_ICLR.cc/2026/Conference — ICLR 2026 Poster_

### Official Review · Reviewer_8xFf · 2025-10-24

**Soundness:** 3
**Presentation:** 3
**Contribution:** 3
**Rating:** 6
**Confidence:** 4

**Summary:**

This paper proposes OmniText, a training-free generalist framework for Text Image Manipulation (TIM) tasks. The authors identify three key limitations in existing diffusion-based text inpainting methods: inability to remove text, lack of style control, and letter duplication issues. To address these challenges, OmniText leverages self-attention inversion to enable text removal by reducing text hallucinations, and redistributes cross-attention to control text content. For controllable text inpainting, the method introduces a cross-attention content loss and a self-attention style loss within a latent optimization framework to improve rendering accuracy and style customization. Additionally, the authors contribute OmniText-Bench, a benchmark dataset for evaluating diverse TIM tasks including text removal, rescaling, repositioning, and styled insertion/editing. The proposed method achieves state-of-the-art performance compared to other text inpainting approaches and demonstrates comparable results to specialist methods while maintaining generalist capabilities across multiple TIM tasks.

**Strengths:**

1. The paper addresses critical practical limitations in text image manipulation that have hindered the deployment of existing methods. The ability to remove text, control styles, and avoid letter duplication are essential capabilities for real-world applications, making this work highly relevant and impactful for the community.

2. Unlike previous approaches that rely on data-driven multi-task training, OmniText demonstrates strong innovation by exploiting the internal attention mechanisms of diffusion models. The self-attention inversion for text removal and cross-attention redistribution for content control represent elegant, principled solutions that leverage the model's intrinsic properties rather than brute-force data scaling.

3. OmniText-Bench provides a comprehensive evaluation framework covering diverse TIM tasks with input images, target text, masks, and style references. This benchmark will significantly benefit the community by enabling standardized evaluation and facilitating future research in this domain.

**Weaknesses:**

1.  As a plug-and-play method, the experiments are conducted solely on TextDiff-2. This raises concerns about the generalizability of the approach—does the mechanism heavily depend on the specific capabilities of the base model? Validation on other diffusion-based text generation models (e.g., different architectures or model scales) is necessary to demonstrate the method's robustness and broader applicability.

2. The paper lacks critical analysis of computational efficiency. Key questions remain unanswered: How much does the proposed framework increase inference time? What is the overhead of the latent optimization process and attention manipulations? For a practical method, understanding the efficiency-performance trade-off is essential.

3. The ablation experiments are fragmented—SAI and CAR modules are only ablated for text removal tasks, while the loss functions are only evaluated for editing tasks. The potential interactions between these components (e.g., how SAI/CAR affect controllable inpainting, or how the loss functions impact text removal) are not explored. A comprehensive ablation study examining all components across all tasks would provide better insights into each module's contribution and their synergistic effects.

**Questions:**

1. Trade-off in Table 5: The ablation results show different loss configurations exhibit trade-offs between style fidelity and rendering accuracy. Can the authors explain this phenomenon? Is this trade-off inherent to the method or can it be mitigated?

2. Multilingual Generalization: The results primarily focus on English text. Can the authors explore whether OmniText works effectively for Chinese or other non-Latin scripts? Are any modifications needed to handle different writing systems?

---

> ### Author Response · Authors · 2025-11-21
> **(1/2)**
>
> We thank the reviewer for the constructive comments and insightful questions.
> To address the reviewer's concerns, we have updated the Appendix.
> Please refer to the updated Appendix attached to the main paper for the changes (highlighted in red text or red captions).
> We categorize and respond to the weaknesses and questions below.
>
> ## Response to Weaknesses
>
> ### Response to Weakness: Generalization of the OmniText Framework
> Our OmniText framework can be applied to **any U-Net–based backbone that resembles a vanilla latent diffusion model**, similar to prior works that use latent optimization [1, 2], which demonstrate generalization across U-Net–based backbones.
> We explore the generalization of our method on different backbones, i.e., TextDiff-2, UDiffText, DreamText, TextSSR, shown in **Section D.2. of Appendix**, that shows the method does **not** depend on any special capability of TextDiff-2, since this backbone is essentially a vanilla latent diffusion model trained on text generation, with minor changes to the text encoder to enable character-level tokenization.
> However, to date, TextDiff-2 is the **only** backbone that closely follows the vanilla latent diffusion design.
> Other methods (e.g., UDiffText, DreamText, TextSSR) introduce more specialized architectures, thus we modify our method to adjust to these backbones.
>
> Regarding **text removal**, our module can in principle be applied to any backbone as well.
> However, effective removal requires a **removal prior**, which exists only in backbones trained on datasets that include **removal cases** (e.g., MARIO-10M).
> Many recent backbones are trained on datasets focused solely on text generation and contain **no removal cases**, so they lack the necessary prior for our removal module to function properly.
>
> ### Response to Weakness: Computational Efficiency
>
> We have updated the Appendix to include comprehensive runtime experiments in **Section I**.
> For all measurements, we use a **single RTX A6000 GPU** and run each sample **15 times**, reporting the average runtime.
>
> For text removal, our OmniText runs competitively compared to other methods, where it requires 2.335 seconds (s) per sample (with the backbone taking 1.954 s), while other methods take between 0.5 s and 8.5 s, as summarized in the Table below.
>
> | Method | Runtime (s) |
> |---|---|
> | AnyText | 1.526 |
> | AnyText2 | 3.106 |
> | TextDiff-2 | 1.954 |
> | UDiffText | 8.257 |
> | DreamText | 5.987 |
> | OmniText (Ours) | 69.461 |
> | (Specialist) LaMa | 1.396 |
> | (Specialist) ViTEraser | 0.061 |
>
> Meanwhile, for other tasks, all baselines take less than 10 seconds per sample, whereas our OmniText takes 69.461 seconds, as shown in the Table below.
>
> | Method | Runtime (s) |
> |---|---|
> | AnyText | 1.526 |
> | AnyText2 | 3.106 |
> | TextDiff-2 | 2.521 |
> | UDiffText | 8.257 |
> | DreamText | 5.987 |
> | OmniText (Ours) | 69.461 |
> | (Specialist) TextCtrl | 6.782 |
>
> Although our framework introduces non-trivial overhead due to per-image latent optimization and attention-based style control in controllable inpainting, it is, to the best of our knowledge, the **first effective approach that enables explicit style control for text image manipulation (TIM)** across a wide range of tasks.
>
> ### Response to Weakness: Interaction Between Text Removal and Controllable Inpainting Components
>
> We address this point by adding **Section L** in the Appendix, where we analyze the interaction (and incompatibility) between text removal (TR) and controllable inpainting (CI).
> In summary, there are two main reasons why we do **not** combine these components in a single pass:
> 1) **Conflicting objectives**: the goals of TR and CI are inherently opposite, where TR aims to suppress text while CI aims to generate it.
> 2) **Conflicting operations**: TR modifies the attention values, while CI extracts these values, which leads to backpropagation issues and results in **NaN losses with black outputs**.
> For these reasons, we treat text removal and controllable inpainting as **separate components** rather than jointly applying them in a single forward–backward loop.

---

> ### Author Response · Authors · 2025-11-21
> **(2/2)**
>
> ## Answers to Questions
>
> ### Q1. Trade-off Between Rendering Accuracy and Style Fidelity (Tab. 5)
>
> We address this question by adding a dedicated **Section K** in the Appendix, as this trade-off is central to our method.
> The trade-off between **rendering accuracy (content fidelity)** and **style fidelity** arises from the fundamental nature of **style transfer**, which has been extensively studied in the literature [3, 4, 5].
> Our work is, to the best of our knowledge, the **first to explicitly demonstrate this trade-off in the *text* image manipulation domain**.
>
> In our setting, one promising direction to **mitigate** this trade-off is to use **adaptive masks** that depend on both the input style (e.g., larger or smaller fonts) and the number of target characters.
> One way to do this is by manual adaptation of the mask as shown in **Sec. O in Appendix**.
> Another concrete approach for automatic adaptation is to **train a mask prediction model** that takes as input the target text length and the input style and predicts the **new text mask**.
>
> ### Q2. Multilingual Generalization
>
> Our method is **not limited to a specific language**.
> If the backbone supports **multilingual text generation**, OmniText can be directly applied **without any modification**.
> To empirically support this claim, we evaluate our method on **Chinese** using the **TextSSR** backbone, which is trained on both Latin and Chinese text.
> As shown in **Section N** and **Fig. 28** of the Appendix, our method **consistently improves style fidelity for Chinese**.
>
> ## Reference
>
> [1] Chefer et al. "Attend-and-excite: Attention-based semantic guidance for text-to-image diffusion models." ACM TOG 2023.
>
> [2] Phung et al. "Grounded text-to-image synthesis with attention refocusing." CVPR 2024.
>
> [3] Gatys et al. "Image style transfer using convolutional neural networks." CVPR 2016.
>
> [4] Huang and Serge. "Arbitrary style transfer in real-time with adaptive instance normalization." ICCV 2017.
>
> [5] Chung et al. "Style injection in diffusion: A training-free approach for adapting large-scale diffusion models for style transfer." CVPR 2024.

---

### Official Review · Reviewer_ziMs · 2025-10-31

**Soundness:** 3
**Presentation:** 3
**Contribution:** 3
**Rating:** 4
**Confidence:** 3

**Summary:**

This paper introduces OmniText, a training-free framework for various Text Image Manipulation (TIM) tasks, positioning it as the first generalist method for text removal, editing, insertion, and style-based manipulation. The method builds on a pre-trained model TextDiff-2 and introduces novel attention manipulation strategies and loss functions to control text rendering without fine-tuning. The authors also present OmniText-Bench, a new dataset for evaluating diverse TIM tasks.

**Strengths:**

- The paper addresses the important goal of creating a unified, generalist model for diverse TIM tasks. Its training-free approach, which avoids costly retraining by manipulating a pre-trained model's internal states, is a significant practical advantage.

- The introduction of the OmniText-Bench dataset is a valuable contribution that addresses a clear gap in evaluation resources for complex, style-oriented TIM tasks.

**Weaknesses:**

- While the specific techniques are new, the high-level approach of using attention control and latent optimization for editing is established in the broader image editing field.

- The paper completely omits any discussion of computational cost. Methods involving per-image optimization are often significantly slower at inference. A runtime comparison against baselines is critical for understanding the practical trade-offs of the proposed framework.

- The method's generalizability is not fully validated, as it is only demonstrated on the TextDiff-2 backbone. Furthermore, the new benchmark, while useful, is relatively small, which may limit the statistical significance of the evaluations performed on it.

**Questions:**

- Can you provide an analysis of OmniText's inference time compared to baseline methods, especially given the latent optimization step?

- How sensitive is the output quality to the choice of loss weights and the number of optimization steps during controllable inpainting?

-  Have you attempted to apply the OmniText modules to other text-synthesis backbones to demonstrate the generalizability of your approach?

---

> ### Author Response · Authors · 2025-11-21
> **(1/3)**
>
> We thank the reviewer for the constructive comments and insightful questions.
> To address the reviewer's concerns, we have updated the Appendix (red text or red caption).
> Please refer to the Appendix attached to the main paper for the updates.
> We categorize and respond to the weaknesses and questions below.
>
> ## Response to the Weaknesses
> ### Clarification on Similarity to Attention Control and Latent Optimization in Image Editing
>
> We acknowledge that our work is conceptually related to prior methods that use attention control and latent optimization in the broader image editing literature.
> However, **text generation relies on inpainting**, which has different characteristics from general image editing that uses unconditional or text-guided generation.
> Moreover, our **technical contributions are tailored specifically to text generation via inpainting**. For example, our content and style objectives use **focal loss** and **KL-divergence–based style loss**, respectively, designed for controllable inpainting that are different from previous works [1, 2].
> To the best of our knowledge, our work is also the **first to systematically analyze how attention-based controls can be used for style control in text generation**, addressing the fact that existing inpainting models for text generation initially lack any explicit style control.
> In addition, we **extensively validate our method to diverse tasks of Text Image Manipulation (TIM)** for the first time, signifying our main contribution of both the methods and experiments.
>
>
> ### Clarification on Runtime Comparison and Answer to Q1 (Inference Time Given the Latent Optimization Step)
>
> In **Section I** of Appendix, we provide a detailed runtime comparison of different components of our method and a comparison against other approaches.
> For all measurements, we use a **single RTX A6000 GPU** and run each sample **15 times**, reporting the average runtime.
>
> For text removal, our OmniText runs competitively compared to other methods, where it requires 2.335 seconds (s) per sample (with the backbone taking 1.954 s), while other methods take between 0.5 s and 8.5 s, as summarized in the Table below.
>
> | Method | Runtime (s) |
> |---|---|
> | AnyText | 1.526 |
> | AnyText2 | 3.106 |
> | TextDiff-2 | 1.954 |
> | UDiffText | 8.257 |
> | DreamText | 5.987 |
> | OmniText (Ours) | 69.461 |
> | (Specialist) LaMa | 1.396 |
> | (Specialist) ViTEraser | 0.061 |
>
>
> Meanwhile, for other tasks, all baselines take less than 10 seconds per sample, whereas our OmniText takes 69.461 seconds, as shown in the Table below.
>
>
> | Method | Runtime (s) |
> |---|---|
> | AnyText | 1.526 |
> | AnyText2 | 3.106 |
> | TextDiff-2 | 2.521 |
> | UDiffText | 8.257 |
> | DreamText | 5.987 |
> | OmniText (Ours) | 69.461 |
> | (Specialist) TextCtrl | 6.782 |
>
>
> We acknowledge that **our full controllable inpainting pipeline is slower at inference**, which is a limitation of all training-free methods that use latent optimization including our approach.
> However, we emphasize that this additional cost yields **strong and controllable content and style fidelity**.
> To the best of our knowledge, our method is *the first to provide explicit style control at inference time* for text image manipulation.
> Note that, other baselines cannot perform style control that is important to these tasks and to date there is no effective baseline that can run effective style control.
> We further study the effect of the **number of latent optimization steps** in **Tab. 18** of the Appendix and the associated discussion in **Lines 1429–1441**.

---

> ### Author Response · Authors · 2025-11-21
> **(2/3)**
>
> ### Regarding Generalization of Our Method and Answer to Q3 (Generalization of OmniText's Modules)
>
> We address the concern about generalization both in the main paper and in the updated Appendix:
> (1) We have provided the application of our self-attention modulation to **UDiffText** that leads to improved style fidelity in **main paper (Lines 431-433)**.
> (2) In addition, we add **Section D.2** to further analyze **generalization to other U-Net-based backbones**, including DreamText and TextSSR, where we need to modify some parts of the method.
> As shown in Table below, the style consistency of both UDiffText and DreamText improves across all tasks in OmniText-Bench.
>
> Text Insertion:
>
> | Method | ACC (%) ↑ | NED ↑ | MSE ↓ ($×10^{-2}$) | MS-SSIM ↑ ($×10^{-2}$) | PSNR ↑ | FID ↓ |
> |---|---|---|---|---|---|---|
> | DreamText | **85.33** | **0.965** | 6.44 | 36.23 | 13.18 | **54.48** |
> | DreamText + Mod (Ours) | 84.00 | 0.946 | **5.65** | **38.05** | **13.96** | 54.87 |
> | UDiffText | **94.67** | **0.985** | 7.94 | 34.83 | 12.03 | 62.00 |
> | UDiffText + Mod (Ours) | 89.33 | 0.975 | **6.74** | **36.68** | **13.08** | **58.03** |
>
>
> Style-based Text Insertion:
>
> | Method | ACC (%) ↑ | NED ↑ | MSE ↓ ($×10^{-2}$) | MS-SSIM ↑ ($×10^{-2}$) | PSNR ↑ | FID ↓ |
> |---|---|---|---|---|---|---|
> | DreamText | **78.67** | **0.941** | 10.06 | 24.11 | 10.67 | **79.71** |
> | DreamText + Mod (Ours) | 74.67 | 0.929 | **7.75** | **28.66** | **12.19** | 94.00 |
> | UDiffText | **88.00** | **0.973** | 11.79 | 24.43 | 9.98 | 82.21 |
> | UDiffText + Mod (Ours) | 86.67 | 0.968 | **9.98** | **26.64** | **10.80** | **81.21**\\
>
> Text Editing:
>
> | Method | ACC (%) ↑ | NED ↑ | MSE ↓ ($×10^{-2}$) | MS-SSIM ↑ ($×10^{-2}$) | PSNR ↑ | FID ↓ |
> |---|---|---|---|---|---|---|
> | DreamText | **84.00** | **0.954** | 7.99 | 34.17 | 12.23 | 65.83 |
> | DreamText + Mod (Ours) | 80.67 | 0.946 | **5.28** | **40.11** | **14.26** | **50.27** |
> | UDiffText | **96.00** | **0.984** | 9.78 | 31.22 | 11.17 | 68.67 |
> | UDiffText + Mod (Ours) | 90.67 | 0.981 | **6.33** | **38.82** | **13.34** | **55.23** |
>
> Style-based Text Editing:
>
> | Method | ACC (%) ↑ | NED ↑ | MSE ↓ ($×10^{-2}$) | MS-SSIM ↑ ($×10^{-2}$) | PSNR ↑ | FID ↓ |
> |---|---|---|---|---|---|---|
> | DreamText | **79.33** | **0.950** | 9.98 | 24.78 | 10.63 | **86.09** |
> | DreamText + Mod (Ours) | 72.67 | 0.920 | **7.14** | **30.25** | **12.65** | 106.23 |
> | UDiffText | **94.67** | **0.986** | 11.97 | 24.96 | 9.86 | 87.70 |
> | UDiffText + Mod (Ours) | 81.33 | 0.964 | **10.22** | **26.32** | **10.58** | **84.78** |
>
>
> Text Rescaling:
>
> | Method | ACC (%) ↑ | NED ↑ | MSE ↓ ($×10^{-2}$) | MS-SSIM ↑ ($×10^{-2}$) | PSNR ↑ | FID ↓ |
> |---|---|---|---|---|---|---|
> | DreamText | **72.00** | **0.931** | 9.22 | 26.82 | 11.32 | 72.84 |
> | DreamText + Mod (Ours) | 45.33 | 0.887 | **6.45** | **29.04** | **13.02** | **65.55** |
> | UDiffText | **88.67** | **0.978** | 11.11 | **28.56** | 10.28 | 76.51 |
> | UDiffText + Mod (Ours) | 56.00 | 0.918 | **7.98** | 28.30 | **11.94** | **62.69** |
>
> Text Repositioning:
>
> | Method | ACC (%) ↑ | NED ↑ | MSE ↓ ($×10^{-2}$) | MS-SSIM ↑ ($×10^{-2}$) | PSNR ↑ | FID ↓ |
> |---|---|---|---|---|---|---|
> | DreamText | **83.33** | **0.958** | 7.87 | 32.44 | 12.03 | 69.15 |
> | DreamText + Mod (Ours) | 82.67 | 0.956 | **5.22** | **39.67** | **14.15** | **54.61** |
> | UDiffText | **92.00** | **0.985** | 10.35 | 30.15 | 10.84 | 74.13 |
> | UDiffText + Mod (Ours) | 88.67 | 0.978 | **6.49** | **38.89** | **13.11** | **58.51** |
>
> Our method also consistently improves TextSSR style fidelity shown in qualitative results of **Fig. 28**.
> The generalization to TextSSR also shows that our method can work on other languages.
>
> Meanwhile, for our text removal, it can also be applied to any U-Net-based backbone.
> However, to be effective it requires a **removal prior**, which is present only in backbones trained on datasets with **robust text removal cases**, such as MARIO-10M.
> Many recent backbones are trained on datasets with purely text generation tasks and **no removal cases**, so they lack the necessary prior for our removal module to work effectively.

---

> ### Author Response · Authors · 2025-11-21
> **(3/3)**
>
> ### OmniText-Bench is Relatively Small
>
> We acknowledge the concern regarding the size of OmniText-Bench.
> To the best of our knowledge, there is currently **only one** publicly available dataset that provides **full-resolution images at $512 \times 512$** with **approximate** ground truth obtained via image retrieval for style fidelity evaluation, while other benchmarks (e.g., Tamper-Syn2k [3]) are **cropped synthetic datasets with varying resolution from $22 \times 50$ until $391 \times 3308$**.
>
> For comparison, ScenePair contains **1,280 images** with **approximate** ground truth, and Tamper-Syn2k contains **2000 images**, whereas OmniText-Bench (Section. E of Appendix) contains **150 images per task**, for a total of **1,050 images**, and provides **exact ground truth** (rather than approximate), explicit **style reference images**, and **masks** for the edited regions.
> These properties make OmniText-Bench **more suitable** for evaluating (testing) **style fidelity across a wide range of TIM tasks** with competitive number of samples.
>
> ## Answer to Questions
>
> ### Q2. Robustness to Loss Weights and Optimization Steps
>
> We refer the reviewer to **Tab. 16** and the new **Tab. 18** in the Appendix, where the ablation studies show that our method is **robust** to both the **loss-weight** and **optimization-iteration** hyperparameters.
> New **Tab. 18** shows that varying optimization iterations between 5-20 only affects the ACC for 1\%, NED for 0.002, MSE for 0.3, MS-SSIM for 0.9, PSNR for 0.4 and FID for 2, which are minor changes relative to each metric, as shown in Table below for the results of varying optimization iterations.
>
> | Method | ACC (%) ↑ | NED ↑ | MSE ↓ ($×10^{-2}$) | MS-SSIM ↑ ($×10^{-2}$) | PSNR ↑ | FID ↓ |
> |---|---|---|---|---|---|---|
> | 20 Iterations | 78.44 | 0.951 | **4.79** | **40.11** | **14.85** | **31.69** |
> | 15 Iterations | **79.30** | **0.953** | 4.81 | 40.09 | 14.83 | 32.54 |
> | 10 Iterations | 78.98 | **0.953** | 4.93 | 39.74 | 14.74 | 33.05 |
> | 5 Iterations | 78.36 | 0.952 | 5.08 | 39.29 | 14.61 | 33.49 |
>
>
> ## Reference
> [1] Chefer et al. "Attend-and-excite: Attention-based semantic guidance for text-to-image diffusion models." ACM TOG 2023.
>
> [2] Phung et al. "Grounded text-to-image synthesis with attention refocusing." CVPR 2024.
>
> [3] Qu et al. "Exploring stroke-level modifications for scene text editing." AAAI 2023.

---

### Official Review · Reviewer_Qwan · 2025-10-31

**Soundness:** 3
**Presentation:** 3
**Contribution:** 3
**Rating:** 8
**Confidence:** 4

**Summary:**

This paper addresses limitations of diffusion-based text inpainting in Text Image Manipulation (TIM): inability to remove text, lack of text style control, and duplicated letter generation. It proposes OmniText, a training-free generalist for diverse TIM tasks, leveraging cross- and self-attention mechanisms. Self-attention inversion enables text removal by reducing text hallucinations, while cross-attention redistribution lowers hallucinations via adjusted text token probability. For controllable inpainting, latent optimization uses cross-attention content loss (improves text accuracy) and self-attention style loss (enables style customization). The study also introduces OmniText-Bench, a benchmark with input images, target text/masks, and style references for tasks like text removal and stylized editing. Experiments show OmniText, the first TIM generalist, achieves SOTA across tasks vs. text inpainting methods and matches specialists.

**Strengths:**

1) This is the first training-free generalist method enabling text insertion and editing, removal, repositioning, and rescaling, and enabling explicit control over style fidelity, text content, and text removal, which is more applicable in real world.
2) The OmniText-Bench is proposed for a mockup-based evaluation. It consists of 150 sets of input images, targets texts with masks, reference images, and ground-truth, and covers five distinct applications.
3) The qualitative and quantitative experimental results show the comparable performance with specialist models.

**Weaknesses:**

1) How about comparing with recent large models such as GPT-4o, Gemini 2.5 Flash Image, Qwen Image since they are also generalist?

**Questions:**

See Weaknesses.

---

> ### Author Response · Authors · 2025-11-21
>
> We thank the reviewer for the thoughtful feedback and insightful comments.
> Please refer to the Appendix attached to the main paper for the updates (red text or red caption).
> Below, we address the reviewer's weakness and question regarding comparison with recent generalist models.
>
> ## Response to Weakness and Question: Comparison with Recent Large Generalist Models (GPT-4o, Gemini 2.5 Flash Image, Qwen Image)
> We appreciate the suggestion to compare our method with recent large generalist models.
> We have already compared (in Appendix section B) our method against GPT-4o, Gemini 2.5 Flash Image, and Qwen Image Edit (better variant of Qwen Image, developed to handle text) suggested by the Reviewer that can serve as generalists for text editing and have been reported to perform well on such tasks.
>
> For the GPT-4o and Gemini 2.5 Flash Image, we evaluate two settings: **with mask** and **without mask**.
> Running large-scale experiments with these models is challenging, as they exhibit unstable behavior and occasionally change the image entirely, which requires manual verification.
> To ensure a tractable evaluation, we therefore select **20 images** from the **ScenePair** dataset, covering diverse cases such as tilted vs. non-tilted text and various text styles.
> For these 20 images, we provide masks for **10 images**, while the **remaining 10** are edited **without masks**, since these models sometimes perform better without them.
>
> As shown quantitatively in **Tab. 6** and qualitatively in **Fig. 9** of the Appendix (where we have included these before rebuttal), two main issues emerge from these models:
> 1. **Failure on tilted and complex text styles.**
>    The models often fail to correctly edit **tilted text** or text with **intricate styles**, although the models can handle simpler, non-tilted cases.
> 2. **Insufficient spatial control and text overflow.**
>    When no mask is provided, the generated text frequently **overflows beyond the intended region**. Even when masks are used, the models still often fail to keep the text reliably within the target boundaries.
> These observations indicate that **precise text image manipulation** is not yet reliably achievable with these large generalist models.
> Due to this, our quantitative results in **Tab. 6** also show that both GPT-4o and Gemini 2.5 Flash Image achieve **rendering accuracy below 50\%**, with Gemini 2.5 Flash Image outperforming GPT-4o in both rendering accuracy and style fidelity.
> Notably, GPT-4o tends to shift the text position substantially even when masks are used, and in some cases alters the entire image, introducing unwanted color shifts and enhancements.
>
> In addition, we compare our method with QwenImageEdit.
> On the full set of ScenePair dataset, QwenImageEdit fails to generate accurate text and achieves a **rendering accuracy below 40\%** because it often does not change the text to match the given prompt.
> Moreover, its style fidelity is worse than that of our OmniText.

---

### Official Review · Reviewer_S1wy · 2025-11-01

**Soundness:** 3
**Presentation:** 2
**Contribution:** 3
**Rating:** 6
**Confidence:** 4

**Summary:**

This paper proposes OmniText, a training-free generalist framework for diverse Text-Image Manipulation (TIM) tasks. It addresses the limitations of existing methods—task specialization, lack of style control, and character duplication—by manipulating self- and cross-attention mechanisms. Specifically, it introduces Self-Attention Inversion (SAI) and Cross-Attention Reassignment (CAR) for text removal, and a latent optimization framework with cross-attention content loss and self-attention style loss for controllable inpainting. Additionally, the paper presents OmniText-Bench, a benchmark dataset covering multiple TIM tasks to fill the evaluation gap. OmniText aims to unify various TIM tasks while maintaining competitive performance compared to specialist and generalist baselines.

**Strengths:**

1. **Novel Generalist Design for TIM Tasks**: OmniText is the first training-free framework to jointly support diverse TIM tasks (removal, editing, insertion, rescaling, repositioning, style-based manipulation), addressing the long-standing task specialization issue in existing works. Its modular design (text removal + controllable inpainting) enables flexible adaptation to different tasks without retraining, filling a critical gap in the TIM field.
2. **Well-Grounded Attention Manipulation**: The method’s core innovations—SAI and CAR for text removal, and attention-based loss functions for style/content control—are rooted in in-depth analysis of attention properties (self-attention for style, cross-attention for content). This mechanistic understanding ensures the design is not ad-hoc, and the clear mathematical formalization of attention manipulation and loss functions enhances the framework’s credibility.
3. **Comprehensive Ablation Studies Validate Component Efficacy**: Ablation experiments systematically verify the necessity of key components—SAI+CAR for text removal, and the combination of cross-attention content loss and self-attention style loss for controllable inpainting. These studies clarify how each module contributes to balancing task performance (e.g., reducing hallucinations in removal, balancing accuracy and style fidelity in editing), strengthening the persuasiveness of the framework.
4. **Valuable Benchmark for TIM Evaluation**: OmniText-Bench addresses the lack of unified evaluation benchmarks for multi-task TIM. It provides diverse samples (input images, masks, style references, ground-truth) covering practical scenarios, enabling fair comparison of generalist TIM methods and promoting further research in the field.
5. **Strong Compatibility and Practicality**: As a training-free framework, OmniText builds on the TextDiff-2 backbone and can be easily integrated into existing diffusion-based text synthesis pipelines without modifying model architectures or requiring additional training data. This low deployment cost makes it highly applicable to real-world scenarios like poster design and product packaging editing.

**Weaknesses:**

1. **Insufficient Qualitative Results for Edge Cases**: The paper provides qualitative results for typical TIM tasks but lacks side-by-side comparisons for edge cases, such as long-text manipulation (over 10 characters), text on complex textures (e.g., patterned fabrics), or low-resolution input images. This makes it difficult to evaluate the framework’s robustness in challenging practical scenarios.
2. **Vague Explanation of Hyperparameter Tuning**: OmniText introduces hyperparameters (e.g., weight coefficients for content/style loss) but does not explain how to tune them for different tasks or datasets. The lack of guidance on hyperparameter adaptation may hinder its usability, especially for users without expertise in TIM.

**Questions:**

Please refer to the detailed points I raised in the "Weakness" section and respond to each numbered item in your rebuttal with clarifications.

---

> ### Author Response · Authors · 2025-11-21
>
> We thank the reviewer for the detailed review and insightful comments.
> Please refer to the Appendix attached to the updated main paper (red text or red caption).
> Below, we address the identified weaknesses and related questions.
>
> ## Response to Weakness: Insufficient Qualitative Results for Edge Cases
>
> We thank the reviewer for pointing out the lack of qualitative results for edge cases.
> To address this concern, we have updated our Appendix and added additional qualitative results in **Section M**:
>
> 1. **Long-text manipulation (over 10 characters).**
> In **Lines 2373–2413** and **Fig. 25**, we present challenging cases where the target text contains more than 10 characters.
> Across these examples, our method qualitatively achieves the best rendering accuracy and style fidelity compared to the backbone.
> We note that, similar to all other compared text generation methods, our framework does **not** aim to support very long, multi-line text (e.g., sentence- or paragraph-level generation).
> Such scenarios are outside the scope of our work and can be addressed by a separate line of research [1].
>
> 2. **Text on complex surfaces and textures.**
> In **Lines 2414–2427** and **Fig. 26**, we provide additional results on complex surfaces such as curved or highly deformable objects.
> On crumpled shirts, our method better preserves style fidelity and produces fewer artifacts than the baseline.
> For extremely complex textures such as crumpled paper with intricate textures, our method sometimes partially follows the surface curvature or locally overlay the background.
> Note that, 3D-aware (or depth-aware) text generation is still an open problem due to the unavailable training dataset that has many intricate textures.
>
> 3. **Low-resolution and low-quality inputs.**
> We further include examples of low-resolution images and images with some artifacts (**Lines 2429-2477** and **Fig. 27**), showing that our method still produces accurate text generation with strong style fidelity in these challenging cases.
> We emphasize that many images in the ScenePair dataset are originally low-resolution and are resized to $512 \times 512$ for our experiments.
> Thus, the quantitative results reported on ScenePair already reflect performance under low-resolution conditions.
>
> ## Response to Weakness: Vague Explanation of Hyperparameter Tuning
> We appreciate the reviewer's concern regarding hyperparameter tuning and usability.
> 1. **Scope of hyperparameter tuning in our experiments.**
> Our method is robust to the choice of loss weights, as shown in **Table 16 in the Appendix**.
> Thanks to this robustness, we reuse the same loss weights for OmniText-Bench without any additional hyperparameter tuning.
> Therefore, when applying OmniText to a new task or dataset, one can simply adopt the same hyperparameters that we use for ScenePair.
>
> 2. **Added guidance for practical hyperparameter selection.**
> To better address the reviewer's concern on usability, we update the Appendix and have added **Section G** in the Appendix, where we provide practical guidelines for hyperparameter selection.
>
> ## Reference
> [1] Wang, Alex Jinpeng, et al. "Beyond Words: Advancing Long-Text Image Generation via Multimodal Autoregressive Models." arXiv preprint arXiv:2503.20198 (2025).

---

### Meta-Review · Area_Chair_91ZF · 2026-01-07

**Summary:**

Initial scores were mixed, from 4 to 8. Reviewers liked the unified framework, but they worried about inference speed and whether the method depends too much on TextDiff-2. The rebuttal addresses generalization: the core pieces, self-attention modulation and the grid trick, work with other U-Net backbones such as UDiffText and DreamText, support Chinese via TextSSR, add qualitative results on hard cases like long text and curved surfaces, and include a comparison to GPT-4o. The AC recommends acceptance for the zero-shot style and spatial control of this submission.

**Reviewer Concerns:**

Addressed:

Generalization: Shown on UDiffText and DreamText, and extended to Chinese via TextSSR.

Hard cases: Added results for long text, wrinkles and other complex textures, curved surfaces, and low-resolution inputs.

Large models: Added comparisons to GPT-4o and Gemini, which do not match the same spatial masking of OmniText.

Hyperparameters: Added sensitivity results over loss weights and iteration counts.

Outstanding:

Speed: Still slow, about 70 seconds per image, because it uses iterative latent optimization. Specialist baselines run in about 2 seconds.

**Reviewer Scores:**

S1wy: likely 6. The added edge-case results and guidance answer the main requests.

Qwan: likely 8. The large-model comparison supports the main claim.

ziMs: likely 4. The generalization concern is mostly resolved.

8xFf: likely 6. Backbone transfer and Chinese results reduce the earlier doubts.

---

### Decision · Program_Chairs · 2026-01-26

Accept (Poster)